# Assessment of vertical air motion among reanalyses and qualitative comparison with VHF radar measurements over the two tropical stations

Kizhathur Narasimhan Uma [1], Siddarth Shankar Das [1], Madineni Venkat Ratnam [2], and Kuniyil Viswanathan Suneeth [1,3]

[1]Space Physics Laboratory, Vikram Sarabhai Space Centre, ISRO, Trivandrum-695022, India
[2]National Atmospheric Research Laboratory, Dept. of Space, Gadanki-517112, India
[3]India Meteorological Department, Minsitry of Earth Sciences, New Delhi-110003, India

**Correspondence:** Dr. K. N. Uma (urmi_nmrf@yahoo.co.in)

**Abstract.**

Vertical wind ($w$) is one of the most important meteorological parameters for understanding a range of different atmospheric phenomena. Very few direct measurements of $w$ are available so that most of the time one must depend on reanalysis products. In the present study, assessment of $w$ among selected reanalyses, (ERAi, ERA5, MERRA-2, NCEP/DOE-2 and JRA-55) and qualitative comparison of those datasets with VHF radar measurements over the convectively active regions Gadanki (13.5ºN and 79.2ºE), India and Kototabang (0º and 100.2ºE), Indonesia are presented for the first time in the troposphere and lower stratosphere. The magnitude of $w$ derived from reanalyses is 10-50 % less than that from the radar observations. Radar measurements of $w$ show downdrafts below 8 to 10 km and updrafts above 8-10 km over both locations. Inter-comparison between the ensemble of reanalyses with respect to individual reanalysis shows that ERAi, MERRA-2 and JRA-55 compares well with the ensemble compared to ERA5 and NCEP/DOE-2. There is no significant improvement in $w$ due to the effect of different spatial sampling for reanalysis data around the Gadanki station. Directional tendency shows that the percentage of updrafts captured is reasonably good, but downdrafts are not well captured by all reanalyses. Thus, caution is advised when using $w$ from reanalyses.

## 1   Introduction

Vertical air motion ($w$) in any region of the Earth's atmosphere reflects the structure and dynamical features of that region. Importantly, in the lower part of the atmosphere, sudden widespread changes in the weather are usually associated with variations in $w$. The magnitude of $w$ is a factor of ten or more smaller than the horizontal wind; nevertheless, it is crucial in the evolution of severe weather (Peterson and Balsley, 1979). Adiabatic cooling associated with upward motion leads to the formation of clouds and precipitation and adiabatic warming associated with downward motion leads to the dissipation of clouds. In addi-

tion, subsidence leads to adiabatic warming, which results in the formation of stable inversion layers. Extensive studies have been done on the relationships between $w$ and precipitation/convection over the tropics (Back and Bretherton, 2006; Uma and Rao, 2009a; Rao et al., 2009; Uma et al., 2012, and references therein). Thus, $w$ plays a vital role in day-to-day changes in the weather. Different scales of variability exist in $w$ ranging from microscale to meso synoptic, and planetary - scales (Uma and Rao, 2009b). It also controls energy and mass transport between the upper troposphere and lower stratosphere (Yamamoto et al., 2007; Rao et al., 2008). In a nutshell, knowledge of $w$ is helpful for evaluating virtually all physical processes in the atmosphere. Hence precise measurements of $w$ could serve a guiding factor for studying many processes in the atmosphere.

The small magnitudes of $w$ make it very difficult to measure, as the errors involved in measurements often exceed the actual values. Direct and indirect methods exist to measure $w$ (e.g. Doppler measurements using radars for profiling, sonic anemometers in the boundary layer, radiosondes and also aircrafts) as well as indirect computational methods (e.g., adiabatic, kinematic and quasi-geostrophic vorticity/omega methods). With respect to radiosondes, very few studies have calculated $w$. Wang et al. (2009) derived $w$ from radiosonde and dropsondes, however the authors pointed out several uncertainties like requirement of high resolution radiosonde data, amount of helium gas associated with such retrievals and,also accuracy of the estimated $w$ was not quantified. Zhang et al. (2019) estimated $w$ using a descending radiosonde system. The authors pointed out the uncertainties involved especially with radiosonde descent speed, calculation of drag coefficient and also on the validation of the retrieval's on $w$ obtained. Using aircrafts Schumann (2019) studied the relationships between horizontal kinetic energy spectra of $w$ and horizontal divergence of the divergent horizontal wind components, by separating it from the rotational wind components by known Helmholtz decomposition methods. Radars provide the direct measurement of $w$ and hence remote sensing measurements of $w$ are thus restricted to locations where radars are situated.

In general, $w$ is derived diagnostically from horizontal winds and temperature, which is an indirect estimation. This estimation gives a general view on the distribution of ascending and descending motion on the synoptic-scale within the quasi-geostrophic framework (Tanaka and Yatagai, 2000; Jagannadha Rao et al., 2003). Reanalyses evaluate the vertical pressure velocity (omega) using indirect estimation (e.g., Dee et al., 2011). Any reanalyses products assimilate as much as 107 observations per day, which is inclusive of both conventional (radiosonde, tower, aircrafts, wind profilers (wherever possible), etc.) as well as various satellite observations. However, reanalyses combine both observations and model outputs to produce systematic variation in the atmospheric state (e.g., Fujiwara et al., 2017). It is to be noted that the $w$ provided by any reanalysis data center is estimated indirectly from the horizontal wind components and temperature, which itself has mismatch among various reanalyses data (e.g., Das et al., 2016; Kawatani et al., 2016). Thus, this can possibly induce the discrepancy in the estimated $w$ among various reanalyses. For example, in the kinematic method, omega is estimated by integrating the mass continuity equation assuming inviscid adiabatic flow. However, this kinematic estimate suffers from uncertainties in the observations as omega is estimated from horizontal divergence (Tanaka and Yatagai, 2000). This source of uncertainty is particularly important for reanalyses, where assimilation increments in horizontal winds may be comparable to the uncertainty. A 10% error in the wind may lead to a 100% error in the estimated divergence (Holton, 2004). Omega from the thermodynamic energy equation is less sensitive to horizontal winds as it mainly depends on the temperature gradient. However, in this method the local rate of change in temperature must be measured accurately, meaning that observations must be taken at frequent intervals in

time to estimate $\delta T/\delta t$ accurately (Holton, 2004). This methodology fails in areas of strong diabatic heating, especially where condensation and evaporation are involved. The quasi-geostrophic method for estimating omega neglects ageostrophic effects, friction and diabatic heating (Stepanyuk et al., 2017). It is to be noted from the above discussions that calculating $w$ from indirect estimation has more uncertainties. Hence reanalyses that use indirect estimation, involve underlying approximations and assimilations and are not error-free (Kennedy et al., 2011). Other indirect methods can be used to derive $w$ from radar measurements in the middle and upper atmosphere, where direct measurements of $w$ are not possible due to technical constraints. These methods include Doppler weather radar, Medium Frequency (MF) radar and meteor radar. Doppler weather radar uses an indirect method to calculate $w$ (Liou and Chang, 2009; Matejka, 2002).

Very-high frequency (VHF) and ultra-high frequency (UHF) vertical pointing radars are the most powerful tools for determining $w$ with high temporal and vertical resolution. However, the magnitude may still not be directly comparable between reanalysis products and observations as the reanalyses provide the intensity of $w$ over wide areas ($> 25$ km$^2$), whereas the radar measurements provide information for a narrower column over a single location. Thus, the best way to assess reanalysis estimates of $w$ against radar measurements is to compare its directional tendencies. A number of studies have evaluated $w$ across reanalyses (in the context of trajectories, wave activity, large-scale motion, etc.), so the primary novelty of this work is the evaluation against radar observations.

Stratosphere–troposphere Processes And their Role in Climate (SPARC) has initiated an activity known as SPARC Reanalysis Intercomparison Project (S-RIP) (Fujiwara et al., 2013; Fujiwara and Jackson, 2013; Fujiwara et al., 2017). The main objectives of S-RIP are to evaluate different reanalysis products, their differences with respect to different measurements, and also to suggest improvement for better usage by the scientific community (http://s-rip.ees.hokudai.ac.jp). The present study hence focuses on the assessment of $w$ in the troposphere and lower stratosphere among various reanalyses using VHF radar measurements from two tropical stations where the convective activity is frequent: Gadanki and Kototabang. Evaluations of this type are critically important as reanalyses estimates of $w$ are widely used by the scientific community to understand and simulate a variety of atmospheric processes. In section 2, the data and methodology are described. Section 3 provides results and discussion followed by summary and concluding remarks in section 4.

## 2 Data and Methodology

### 2.1 Radar measurements

Remote sensing measurements of $w$ are obtained from the Indian Mesosphere-Stratosphere-Troposphere Radar (IMSTR) located at Gadanki (13.5°N and 79.2°E), India and the Equatorial Atmosphere Radar (EAR) located at Kototabang (0.2°S and 100.2°E), Indonesia. Figure 1a and 1b show the topography map of the location of both the radars, i.e. Gadanki and Kototabang respectively, generated by using the Shuttle Radar Topography Mission (SRTM) data (Farr et al., 2007). Gadanki is located in the southern peninsula of tropical India, about 90 km off the east coast and it is surrounded by hills. Kototabang is located in the western part of Sumatra Island and EAR is situated in the mountainous region with the highest peak of about 2 km. Both the IMSTR and EAR are pulsed coherent radars operating at 53 MHz and 47 MHz, respectively. These instruments are

used to estimate *w* by measuring the Doppler shift in the vertical beam. The technical details and operational parameters of the IMSTR have been given by Rao et al. (1995) while those for the EAR have been given by Fukao et al. (2003). Both the radars specifications, parameters including velocity resolution used for the present measurements are listed in Table 1.

In the present study measurements of *w* from VHF radars are used to assess vertical motion between the surface and the lower stratosphere. Data collected from the IMSTR between 17:30 and 18:30 LT (LT=UTC+5:30 hr) from 1995 to 2015 are analyzed using the adaptive method (Anandan et al., 2001). This is the common operational mode of the IMSTR for deriving the winds and represents the only data available for such a long period of time. The three components of wind : zonal, meridional and vertical can be computed with the radial velocity obtained in atleast 3 non-coplanar directions. However, for the present analysis we have computed the *w* directly only using the vertical beam using equation (1)

$$w = (-\lambda/2)f_d \qquad (1)$$

Where, $\lambda$ is the radar wavelength (in cm) and $f_d$ is the Doppler velocity (Hz).In general, 4-8 vertical profiles are averaged to create daily 16:30-17:30 IST (11:00-12:00 UTC) averaged profiles. Averaging is conducted using the arithmetic mean as it represents the central tendency, which is generally used for wind averaging. In a vertically pointing beam, signal-to-noise ratio (SNR) decreases with height except in stable layers (like the tropopause) and in the presence of strong turbulence. Above 25 km, the SNR becomes constant in the absence of atmospheric signals. Data in this region can be therefore treated as noise and used to estimate the threshold SNR (Uma and Rao, 2009b). Noise levels estimated in this way lie between -17 dB and -19 dB with a $2\sigma$ value of 3 dB (where $\sigma$ is the standard deviation). Thus data having SNR less than -15 dB are discarded from the present analysis. Data from intense convective days (checked for individual profiles), defined as *w* being less/greater than $\pm 1$ ms$^{-1}$ are also discarded as these data severely bias the climatological mean *w* (e.g., Uma and Rao, 2009b). The data discarded is less than 1% of the total data. Quality control metadata for the EAR measurements are available online (http://www.rish.kyoto-u.ac.jp/ear/data/index.html). The EAR operates continuously and this study uses hourly data (diurnal data of single day) of *w* computed using the vertical beam (equ. (1)) from 2001 to 2015. The EAR data during convective periods are eliminated following the same criteria as for the IMSTR, a second screening step. Each full diurnal cycle (after removing convective profiles) is averaged and considered as a single daily profile for the EAR.

## 2.2   Accuracy and uncertainty in the *w* measured from Radar

The assumption in the radar measurements of wind components is the spatial homogeneity in the given time frame, when we used 3 non-coplanar beams (e.g., two off-zenith and one vertical). Thus, to avoid the bias, we use only vertical beam (equ. (1)) for the direct estimation of *w* which also provides a better time-resolution (Peterson and Balsley, 1979; Koscielny et al., 1984). The accuracy of the *w* measured made using the vertical beam of VHF radar depends on the alignment of the beam along the zenith direction. Any error in the beam pointing would mean that the line-of-sight velocity measured by the radar will have a component of the horizontal wind (Huaman and Balsley, 1996). The beam pointing error is found to be $\pm 0.2^{\circ}$ off-zenith, which was provided by calibrating the beam pointing with a known radio source Virgo-A (Damle et al., 1991; Rao et al., 1995) and Cygnus-A for EAR (Fukao et al., 2003). The uncertainty in the *w* due to beam pointing error by an angle ($\theta$) with a horizontal wind (u) is given by (u.sin$\theta$). Thus, with a horizontal wind of 10 m s$^{-1}$ and beam pointing error of 0.2

degree turns out to be 0.03 m s$^{-1}$ uncertainty in the $w$ measured from VHF radar. The beam pointing accuracy can further be determined by comparing the $w$ obtained using two orthogonal polarizations, i.e., east-west and north-south polarizations,

which are phased independently. Significant correlation was observed between both the polarizations, suggesting that the radar measures the true $w$ (Viswanathan et al., 1993). In addition, Rao et al. (2008) also estimated the vertical velocities from zenith beam and compared it with those estimated from 10-degree off-zenith beams using IMSTR. The differences were observed to be meager, which shows that the error due to beam pointing is negligible.

Tilting of reflecting layers contributing to the diffuse reflection can also adversely bias the mean $w$ (Röttger, 1980). These
130 tilting layers can be due to the presence of Kelvin-Helmholtz instabilities (Muschinski, 1996), gravity waves, which includes inertia-gravity waves and mountain waves and causes imbalance in the echo power between the two polarizations in the same plane (Yamamoto et al., 2003). Rao et al. (2008) estimated the echo power imbalance in the east-west and north-south polarizations for both EAR and IMSTR and found the difference to be within $\pm 1$ dB, statistically indicating the bias due to the tilting layers is negligible over both the locations.

Nastrom and VanZandt (1994) proposed that $w$ can be biased by gravity waves. Thus, Rao et al. (2008) have investigated the biases caused by gravity waves by calculating the variances and found that downward wind measurements below 10 km are essentially unaffected by gravity waves. It is also to be noted that the topography over the two locations can generate mountain waves, if strong low-level winds are prevailing. Strong low-level winds are prevalent over Gadanki only from June to August and during these months, there is a critical level existing between 6 and 7 km due to the presence of strong wind shear, which
will not support the propagation of mountain waves to higher altitudes. This wind shear between 6 and 7 km exists throughout the year over Kototabang. Hence the effect of mountain waves will be minimal over both these locations on $w$. Their analysis clearly showed that the mean downward motion below 10 km and upward motion above 10 km are real and not caused by measurement biases, and also that the known biases do not change the direction of the background $w$ when measurements are averaged over longer period of 10 years.

## 145 2.3 ERA-Interim (ERAi)

ERAi is global reanalyses data which is developed by European Centre for Medium-Range Weather Forecasts (ECMWF). The data assimilation scheme used is 4D-Var of the upper-air atmospheric state and have effectively anchored both satellite and in-situ observations. This scheme updates parameters that define bias corrections required for satellite observations. The model has improved in the representation of moist physical processes. Advances have also been made with respect to soil hydrology
and snow in land surface models. The detail of the model is given in (Dee et al., 2011). We use 6-hourly vertical velocities from the ECMWF Interim reanalysis (ERAi) from 1995 to 2015. The grid resolution of ERAi is 0.75º (latitude) x 0.75º (longitude). The nearest grid points are taken for Gadanki (13.68º N, 79.45ºE) and Kototabang (0.35º S, 100.54ºE). ERAi has 37 pressure levels, from surface up to 1 hPa. The pressure coordinate is converted into pressure height by using the hypsometric equation (Holton, 2004), which is followed for other reanalyses also. The difference between the pressure height and geometric height
(radar measurements) is found to be <100 m, which do not bias the results as the radar range resolution itself is 150 m. In the present analysis, we restrict the dataset up to 21 km, which is about 50 hPa, as that is the maximum radar range.

## 2.4   ERA5

ERA fifth-generation (ERA5) is the atmospheric reanalysis produced by ECMWF. It is an improved version of ERAi. The data assimilation scheme used is 4D-Var and it assimilates the NCEP stage IV quantitative precipitation estimates produced over the USA by combining precipitation estimates from the Next-Generation Radar (NEXRAD) network with gauge measurements. The moist physics scheme is improved by including freezing rain. The long wave radiation scheme is modified in ERA5. The evolution of the top soil layer, snow and sea ice temperatures are included. It uses observations from various satellites which include upper air temperature, humidity and ozone. It also use bending angles from GNSS. It provides much higher spatial (30 km) and temporal resolution (hourly) from the surface up to 80 km (137 levels). ERA5 also features much improved representation especially over the tropical regions of the troposphere and better global balance of precipitation and evaporation. Many new data types not assimilated in ERAi are ingested in ERA5 (Hoffmann et al., 2019). The grid resolution of ERA5 is 0.28º (latitude) x 0.28º (longitude). The details are available in (Hersbach et al., 2020). We have taken hourly data from ERA5. The nearest grid points are again taken for Gadanki (13.63ºN, 79.31ºE) and Kototabang (0.14ºS, 100.40ºE), and the data period is 2002-2015.

## 2.5   MERRA-2

The Modern-Era Retrospective analysis for Research and Applications, version 2 (MERRA-2) is the latest reanalysis of the modern satellite era produced by the National Aeronautics and Space Administration's (NASA) Global Modelling and Assimilation Office (GMAO). The scheme used in MERRA-2 is an improved version of MERRA. It uses a three-dimensional variational (3D-Var) algorithm based on the grid point statistical interpolation and also uses an incremental analysis update. It assimilates bending angle observations, satellite radiances from both polar as well as geostationary infra-red and microwave sounders. In addition it also assimilates water vapor and ozone. MERRA-2 includes aerosol analysis and provide data for 42 pressure levels from the surface to 0.01 hPa with a temporal resolution of 3 h and horizontal resolution of 0.5º (latitude)x 0.625º (longitude). We used MERRA-2 Assimilation (ASM) data. Details have been provided by Gelaro et al. (2017)). The nearest grid points are used for Gadanki (13.5º N, 79.37º E) and Kototabang (0.14º S, 100.00º E), with data spanning from 1995 to 2015.

## 2.6   NCEP/DOE-2

The National Center for Atmospheric Research and Department of Energy (NCEP/DOE-2) reanalysis is an updated version of NCEP-1 by fixing the known processing errors in NCEP-1. The variational scheme used is 3D-Var and it provides more accurate pictures of soil wetness and near-surface temperature over land, the land surface hydrology budget, snow cover, and radiation fluxes over the ocean. It is based on the NCEP operational model with a horizontal resolution of 209 km and 28 vertical levels. The temporal coverage is four times per day. NCEP/DOE-2 products are improved relative to NCEP-1, having fixed errors and updated parameterizations of physical processes, as evaluated by Kanamitsu et al. (2002). The grid resolution

of NCEP/DOE-2 is 2.5° (latitude) x 2.5° (longitude). The data for the present study covers from 1995 to 2015 and is extracted at the nearest grid points to Gadanki (12.5° N, 77.5° E) and Kototabang (0°, 100.00° E).

## 2.7 JRA-55

The Japanese 55-year reanalysis (JRA-55) is an updated version of the earlier JRA-25 with new data assimilation and prediction systems (Kobayashi et al., 2015). New radiation schemes, higher spatial resolution and 4D-var data assimilation with variational bias correction for satellite radiances have been used to generate the JRA-55 products. This reanalysis includes variation in greenhouse gas concentrations with time, as well as the new representations of land surface parameters, aerosols, ozone and sea surface temperature. The grid resolution of JRA-55 is 1.25° (latitude) x 1.25° (longitude). The nearest grid points are taken for Gadanki (13.75° N, 78.75° E) and Kototabang (0°, 100° E) and the data period is 1995-2015.

For all the reanalyses data, $w$ (in cm s$^{-1}$) is estimated using equation (2) :

$$w = (-1/g)\omega(RT/p) \qquad (2)$$

Where, $\omega$ is the vertical velocity in pressure coordinates (in Pa s$^{-1}$), T is the absolute temperature (K), p is the atmospheric pressure (hPa) and R (=287 J kg$^{-1}$ K$^{-1}$) is the gas constant for dry air. To compare measured vertical wind with the reanalysis products, we take the reanalysis data corresponding to 12 UTC for Gadanki and the daily mean for Kototabang. The details of the schemes used in reanalysis are provided in Table 2.

## 3 Results and Discussion

Figure 2 shows the inter-comparison of layer averaged daily $w$ measured from IMSTR with different reanalyses (ERAi, ERA5, MERRA-2, NCEP/DOE-2, and JRA-55) over Gadanki for (a) January 2007, and (b) August 2007. Both radar and all the re-analyses data sets are taken at 12 UTC, and the month and year are chosen in such a way to have maximum days of radar observations in two different seasons (winter and summer). Similarly, EAR observation is also compared with different re-analysis data but for January 2008 and August 2008 as shown in Fig. 3. However, both EAR and reanalysis data are diurnal averaged (24 hrs). It is observed that the magnitude of $w$ measured from radar observations is an order higher than the reanalysis data over both the locations (Gadanki and Kototabang). Most of the time, reanalysis data are comparable in direction with radar observations, whenever updrafts are observed. It is also observed that there is mismatch between the $w$ estimated in the different reanalyses.

Gage et al. (1992) described that by averaging radar data for a long-period of time can give a better measurement of $w$ in clear-air condition and the authors have used three years data to arrive at the above conclusion. Thus in this context, we have taken 20 years of data for averaging. Figure 4 shows the climatological monthly mean altitude profile of $w$ obtained from the IMSTR (observations) and the ERAi, ERA5, MERRA-2, NCEP/DOE-2 and JRA-55 reanalysis data over Gadanki. Although the magnitudes are of the same order between the observations and reanalyses, significant differences are identified in the figures. Convective days are discarded from the radar data (observations) as mentioned in the previous section and those days are also eliminated from all reanalysis data sets. The quantitative differences may be attributed to the spatial averaging implicit

in the reanalyses products, whereas the radar measurements are for a single point. Thus we only discuss the tendency of $w$ as it is used to represent the variation of $w$, rather than its magnitude. The IMSTR observations show updrafts between 8 and 20 km from December to April, with the largest values in the tropical tropopause layer (TTL, 12-16 km), These features are not reproduced by any of the reanalyses, which all show downdrafts from December to April between 1 km and the tropopause level (mean tropopause is 16.5 km). By comparison, downdrafts are observed in the IMSTR below 6 km in April, which

may be attributed to pre-monsoon (March-May) precipitation and evaporation (Uma and Rao, 2009a). $w$ in ERAi differs in both magnitude and direction from other reanalyses, especially in the lower troposphere from March to June. Meanwhile, the magnitude of $w$ in ERA5 is a little larger than that in the other reanalyses from May to June. Updrafts are observed in the TTL by the IMSTR during June, when all reanalyses show similar features but only located below the TTL. During July and August both the radar observations and the reanalyses show updrafts in the vicinity of the TTL. Updrafts are observed in the

TTL from September to November but the peak in the updrafts is shifted lower than that observed by the IMSTR. Below 8 km, the IMSTR shows downdrafts from April to October. The reanalyses data are unable to reproduce downdrafts above 2 km.

     We have also analyzed $w$ from the EAR (Kototabang) where the observations are available for the full diurnal cycle (measurements of hourly averages for 24 hrs of observations). All reanalyses data over Kototabang are averaged for the full diurnal cycle. Figure 5 shows the monthly mean climatology of daily mean $w$ from the EAR observations and the five reanalyses over

Kototabang. All the reanalyses agree well with each other over Kototabang. The updrafts in the TTL are well reproduced by all five reanalyses although the magnitude and vertical location of the maximum in $w$ remain lower than observed. However none of the reanalyses reproduces the downdrafts. A distinct bimodal distribution in $w$ from May to September (two peaks between 8-10 km and 14-17 km) with a local minimum between 12 and 13 km is observed in the EAR measurements which are not observed in the reanalysis. The magnitudes of both updrafts and downdrafts are larger than those observed over Gadanki.

JRA-55 produces the largest $w$ among the reanalyses. The monthly means show significant differences in the direction of $w$ between the observations and the reanalyses below 6 km.

     Gage et al. (1992) studied the long-term diurnal variability of $w$ at Christmas Island (2$^{o}$N) and found the $w$ varies between $\pm 4$ cm s$^{-1}$. The observations showed updrafts below 4 km, downdrafts between 4-14 km and updrafts above 12 km. Gage et al. (1991) have explained that the downward motion in the troposphere is consistent with a heat balance in the clear-air between

adiabatic warming of descending air and radiative cooling to space. The ascending motion in the upper troposphere and lower stratosphere is due to large diabatic heating caused by ice particle in the cirrus. Rao et al. (2008) have shown the long-term (11 years) mean of $w$ over Gadanki and Kototabang and found $w$ varies between -0.3 to +0.6 cm s$^{-1}$. The authors observed downdrafts below 6 km and updrafts above it in all the seasons. The mean pattern of $w$ profile observed by radars over all the tropical sites (i.e. Christmas Island, Gadanki and Kototabang ) show similar characteristics and explain that the vertical

transport of air from the troposphere to the lower stratosphere is a two-step process as discussed by Rao et al. (2008). Uma and Rao (2009b) have reported the diurnal variation (using hourly data) of $w$ in different seasons, although their observations had only 1-2 diurnal cycles per month over Gadanki. They found significant variations in the seasonal variability of diurnal cycle as large as $\pm 6$ cm s$^{-1}$ over Gadanki using IMSTR. The present observations are limited to 16:30 to 17:30 IST, with all reanalyses data over Gadanki taken at 12 UTC (17:30 IST). Thus, time-averaged climatological mean biases can be neglected.

To establish the robustness of the results we have used different averaging procedures to assess the consistency of the variability in $w$ at monthly scales. Monthly mean climatological profiles of $w$ from radar observations and various reanalyses over Gadanki and Kototabang are shown in Fig. A1 (supplementary). Downdrafts in the troposphere are not captured by any of the reanalyses over either location. By contrast, updrafts in the TTL are generally reproduced in the monthly mean, though their magnitudes are often underestimated by the reanalyses. ERAi underestimates the magnitude of both updrafts and downdrafts over Gadanki, while NCEP/DOE-2 underestimates the magnitude of updrafts over Kototabang. Monthly means calculated over five-year periods from both the radar data and ERAi are shown in Fig. 6 for Gadanki and Fig. 7 for Kototabang. The reanalysis shows similar behavior to the overall climatology in each five-year average. The overall patterns of updrafts and downdrafts in the radar measurements of $w$ are also similar, indicating a consistent performance of the radar over the full 20 year analysis period.

To further elucidate potential biases in the results due to averaging, we have taken ERA5 at 12 UTC and compared it to the daily mean (obtained by averaging $w$ at different times of the day) to show that the sampling restrictions at Gadanki do not bias the results obtained. Figures 8 and 9 show the mean $w$ obtained at 12 UTC and also the mean obtained by averaging hourly analyses for each day for Gadanki and Kototabang, respectively. ERA5 is chosen for this evaluation as the data are available at one-hour intervals. The analysis shows some differences in the magnitude of $w$, with 12 UTC generally showing larger magnitudes compared to the daily means over Gadanki (although no such systematic differences are observed in Kototabang). The directional tendencies are also similar in both the profiles at both locations. This analysis shows that the results are not biased by taking data only at 12 UTC over Gadanki.

Our analysis to this point shows the level of consistency between the features observed by the radar and those in the reanalysis. To further understand the relative differences among the reanalyses we perform a monthly mean comparative analysis among the reanalyses, as shown in Figures 10 and 11 for Gadanki and Kototabang, respectively. We take an ensemble mean of all the five reanalyses and then subtracted the ensemble mean from each reanalysis. The differences are less than $\pm0.5$ cm s$^{-1}$ during December-January-February (DJF, winter). During MAM, the difference between the ensemble and reanalysis show $\pm2$ cm s$^{-1}$ below 5 km. Below 5 km NCEP/DOE-2 and ERAi is less, whereas ERA5, Merra-2 and JRA-55 are more than the ensemble. The difference above 6 km is less than $\pm0.5$ cm s$^{-1}$ above 6 km. JRA-55 shows a good comparison with the ensemble and above 10 km all the reanalyses the differences are minimal with the ensemble. During the monsoon (JJA), the difference is comparatively high in June compared to July and August. NCEP/DOE-2 and ERA5 are more and other reanalyses are less than the ensemble, however during July and August NCEP/DOE-2 it is less in the upper troposphere (10-18 km). Merra-2 and ERAi shows a good comparison with respect to the ensemble during July and August, JRA-55 also shows a good comparison in addition to Merra-2 and ERAi. During SON, the differences are comparatively less than MAM and JJA. The difference is less than $\pm0.5$ cm s$^{-1}$ during October and November except in September between 10 and 15 km where ERA5 and Merra-2 are more and ERAi and NCEP/DOE-2 are less than the ensemble. In general, ERA5 and NCEP/DOE-2 shows considerably more difference with the ensemble and other reanalyses (ERAi, Merra-2 and JRA-55) compare well with the ensemble.

Over Kototabang (Figure 11), it is interesting to note the difference between the ensemble and different reanalyses show a consistent pattern during all the months. JRA-55 and ERAi show good comparison with the ensemble, as the differences are

less than $\pm0.2$ cm s$^{-1}$ in all the seasons, except in November where it exceeds $\pm0.5$ cm s$^{-1}$ in the lower and middle troposphere. Merra-2 is more and NCEP/DOE-2 is less than the ensemble at all the height regions. ERA5 is less below 10 km and more above with respect to the ensemble.

There may be some probable reasons for the differences in the $w$ measured by observations and those retrieved from re-analysis. The main bias in $w$ might occur in the reanalysis due to the following (1) Indirect estimation of omega, (2) local
topography influence in the reanalysis, (3) use of different schemes in the boundary layer, (4) interactions between subgrid physical parameterizations and the large-scale flow and (5) spatial and temporal sampling. However, it is difficult to address the above issues other than the spatial and temporal sampling. To elucidate the spatial-temporal averaging on the vertical veloc-ity we have chosen different grid resolutions with Gadanki as a centroid and the map is shown in Fig. 12a. G1 to G5 represent different grid resolutions, varying from 0.7$^o$ to 5$^o$.The data chosen is for January and July 2007 from ERAi. The height profile
of w at different grid resolution and time is shown in Fig. 12b for January and in Fig. 12c for July. It is observed that the grid resolution does not have any influence on the $w$. However, a significant change is observed between 00 and 12 UTC in the month of January which affected the diurnal mean in $w$ (shown in the last panel). The same is not reflected in the month of July. The result shows that the narrowing down the reanalysis data spatially (reducing the horizontal sampling) will not improve the retrieval of $w$ in any reanalyses.

The direction of $w$ is an essential metric for comparing the reanalysis with the observations. We therefore show the directional tendencies of reanalysis data relative to the radar measurements. The directional tendencies would be 100 % when all radar measurements at certain height range are reproduced by a reanalysis in terms of $w$ direction. Figure 13a shows the directional tendencies based on the IMSTR and the reanalyses over Gadanki, while Figure 13b shows the directional tendencies based on the EAR and the reanalyses over Kototabang. The directional tendency is calculated at each height for every month when the
radar or reanalysis data exceed 0.1 cms$^{-1}$ in either directions. The directional tendency for each month is estimated and then aggregated into seasons. These directional tendencies are given in terms of percentage of occurrence with respect to height. The tendency is calculated separately for updrafts and downdrafts.

Over Gadanki during DJF all reanalyses produce updrafts (simultaneously by both radar and reanalysis) less than 10% of the time throughout the profile. During MAM these ratios increase to around 15%, with NCEP/DOE-2 reproducing updrafts about
25% of the time. During JJA and SON, the percentage occurrence increases with the height from 25% to a maximum of 50% between 12 and 14 km. The percentage occurrence of updraft then decreases from 14 to 20 km. This tendency trend is similar for all reanalyses. The maximum ratio of updrafts over Gadanki is located between 12 and 15 km altitude. The percentage occurrence of downdrafts over Gadanki is also less than 50% at all levels. During DJF and MAM the reanalyses reproduce downdrafts 40 to 50% of the time, a much higher frequency than that for updrafts (<10%). This fraction decreases above 10
km. By contrast, the percentage of downdrafts reproduced during JJA and SON is less than that of updrafts, with frequencies less than 25% at all levels during these seasons.

Over Kototabang the percentage occurrence of updrafts increases with height in all seasons reaching a maximum of 75- 90% between 10 and 14 km. Above 14 km the percentage decreases to a minimum of 5% at 19 km. Updrafts are rarely reproduced by the reanalyses altitudes less than 4 km. It is important to note that none of the reanalyses reproduce daily mean downdrafts

exceeding 1 cm s$^{-1}$ except ERAi and ERA5 which reproduced downdrafts below 6 km. The percentage of downdrafts increases above 17 km where it reaches a maximum and show occurrence frequencies around 65 to 75% above 18 km.

## 4    Conclusions

The present study assesses the vertical motion ($w$) in reanalyses against radar observations in the troposphere and lower strato-
sphere from the convectively active regions Gadanki and Kototabang. The assessment is carried out for five different reanalyses:
ERAi, ERA5, MERRA-2, NCEP/DOE-2 and JRA-55. Measurements were collected using VHF radar at both locations. We
have used 20 years of data from Gadanki and 17 years of data from Kototabang. The following points summarize the results of
this unique study:

     1. The magnitude of $w$ obtained from reanalyses is underestimated by 10-50% relative to the radar observations.

     2. Observations over Gadanki showed updrafts from 8 to 20 km year around. All the reanalyses only reproduced this feature
during JJA and SON when magnitudes were larger than 0.5 cm s$^{-1}$ in the reanalyses data. However, the vertical location of the
updrafts differs between the observations and the reanalyses. Downdrafts below 8 km are not captured well by reanalyses data.

     3. Over Kototabang, all five reanalyses did not consistently reproduce downdrafts below 8 km in all months. Updrafts in the
UTLS are captured well; however, the peak in the vertical distribution of $w$ is different as over Gadanki.

     4. Inter-comparison between the ensemble and each reanalysis data shows the ERAi, MERRA-2 and JRA-55 compares well
with the ensemble compared to ERA5 and NCEP/DOE-2. Analysis also showed that the reduction in spatial sampling in all
the reanalyses data does not have significant improvement in the magnitude of $w$ .

     5. Assessment of directional tendencies show that updrafts are reproduced reasonably well in all five reanalyses data but
downdrafts are not reproduced at all.

     The present analysis reveals that downdrafts are not well captured in all the five reanalyses data. The location of the largest
updrafts is also shifted lower in reanalyses than in the observations. It is to be noted that $w$ measured from radar is limited
over a geographical area and thus the results may be valid to a limited region. However, the results demonstrate that how
approaches to generating global reanalysis products (encompassing different models, assimilation methods, spatial resolution,
etc.) can impact estimates of $w$. Hence, reanalysis data should be used with caution for representing various atmospheric motion
calculations (viz. diabatic heating, convection, etc.) that mainly depend on the direction of $w$.

*Data availability.*   Analysed data (both radars and reanalyses) used in this study can be obtained on request. Raw time series data are available
through open access in the following websites: For Indian MST Radar : www.narl.gov.in For EAR radar : www.rish-kyoto-u.ac.jp/ear/index-
e.html. ERAi, ERA5, JRA-55 and NCEP/DOE-2 were downloaded from https://rda.ucar.edu and MERRA-2 from https://disc.gsfc.nasa.gov.in

*Author contributions.* KNU conceived the idea for validation of vertical velocity among the reanalyses. SSD, MVR, and KVS collected and analysed the MST radar spectrum data. All the authors contribute for generation of figures, interpretation and manuscript preparation. The data used in the present study can be obtained on request.

*Competing interests.* The authors declare that there is no conflict of interest.

*Acknowledgements.* Authors would like to acknowledge all the technical and scientific staffs of National Atmospheric Research Laboratory (NARL) and Research Institute of Sustainable Humanosphere (RISH), who directly or indirectly involved in the radar observations. Thanks to all the reanalyses data centres for providing the data through the portal of Research data archival (RDA) of NCEP/UCAR. One of the author KVS thank Indian Research Organisation for providing research associateship during this study. We sincerely thanks all the referees, Executive Editor and Editor for their constructive comments and suggestions.

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

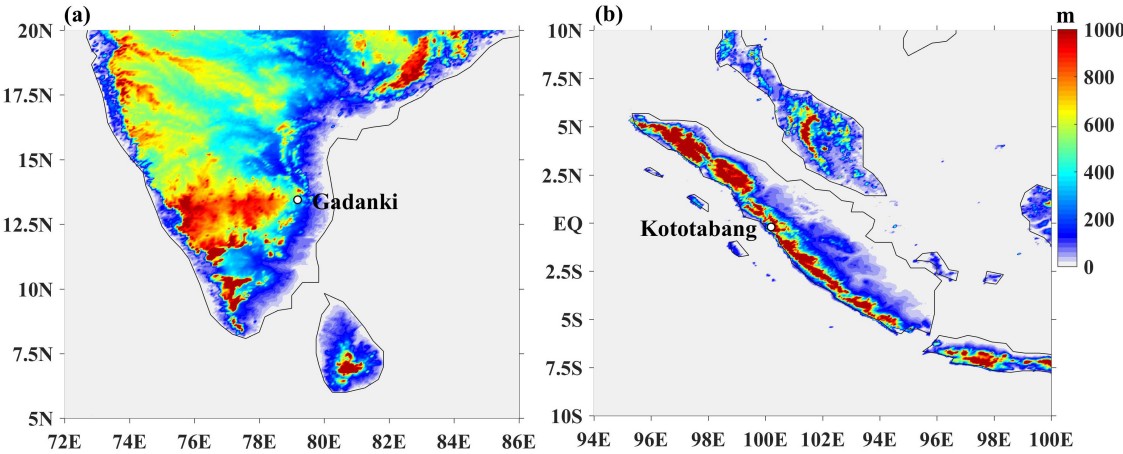

**Figure 1.** Topographical maps of the (a) IMSTR, and (b) Kototabang EAR sites in MSL, generated by using the Shuttle Radar Topography Mission (SRTM) data (Farr et al., 2007). Dots in the map indicate the radar locations.

Yamamoto, M. K., Fujiwara, M., Horinouchi, T., Hashiguchi, H., and Fukao, S.: Kelvin-Helmholtz instability around the tropical tropopause observed with the Equatorial Atmosphere Radar, Geophysical Research Letters, 30, https://doi.org/10.1029/2002GL016685, 2003.

Yamamoto, M. K., Nishi, N., Horinouchi, T., Niwano, M., and Fukao, S.: Vertical wind observation in the tropical upper troposphere by VHF wind profiler: A case study, Radio Science, 42, https://doi.org/10.1029/2006RS003538, 2007.

Zhang, J., Chen, H., Zhu, Y., Shi, H., Zheng, Y., Xia, X., Teng, Y., Wang, F., Han, X., Li, J., and Xuan, Y.: A Novel Method for Estimating the Vertical Velocity of Air with a Descending Radiosonde System, Remote Sensing, 11, https://doi.org/10.3390/rs11131538, 2019.


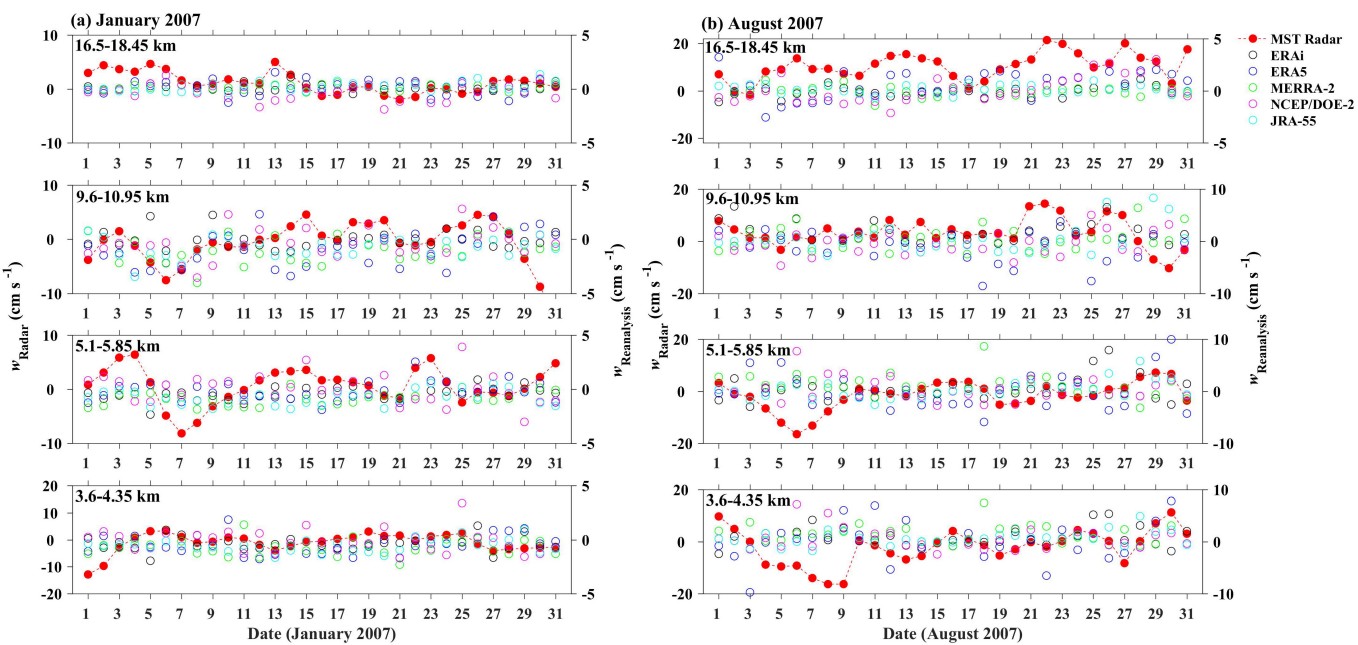

**Figure 2.** Intercomparison of layer averaged daily $w$ (12 UTC) measured from IMSTR with different reanalyses (ERAi, ERA5, MERRA-2, NCEP/DOE-2, and JRA-55) (12 UTC) over Gadanki for (a) January 2007, and (b) August 2007.

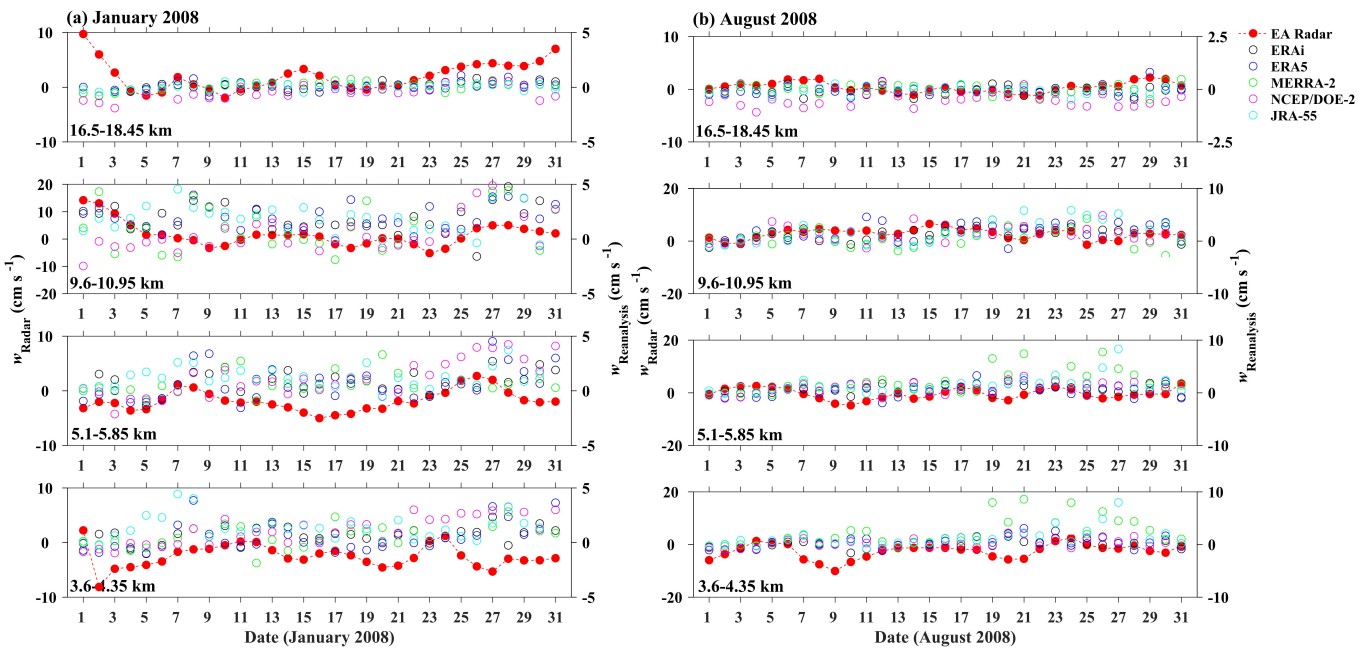

**Figure 3.** Same as Fig.2, but for EAR over Kototabang. Please note that *w* is diurnal mean (24 hrs mean) for both EAR and reanalyses for (a) January 2008, and (b) August 2008.

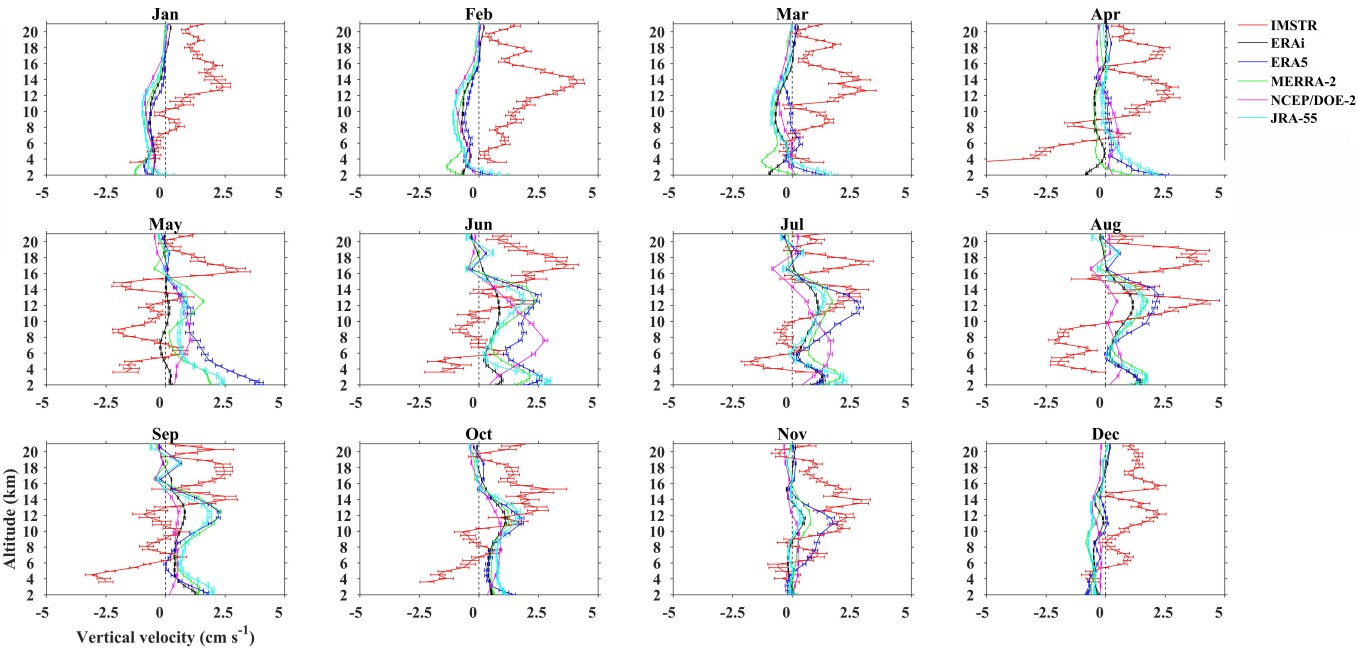

**Figure 4.** Climatological monthly mean altitude profile of *w* obtained from IMSTR and 5-reanalysis over Gadanki from 1995-2015. Horizontal lines indicate the standard error.

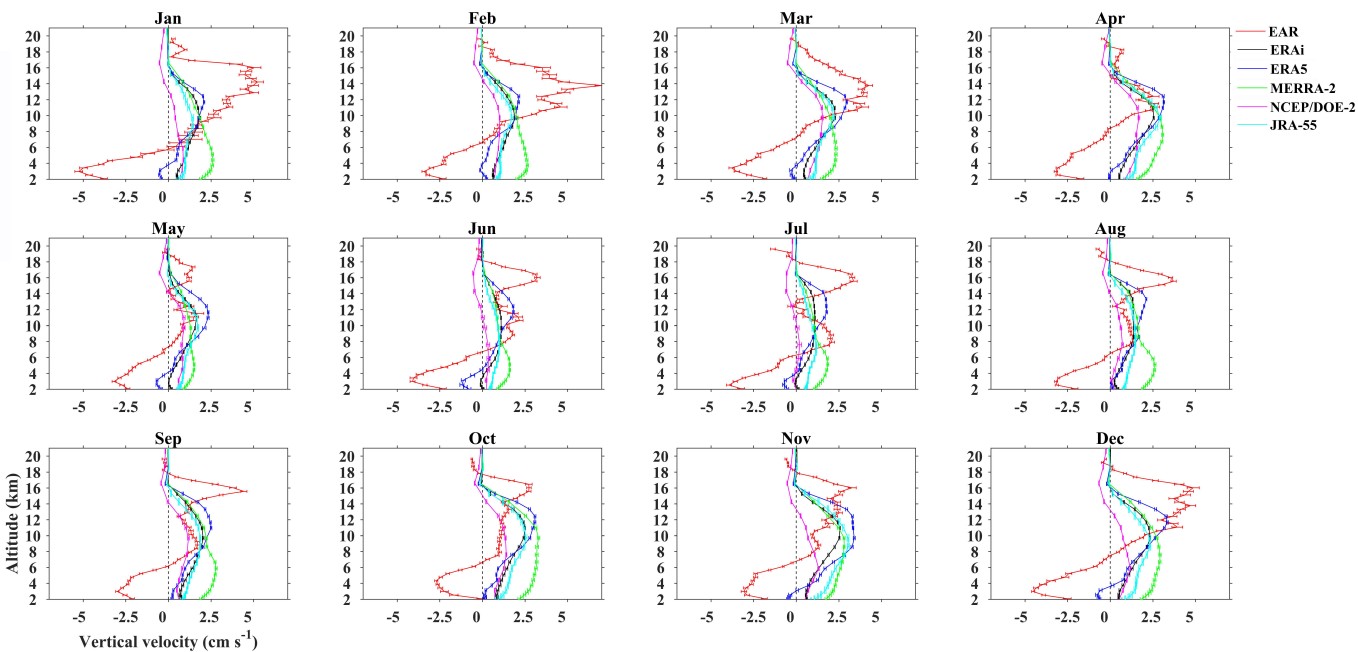

**Figure 5.** ame as Fig.4, but from EAR over Kototabang from 2001 to 2015.

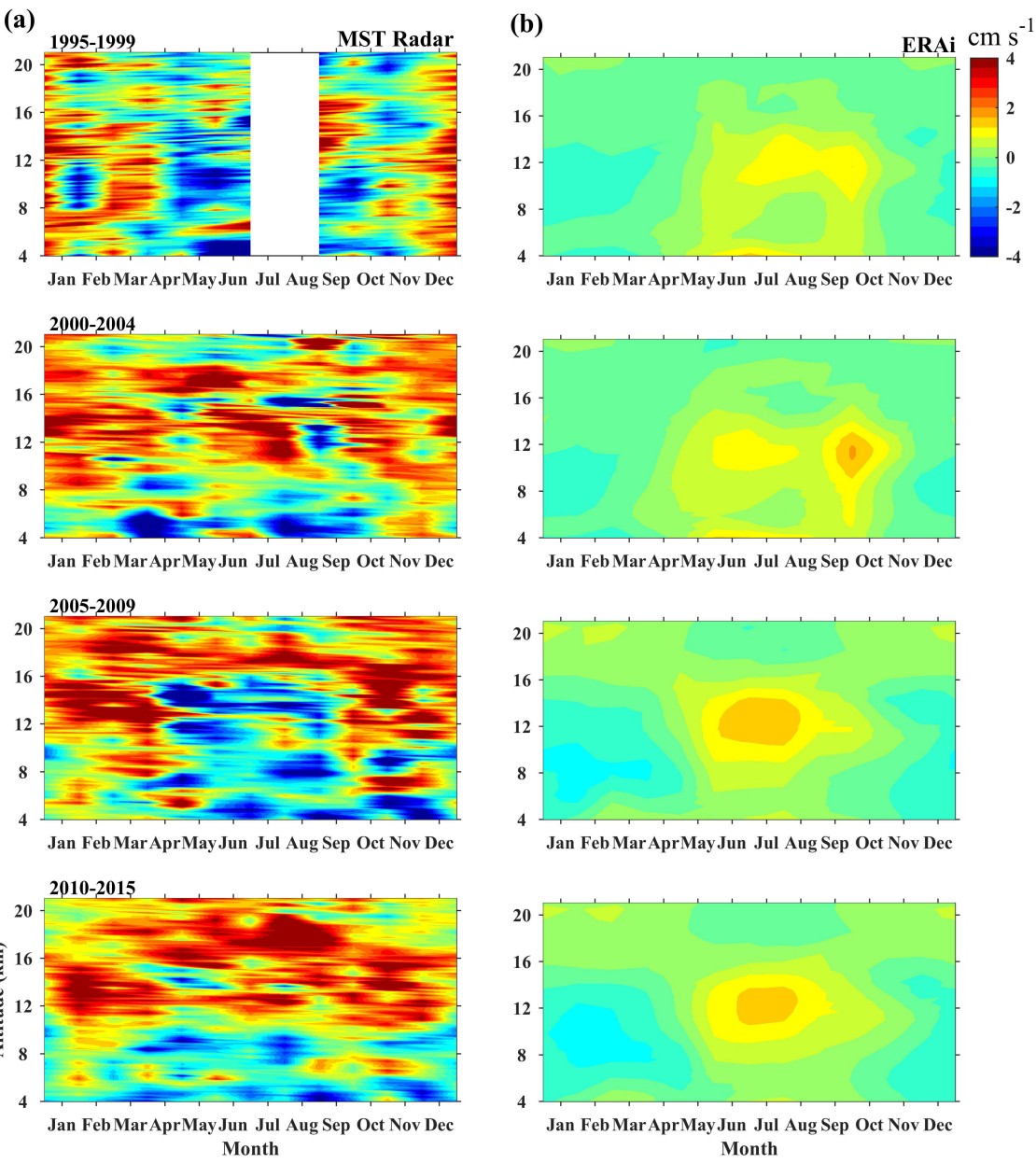

**Figure 6.** Monthly mean *w* obtained from (a) IMSTR and (b) ERAi for 5 years interval (from top to bottom) over Gadanki (12 UTC).

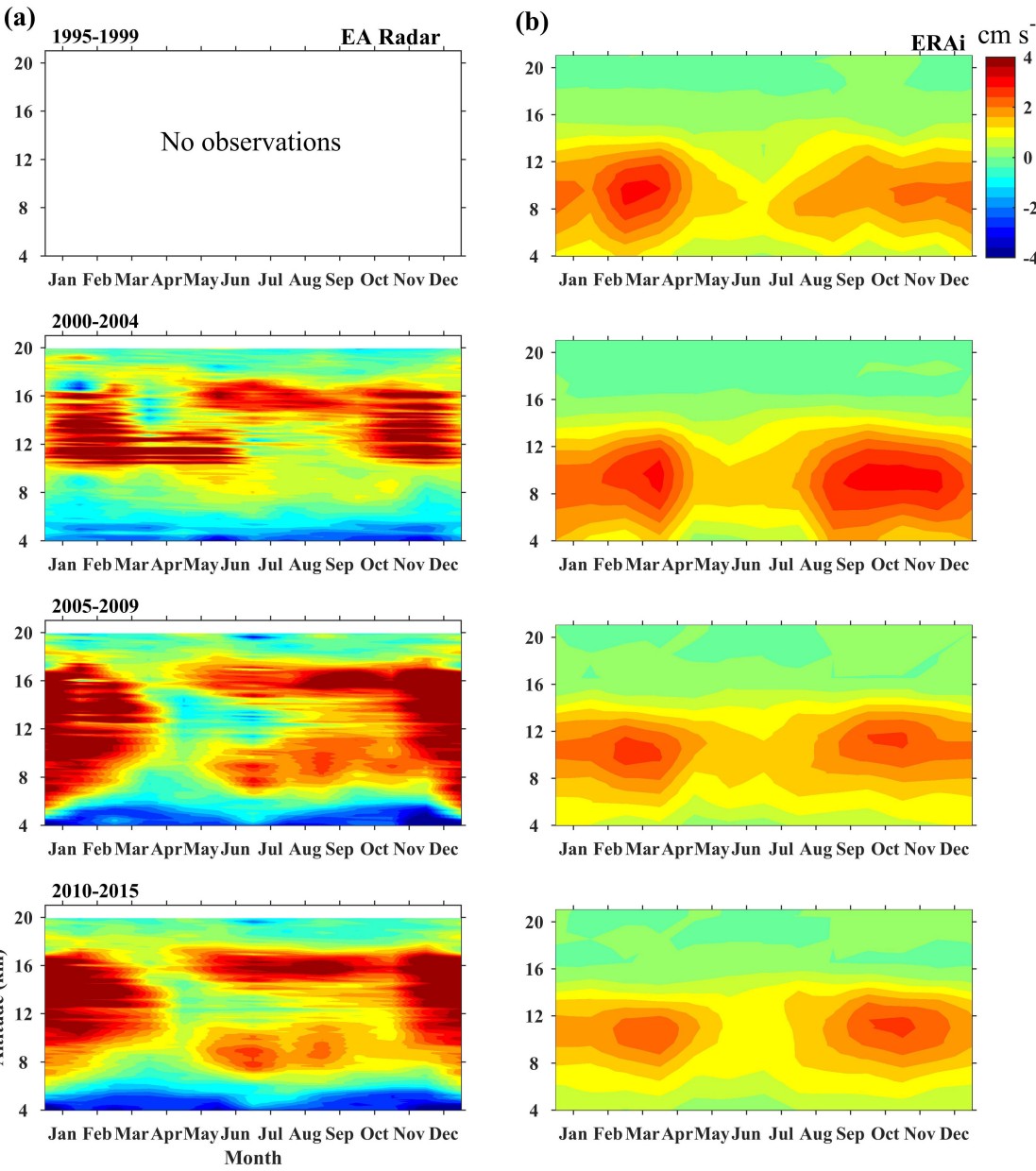

**Figure 7.** Same as Fig.6 but for diurnal mean from EAR over Kototabang.

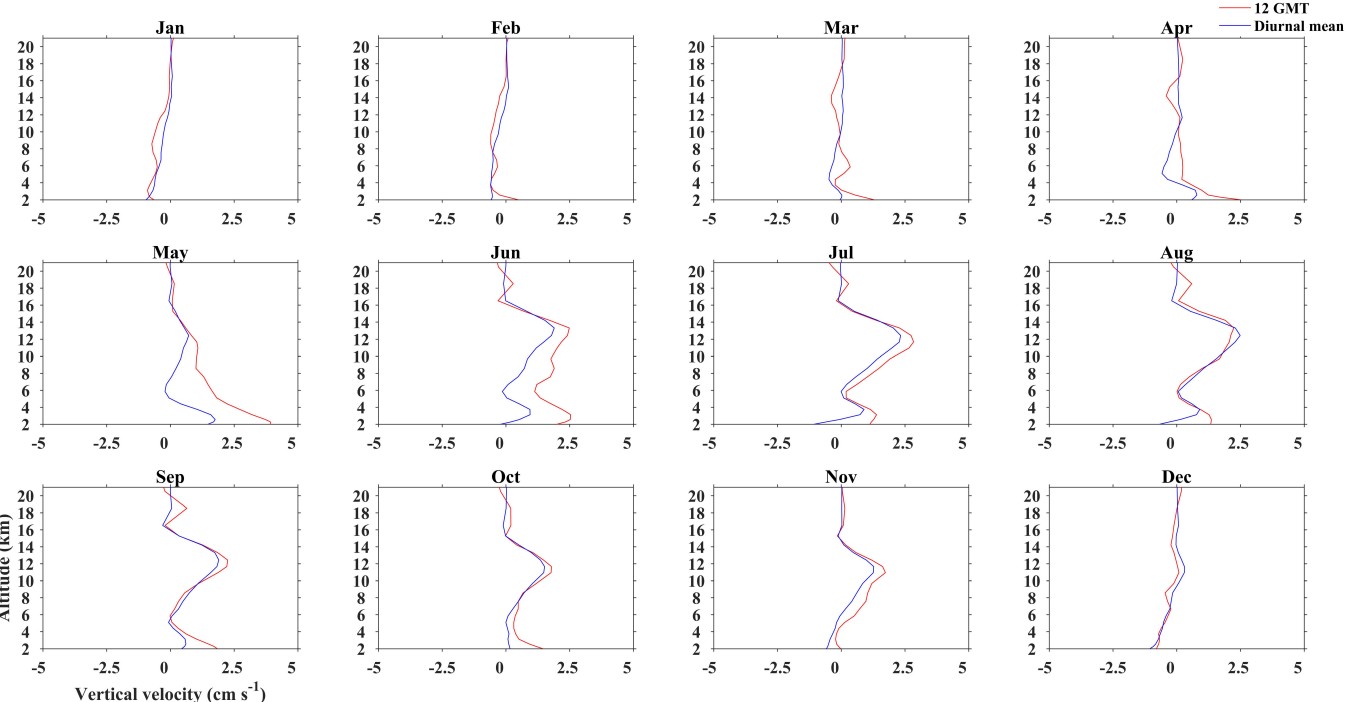

**Figure 8.** Height profile of *w* at 12 UTC and diurnal mean (with 1 hour resolution) over Gadanki extracted from ERA5 during 1995-2015 (highest available time resolution).

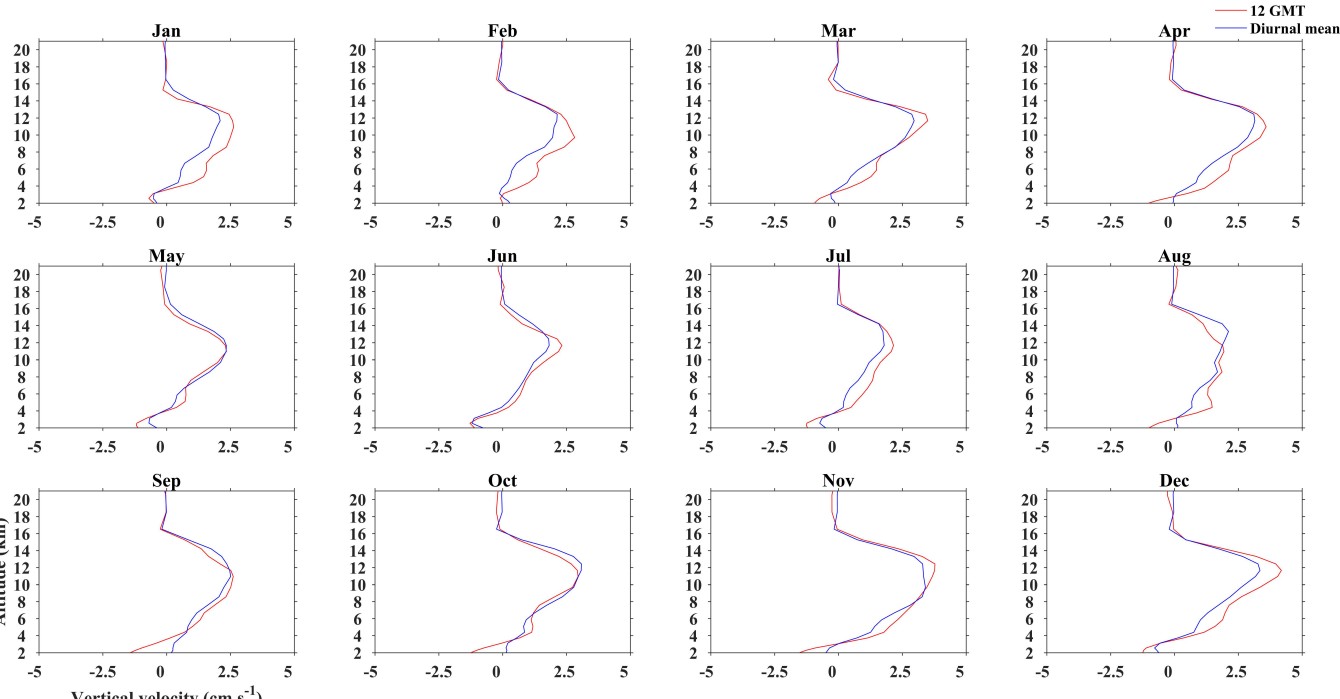

**Figure 9.** Same as Fig.8 but for Kototabang during 2001-2015.

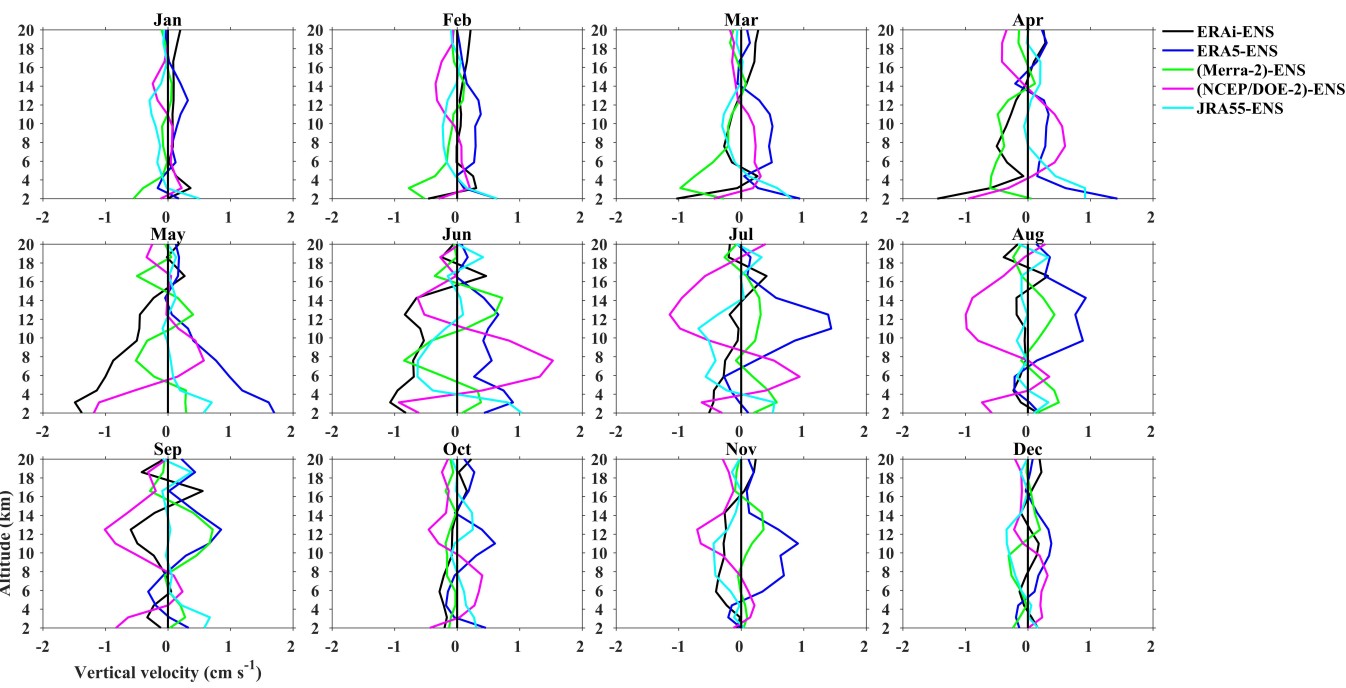

**Figure 10.** Comparison of relative differences in *w* between the reanalysis ensemble mean and each reanalysis for Gadanki from 1995 to 2015. Individual month differences are estimated and then averaged for each month.

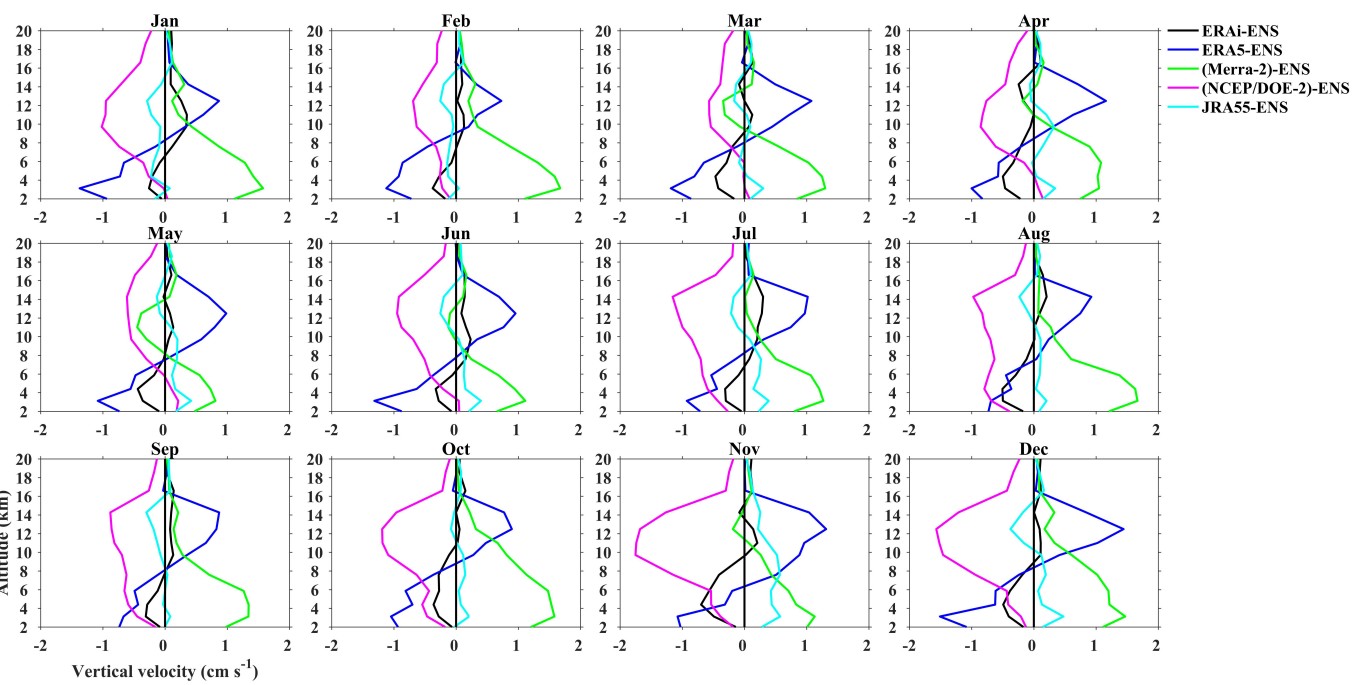

**Figure 11.** Same as Fig.10, but for Kototabang from 2001 to 2015.

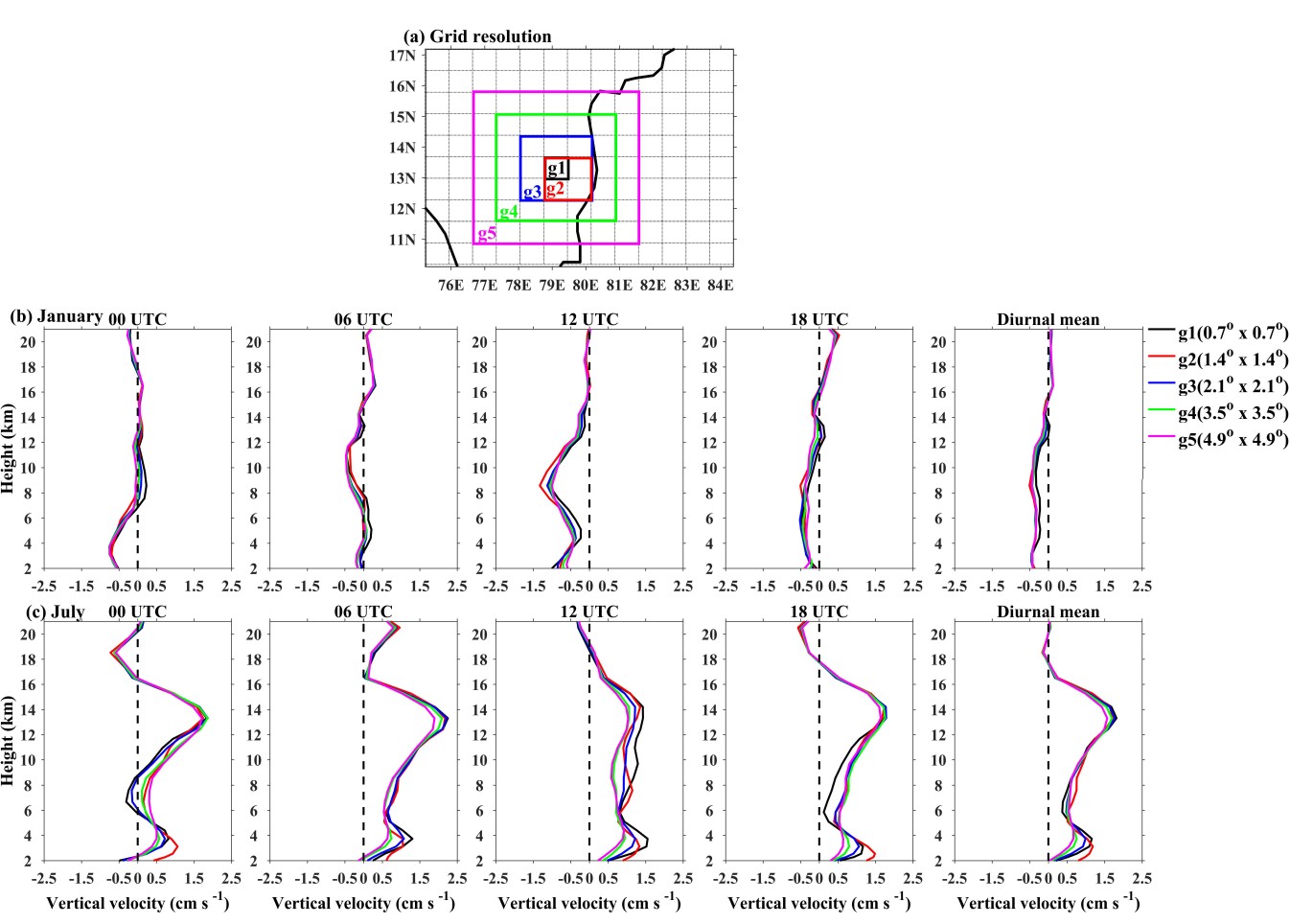

**Figure 12.** (a) Map for spatial averaging (grid resolution), and height profiles of *w* for different spatial averaging at 00, 06, 12, and 18 UTC respectively for ERAi reanalysis during 2007.

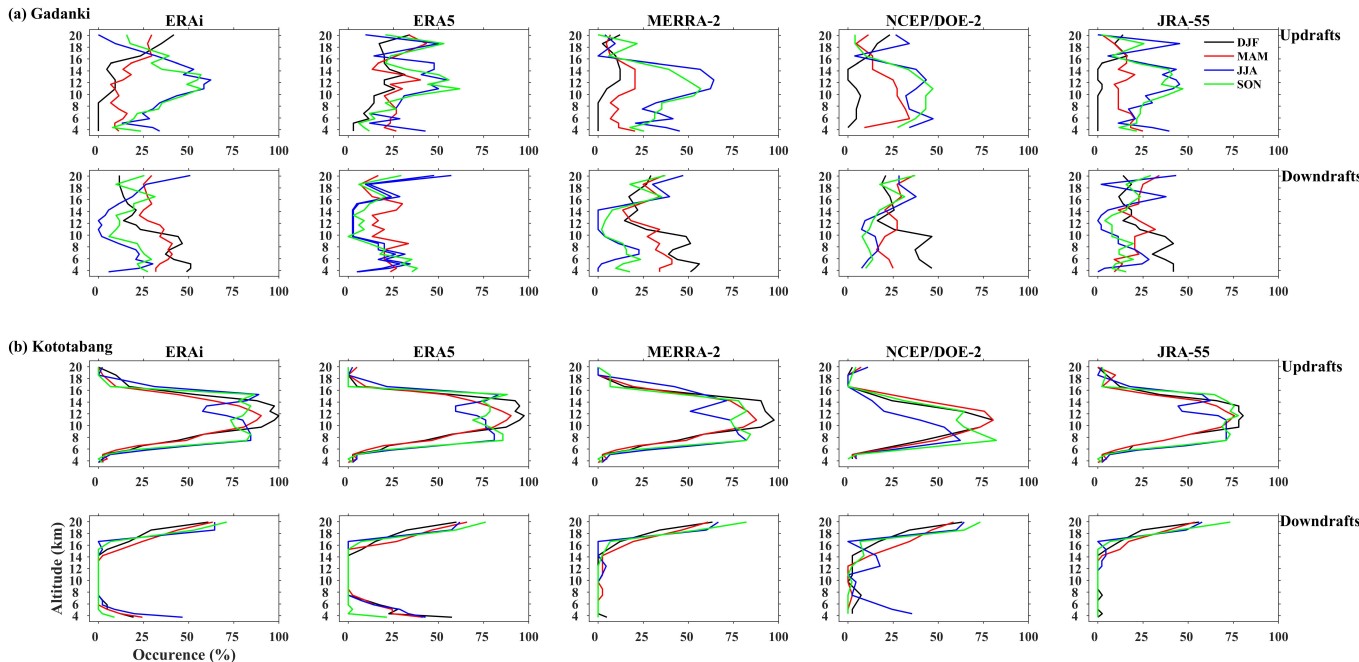

**Figure 13.** Comparison of directional tendency of *w* between the radars and various reanalysis data sets for (a) Gadanki (1995-2015) and (b) Kototabang (2001-2015). Updrafts are shown in top and third panels and downdrafts are shown in middle and bottom panels (for details see text).

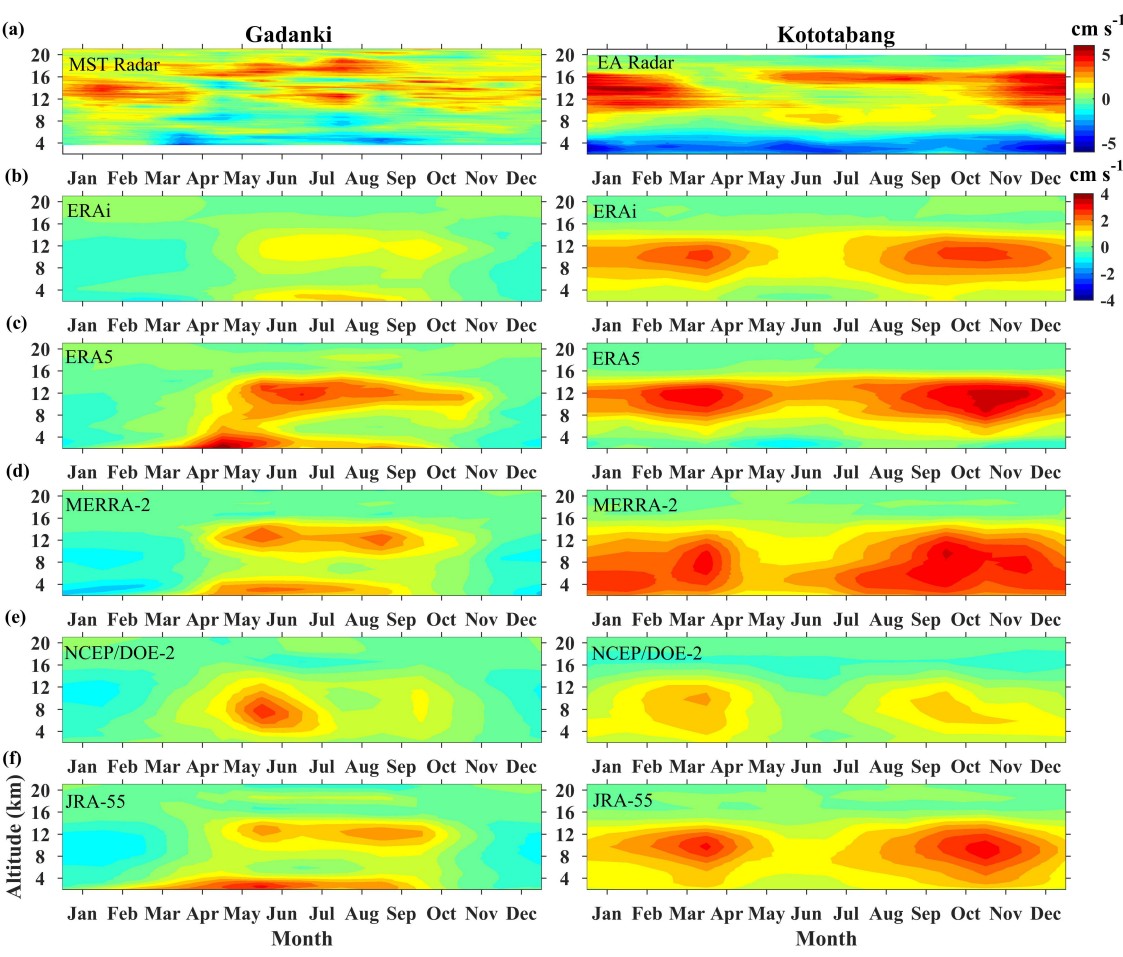

**Figure A1.** Monthly mean climatology of *w* obtained from (a) radars, (b) ERAi, (c) ERA5, (d) MERRA-2, (e) NCEP/DOE-2, and JRA-55 over Gadanki (left) (1995-2015) and Kototabang (right) (2001-2015). Gadanki data are at 12 UTC and Kototabang data are diurnal mean.

**Table 1.** The radars specifications and parameters used for the present measurements.

| Parameter | IMSTR | EAR |
|---|---|---|
| Frequency | 53 MHz | 47 MHz |
| Peak power | 2.5 MW | 100 kW |
| Maximum duty cycle | 2.5% | 5 % |
| Antenna | 1024, three-element Yagi antennas | 560, three-element Yagi antennas |
| Beam width | 3 degree | 3.4 degree |
| Mode of operation | | |
| Pulse width | 16 $\mu s$ with complimentary with 1 $\mu s$ baud | 0.5 to 256 $\mu s$ |
| Inter pulse period (IPP) | 1000 $\mu s$ | 200 and 400 $\mu s$ |
| Range Resolution | 150 m | 150 m |
| No. of FFT point (NFFT) | 256 | 256, 512 |
| No of coherent integration (NCI) | 64, 128, 256, and 512 | 16 and 32 |
| No. of Incoherent integration | 1 | 5 and 7 |
| No. of beams | 6 | 5 |
| | 10-degree off-zenith in East, West, North and South along with two orthogonal in zenith beams | 10-degree off-zenith in East, West, North and South along with one zenith beams |
| Velocity Resolution | 0.03 ms$^{-1}$ (CI=64, NFFT=256, IPP=1000 $\mu s$)<br>0.002 ms$^{-1}$ (CI=512, NFFT=256, IPP=1000 $\mu s$) | 0.002 ms$^{-1}$ (CI=32, NFFT=512, IPP=400 $\mu s$)<br>0.005 ms$^{-1}$(CI=16, NFFT=256, IPP=200 $\mu s$) |
| Data format | Spectrum | Spectrum |

**Table 2.** Schemes of different reanalyses data used in the present study.

| Description | ERA-Interim | ERA5 | MERRA2 | JRA55 | NCEP2 |
|---|---|---|---|---|---|
| Spatial Resolution | 0.75ºx0.75º | 0.28ºx0.28º | 0.5ºx0.65º | 1.25ºx1.25º | 2.5ºx2.5º |
| Longwave | Mlawer et al. (1997) | Morcrette (1991) | Chou et al. (2002) | Chou et al. (2002) | Mlawer et al. (1997) |
| Shortwave | Fouquart et al. (1990) | Iacono et al. (2008) | Chou and Suarez (1999) | Briegleb (1992) | Chou (1992) Chou and Lee (1996) |
| Convective Parametrization | Tiedtke (1989) | Convective mass flux scheme Tiedtke (1989) | Relaxed Arakawa-Schubert (RAS) Moorthi and Suarez (1992) | Prognostic Arakawa Schubert with DCAPE | Simplified Arakawa and Schubert (1974) |
| Cloud Scheme | Bechtold et al. (2004) | Bechtold et al. (2008) | Molod et al. (2015) | Kawai and In-oue (2006) | Campana and Cullather (1994) |
| Data Assimilation | 4D var | 4D var | 3D var with IAU | 4D var | 3D var |
| References | Dee et al. (2011) | Hersbach et al. (2020) | Gelaro et al. (2017) | Kobayashi et al. (2015) | Kanamitsu et al. (2002) |
| Vertical Levels | L60 | L137 | L72 | L40 | L28 |