# Peer review of "Assessment of vertical air motion among reanalyses and qualitative comparison with VHF radar measurements over the two tropical stations"

_Atmospheric Chemistry and Physics, 2020_

## Short Comment (SC1) · 28 Mar 2020

I fully agree that vertical air motion is crucial for atmospheric dynamics and underre-solved in most of present numerical weather prediction products. This is true not only for the tropics but at all latitudes. I appreciate this study as a relevant contribution using interesting radar data (with which I have only little experiences).

Please let me point out that vertical wind can be measured – at least in principle – also by other instruments including research aircraft. I recently published a study (Schu-

mann, 2019) on relationships between horizontal kinetic energy spectra of vertical wind and horizontal divergence of the divergent horizontal wind components, which can be separated from the rotational wind components by known Helmholtz decomposition methods. I compared with airborne wind measurements in the upper troposphere and lower stratosphere at mid-latitudes and compared to some available model data. In particular, I found a total of 80 % of w variance near the tropopause occurring at scales between about 0.5 and 80 km. Perhaps these findings, and some of the related literature cited in my paper are worth mentioning in your paper.

Reference:

Schumann, U., 2019: The horizontal spectrum of vertical velocities near the tropopause from global to gravity wave scales. J. Atmos. Sci., 76, 3847-3862, doi: 10.1175/JAS-D-19-0160.1.

---

## Author Comment (AC1) · 29 Mar 2020

We are very much thankful to the Dr. Schumann for reviewing our manuscript and providing positive suggestions. We have gone through the paper suggested by him. The paper is very interesting and we will include all the necessary points of the paper in the revised manuscript.

---

## Referee Comment (RC1) · Anonymous Referee #1 · 20 Apr 2020

This study compares and assesses vertical wind data from five global atmospheric reanalysis data sets. As independent measurements, it uses VHF radar measurements from two tropical stations, one at Gadanki, India, and the other at Kototabang, Indonesia. This is a very important trial and should be published in the end.

The main issue of the manuscript is, I think, in the apparent large discrepancy between the radar measurement results and the reanalysis data. On the other hand, I think we can say that the five reanalysis data sets show qualitatively similar seasonal and vertical distribution of w; there are differences, but considering that omega depends heavily

on forecast model of each reanalysis system, without direct observations assimilated, these differences (among the reanalyses) may be understandable.

I am afraid that we even need to start from suspecting any errors in the data processing and analysis for reanalysis omega. Did the data really come from the correct (intended) grid point? Is the conversion from omega to w really correct? I do believe that the authors did the correct procedure, but we need some more cross-check information to confirm that they really did, simply because the difference from the radar measurements is too large. One note is that the authors should remove data at some lowest levels where the reanalysis systems simply extrapolate data below the surface (the altitude of Gadanki and Kototabang is 360 m and 865 m, respectively).

Regarding the radar measurements, it would be very useful to discuss why w shows such seasonal and vertical distributions. What processes produce upper tropospheric ascending motion and lower tropospheric descending motion, the latter for Kototabang for all seasons and for Gadanki for April to October? Are there any publications that discuss this? There are several publications on the measurements of VHF wind profiler at Christmas Island (2N, 157W) (Gage et al., 1988, 1991, 1992). Their vertical wind profiles may look rather similar to what Kototabang measurements show. Thus, their discussion may be useful. Once we (rather theoretically) understand how the actual w distributions may look like, we can get more insight on why all these reanalyses show such distributions.

In summary, I think we need much more information (maybe direct or maybe indirect) that may be useful to understand why the radar measurements and reanalysis data show such different distributions.

Some specific comments (here, for this time, I only list major ones):

Page 2, lines 43-49, and page 3, line 68, and other places: The use of the terms "direct" and "indirect" may need to be reconsidered. "Direct" may be used for in situ measurements (e.g., radiosonde horizontal wind measurements), while "indirect" may

be used as "indirect estimation" e.g., of w from horizontal wind measurements/data to consider their divergence/convergence. For the case of radar measurements, we may use the term "remote sensing" measurements, because these may not be "direct" measurements (they are not in situ measurements) but at the same time these may not be "indirect" which implies indirect estimation from other variables in the context of this manuscript.

Pages 5-6, Section 2.1: The full location information on the two radar sites need to be written in this section. The information on altitude, country/island, and the institutes that operate these radars is missing. Also, please explain the topography around each of these radar sites rather extensively. Gadanki is located within high land of a continent (with a horizontal distance of ** km from the oceans), while Kototabang is located within a narrow mountain range of an island (of a scale of ** km in northwest-southeast and ** km in northeast-southwest), etc. The topographic information may be very important to judge the representativeness of reanalysis data at a particular grid point (and at the same time, the representativeness of each of these radar measurements). The direct time information in UTC should be provided, because reanalysis data are in UTC. It would be useful to show the profiles of data number, i.e., of the original ones, of the quality controlled ones, of the finally used ones (after discarding data points >1 m/s and <-1 m/s), etc. The information on the quality control procedure is also needed. (The authors listed possible issues in the radar measurements, but they did not explain what they actually did to avoid such issues.)

Page 7, line 157: Which MERRA-2 data product was used, ASM or ANA?

Pages 9-, Section 3: Please see my comments above. Also, investigation of a case or two (e.g., for a week or for a month) might be useful to understand what is going on for both radar measurements and reanalysis data.

Figure 8: Results from Gadanki and Kototabang should be shown in separate panels/figures. ERA-Interim is used as a reference. But, considering that ERA-Interim

is an outlier for some cases, the Reanalysis Ensemble Mean (REM) may be a better reference. Finally, a reference should be "subtracted", e.g., ERA5 minus REF, not REF minus ERA5.

References:

Gage et al., A Comparison of Winds Observed at Christmas Island using a Wind-Profiling Doppler Radar with NMC and ECMWF Analyses, Bulletin American Meteorological Society, Vol. 69, No. 9, 1041-1046, 1988.

Gage et al., Long-Term Mean Vertical Motion over the Tropical Pacific: Wind-Profiling Doppler Radar Measurements, Science, Vol. 254, No. 5039, 1771-1773, 1991.

Gage et al., DIURNAL VARIATION IN VERTICAL MOTION OVER THE CENTRAL EQUATORIAL PACIFIC FROM VHF WIND-PROFILING DOPPLER RADAR OBSERVATIONS AT CHRISTMAS ISLAND (2 N , 157W ), Geophysical Research Letters, Vol. 19, No. 18, 1827-1830, 1992.

---

## Referee Comment (RC2) · Anonymous Referee #2 · 1 Jun 2020

I like the premise of this paper very much: observational evaluation of reanalysis vertical velocities is much needed, perhaps especially within the broader Asian monsoon region. This is an important topic and I would like to see the paper published in the end. However, the presentation contains some significant gaps and unclear reasoning that should be addressed.

My main reservation is that the comparison as presented is almost entirely descriptive, with little analysis of the causes of biases or how they might inform further improvement of the reanalysis products (see also comment D below). It would be very helpful – if not

essential – to include more interpretation of both the differences among the reanalyses and the biases relative to observations. For example, the introduction indicates that section 4 includes both a discussion and a summary of the results, but section 4 itself includes little discussion, only summary. Some questions to consider:

- Can we understand anything about what the reanalyses are doing wrong (or right) from the observational validation or reanalysis-only intercomparison results?

- Do the differences indicate major problems or can they be largely understood in terms of spatiotemporal sampling? For example, the narrow column observed by the radar relative to the reanalysis grid scale – do comparisons of reanalyses with grid scales spanning a factor ten offer any context here? In a related question, does resolution of the nearby topography come into play in any obvious ways?

- Are there any clues as to how different types of data assimilation (3D-Var vs IAU vs 4D-Var) influence biases in vertical velocities? What about details of the model physics, such as convective or boundary layer scheme?

- How robust are the results between the two sites? Does this have any implications for which conclusions, if any, can be generalized?

I appreciate the authors' attention to earlier editorial comments. I have included an annotated manuscript with some additional (optional) suggestions, which also references the specific comments below.

**General comments and figures**

A (Sect. 2) The descriptions of the reanalyses in section 2 should include indications of how vertical velocities are computed in each reanalysis, whether these estimates are impacted by data assimilation (i.e. forecast versus analysis versus IAU in the case of MERRA-2), and whether they represent time-average or instantaneous estimates. It would also be helpful to give some basic information about the model vertical coordinates, as I think vertical pressure velocities are usually estimated in these coordinates and then interpolated to the pressure grid. To the extent that these procedures are the same across multiple reanalyses the description could be consolidated, emphasizing differences in each subsection. Any other aspects of the model / data assimilation that might aid interpretation of the differences should also be included here.

B (Method) Some of the specifics of the analysis are difficult to follow. For example, in l.287 it is indicated that directional tendencies are reported for a given height for every month when either the radar or reanalysis data exceeded $\pm 1\,\mathrm{cm\,s^{-1}}$. This screening is based on the monthly mean, correct? Then, a few lines later, it is indicated that the directional tendency is calculated only for absolute magnitudes greater than $0.1\,\mathrm{cm\,s^{-1}}$. Is this now referring to the daily data? Are the results sensitive to these thresholds, especially in terms of the reanalysis evaluation?

C (Sect. 3, last two paragraphs) The directional tendency results for ERA5 are difficult to understand. The lack of strong updrafts or downdrafts at Gadanki seems to contradict the results shown in Fig. 3 (where ERA5 seems to have relatively strong vertical velocities and it is ERA-Interim more than ERA5 that looks like the outlier) and Fig. 6 (which shows a pretty robust seasonal cycle with many monthly averages well above the threshold). The results for Kototabang are likewise preplexing in the context of Fig. 3 and Fig. 7. How do you reconcile the directional tendency results in Fig. 9 with the profiles shown in these earlier figures?

D (Sect. 4, l.339-340) I like this thought, but more needs to be done to really provide a useful platform for improving the reanalyses. For one, it is not clear whether it is the 'methodology for calculating $w$' that lies behind the identified biases, as opposed to, e.g., different diurnal or day-to-day variations in convective occurrence (see comment 20), interactions between subgrid physical parameterizations and the large-scale flow, crude representation of local topography or land surface conditions, even just differences in spatial sampling area. This last is even suggested as the main reason for the differences in l.195-196, and it is not clear how reanalyses can address this beyond moving to finer and finer horizontal resolutions, which they are already doing. I recognize that it would be a monumental task to try to diagnose all of these and do not ask for a exhaustive investigation, but some investigation and discussion would be warranted. For example, operating on the hypothesis that the differences are mainly related to averaging, you could try imposing 'subgrid fluctuations' on the reanalysis products. What scale of fluctuation would need to be imposed to bring the reanalysis products in line with the observations? Is there any relationship between this value and the re-analysis grid size? Do the results make physical sense, or do they suggest that other factors must contribute?

E (Data availability) The data citations are incomplete. Both NASA (for MERRA-2) and the NCAR RDA (for all other reanalyses) have assigned doi numbers to the datasets used in this paper. These doi values should be used in data citations (input data doi at https://citation.crosscite.org/ for citation details) to help the data providers track the impact of their investment. Dates of access should also be included (since reanalyses occasionally undergo reprocessing to fix errors), along with the specific variables and resolutions used (to ensure reproducibility).

F (Figure 3) Is it possible that the vertical profiles for MERRA-2 at Kototabang have been inverted somehow? The differences between these and ERA5/ERA-Interim are pretty striking, perhaps especially the downward shift of the maximum during May–June in MERRA-2 relative to the upward shift of the maximum in ERA5 and ERA-Interim (not to mention the radar profiles). I know that the orientation of the vertical coordinate may differ (top-to-bottom, bottom-to-top) in data files released by these different reanalysis producers. Please double-check this for MERRA-2, and perhaps also for NCEP-2.

G (Figure 8) Could Fig. 8 be made more effective building off of the presentation in Fig. 3, using difference plots relative to a particular reanalysis-based benchmark? I agree that the current presentation could help in terms of explicitly comparing quantitative biases across different reanalyses, but it is very difficult to pick out details of the individual profiles in the current figure. Another option might be to consolidate some months with similar profiles (it looks like the canonical seasons might work, but warm/cold/transition seasons could also work) and then split the Gadanki and Kototabang profiles (results for the two sites do not seem to share that much in common in the vertical distributions).

H (Figure 9) The caption says this figure shows a comparison between the radars and various reanalysis products. Where are the directional tendencies based on the radar data? It is difficult to evaluate the reanalyses without this information. Please excuse me if I am missing something really basic about the presentation.

**Specific comments on text**

1. (l.33) Perhaps also mention the role of subsidence and adiabatic warming in the formation of stable inversion layers?

2. (l.36,38) I think the use of 'control' here rather overstates the case, especially in l.36. It would be enough to delete 'controlling'; the second use in l.38 is ok on its own.

3. (l.49) Please clarify what is meant by 'global estimates' and its relationship to 'direct measurements' versus 'indirect estimates'; there is also an extra comma here.

4. (l.57) Maybe mention here that this source of uncertainty is particularly important for reanalyses, where assimilation increments in horizontal winds may be comparable to this. It might also be helpful to rephrase the sentence to emphasize this assimilation adjustment rather than 'error'.

5. (l.65) The connection of the above discussion to reanalysis estimates of vertical velocity should be made more explicit (i.e., why do these concerns apply to reanalysis products specifically)

6. (l.66) Suggest rewriting this sentence: 'reanalyses involve many approximations and assimilation-related adjustments, and are not error-free'

7. (l.84) Technically a reanalysis vertical profile is also a column over a single location, just one with a broader footprint. Here it would be helpful to specify how the area of the column differs between the radar, the finest-grid reanalysis (0.25°, right?) and the coarsest-grid reanalysis (2.5°).

8. (l.86) Phrasing needs care here: a number of studies have evaluated vertical motion across reanalyses (in the context of trajectories, wave activity, large-scale motion, etc.), so the primary novelty of this work is the evaluation against radar observations.

9. (l.88) This point is a little repetitive.

10. (l.119) Rephrase: 'Quality control metadata for the EAR measurements are available online'

11. (l.140) How long is 'long'?

12. (l.146) How is this 21 km upper limit identified in the reanalysis profiles, and approximately what pressure level does this correspond to? More generally, many of the results are presented in altitude coordinates. Are heights computed from the pressure levels assuming a constant temperature, from geopotential outputs from each reanalysis, another approach? Are data linearly interpolated to a common height grid? Could this have any influence on the comparisons in this paper (e.g., the reanalysis-derived updraft maxima being located lower than those observed by the radar)?

13. (l.152) Citation year for Hoffmann et al. should be 2019

14. (l.153) The ERA5 paper is now in early online release (https://rmets.onlinelibrary.wiley.com/doi/10.1002/qj.3803)

15. (l.165) NCEP-2 (as denoted here) was undertaken as a cooperation between NCEP and the Department of Energy (DOE); care should be taken in this text to acknowledge this and distinguish it from the NCEP-NCAR Reanalysis 1

16. (l.171) The paragraph mentions only the original model resolution, but from this description the data are taken from the 2.5° data grid

17. (l.178) This information is not provided for the other reanalyses nor is it clear how it relates to the estimation of vertical velocity. I suggest that the authors try to provide a consistent and concise set of information for each reanalysis in this section, focusing on the points relevant to the data used in this paper.

18. (l.186) for dry air

19. (l.188) daily mean is evaluated for 00–24 UTC or shifted to match local solar time? I guess it shouldn't matter much as long as this is consistent between the radar and reanalysis

20. (l.195) It looks like convective days in the reanalysis products are defined based on the screening from the radar data. How consistently do the reanalyses identify convective versus non-convective days based on measurements at the radar site? Wouldn't this screening also be sensitive to the differences in grid size? Some sensitivity testing would be helpful here, perhaps using precipitation thresholds as well as $w$.

21. (l.198) The meaning of 'global' here is not clear — should this be 'qualitative' to set it against 'quantitative' differences that might result from differences in sampling area (but see also comment 20)?

22. (l.206) ERA5 should be written without the hyphen (as it is now in much of the manuscript; thank you)

23. (l.216) It is not clear from the text whether these 'significant variations' are in the seasonal cycle, in the diurnal cycle, or both (since the previous sentence discusses seasonal variations in the diurnal cycle). Some additional clarification would be helpful

24. (l.220) The temporal treatment is another source of potential differences in vertical velocities between the observational data (time averages over at least one hour) and reanalyses (usually instantaneous outputs, I think – please check – and usually only four times per day). Naively, it seems like this might offset some of the smoothing effect of the spatial sampling difference, but it should be mentioned and discussed either way.

25. (l.231) Could this comparison be sensitive to the definition of 'convective' days? This may be especially relevant for Gadanki where the reanalyses may even have different diurnal cycles of convection. For example, Bechtold et al. (2014) reported that changes to the forecast model between ERA-Interim and ERA5 resulted in substantially improved representation of the diurnal cycle of convection over land.

26. (l.249) Does this result generalize to the observational validation — i.e. do the

EAR results for Kototabang also support this conclusion? This information could be added to Fig. 7 and related discussion.

27. (l.272) Some additional care might be needed in the presentation here, to distinguish when the values being quoted are absolute magnitudes of $w$ (as here) versus when they are biases relative to a particular benchmark (as earlier in the paragraph).

28. (l.278) This presentation ('underestimates'/'overestimates') is a little strange, since it seems to imply an evaluation of ERA-Interim against multiple benchmarks as opposed to an intercomparison among (presumably) equally uncertain reanalysis-based products. A more specific and less judgemental phrasing might help, something like: 'XX shows smaller positive values than YY and larger positive values than ZZ', or 'XX and YY both show downdrafts, but with larger amplitudes in XX', or maybe using stronger/weaker updrafts/downdrafts if you prefer.

29. (l.292) Should delete the space between 10 and % (also elsewhere in this and following paragraphs).

30. (l.293) Is this ratio indicating that 10% of all 12 UTC values exceed $0.1\,\mathrm{cm\,s^{-1}}$ or that 10% of all positive values have magnitudes grater than $0.1\,\mathrm{cm\,s^{-1}}$? If the first, it is enough to remove 'of updrafts'; if the second, some additional text to clarify is necessary.

31. (l.301) Suggest to be more explicit: 'The fraction of downdrafts decreases above ...'

32. (l.311) Is this 'reaches a maximum' specifically referring to the increase above 17 km or to both the increase below 6 km and above 17 km? If the latter, 'a maximum' should be changed to 'maxima'.

33. (l.330) Here, do you mean 'the location of the peak $w$ differs between radar and reanalysis data, as it also does over Gadanki' or just that 'the vertical location of the $w$ maximum over Kototabang is different from that over Gadanki'?

34. (l.331) Again, suggest revising this presentation style to be clearer and more objective (see also comment 28).

35. (l.338) These examples are a little strange. It is true that the behavior of physical parameterizations in the reanalyses (used to generate diabatic heating and convective mass fluxes) may be impacted by large-scale convergence / divergence (and hence by the same factors used to compute $w$), though the feedbacks between $w$ and model physics are two-way, complex, and pretty different from how they are implied to be here. Or perhaps the authors refer to diagnosed diabatic heating (e.g., Yanai et al., 1973) and vertical motion along Lagrangian trajectory pathways? Note that the latter should be distinguished from 'convection', which is included in some transport models but I think usually based on vertical stability rather than $w$.

36. (Fig. 5 caption) Should this reference be to Fig. 4?

**Supplement:**

[revised manuscript text omitted]

Very-high frequency (VHF) and ultra-high frequency (UHF) vertical pointing radars are the most powerful tools for determining  vertical air motion (velocity) directly with high temporal and vertical resolution. However, the magnitude may still not be directly comparable between reanalyses products and observations as the reanalyses provide the intensity of vertical air motion over wide areas ($> 25$ $km^2$), whereas the direct radar measurements provide information for the column over a single location. Thus, the best way to assess reanalysis estimates of $w$ is to compare  directional tendencies . To the author's knowledge, no studies yet exist concerning  the assessment of $w$ products derived from

different reanalyses and evaluation of these products against radar measurements. The present study, which is therefore first of its kind, focuses on assessment of $w$ among various reanalyses using VHF radar measurements from two tropical stations where convective activity is frequent:

[revised manuscript text omitted]

*introduce how vertical velocities are computed (i.e. continuity on η levels) and whether they are impacted by data assimilation*

**2.2** **ERA-Interim**

We use 6-hourly vertical velocities from the European Centre for Medium-Range Weather Forecasts (ECMWF) Interim reanalysis (ERAi) from 1995 to 2015 (Dee *et al.*, 2011). The nearest grid points are taken for Gadanki (13.68°N, 79.45°E) and Kototabang (0.35°S, 100.54°E). Although 37 pressure levels up to 1 hPa resolution are available, we have restricted the dataset to 21 km, as that is the maximum radar range.

**2.3** **ERA5**

When compared to ERAi, the fifth ECMWF reanalysis (ERA5) provides much higher spatial (30 km) and temporal resolution (hourly) from the surface up to 80 km (137 levels). ERA5 also features much improved representation especially over the tropical regions of the troposphere and better global balance of precipitation and evaporation. Many new data types not assimilated in ERAi are ingested in ERA5 (Hoffmann *et al.*, 2018). The details are available in Copernicus climate change service report (Hersbach and Dee 2016 and https://cds.climate.copernicus.eu/cdsapp#!/home). The nearest grid points are again taken for Gadanki (13.63°N, 79.31°E) and Kototabang (0.14°S, 100.40°E), and the data period is 2002-2015.

**2.4** *MERRA-2*

The Modern Era Retrospective analysis for Research and Applications, version 2 (MERRA-2) is the latest reanalysis of the modern satellite era produced by the National Aeronautics and Space Administration's (NASA) Global Modelling and Assimilation Office (GMAO). MERRA-2 data are provided on 42 pressure levels from the surface to 0.01 hPa with a temporal resolution of 3 h and horizontal resolution of 0.5° in latitude by 0.625° in longitude◦. Details have been provided by *Gelaro et al.* (2017). The nearest grid points are used for Gadanki (13.5°N, 79.37°E) and Kototabang (0.14°S, 100.00°E), with coverage from 1995 to 2015.

**2.5  *NCEP-2**

[Figure]

The National Centers for Environmental Prediction – National Center for Atmospheric

Research (NCEP-NCAR) reanalysis is based on the NCEP operational model with a horizontal resolution of 209 km and 28 vertical levels. Its temporal coverage is four times per day. NCEP-2

products are improved relative to NCEP-1, having fixed errors and updated parameterizations of physical processes, as evaluated by *Kanamitsu et al.* (2002). The data for the present study covers 1995 to 2015 and is extracted at the nearest grid points to Gadanki (12.5$^{o}$N, 77.5$^{o}$E) and

Kototabang (0, 100.00$^{o}$E)

**2.6   *JRA-55**

The Japanese 55-year reanalysis  (JRA-55) is an updated version of the earlier JRA-25

with new data assimilation and prediction systems (*Kobayashi et al*., 2015). New radiation schemes, higher spatial resolution and 4D-var data assimilation with variational bias correction for satellite radiances have been used to generate the JRA-55 products.  This reanalysis includes variation in greenhouse gas concentrations with time, as well as the new representations of land surface parameters, aerosols, ozone and SSTs. The horizontal resolution of the forecast model is

~60 km for JRA-55. The nearest grid points are taken for Gadanki (13.75$^{o}$N, 78.75$^{o}$E) and

Kototabang (0, 100$^{o}$E) and the data period is 1995-2015.

For all the reanalyses data, $w$ (in cm s$^{-1}$) is estimated using the formula:

$$w = -\frac{1}{g}\omega\frac{RT}{p}$$     (2)

where $\omega$  is the vertical velocity in pressure coordinates (in Pa s$^{-1}$), T is the absolute temperature (K), p is the atmospheric pressure (hPa) and R (=287 J kg$^{-1}$ K$^{-1}$) is the gas constant. To compare

measured vertical wind with the reanalysis products, we take the reanalysis data corresponding to

12 GMT for Gadanki and the daily mean for Kototabang.

[Figure]

**3    Results and Discussion**

Figure 1 shows the climatological monthly mean altitude profile of *w* obtained from the

IMSTR (observations) and the ERAi, ERA5, MERRA-2, NCEP-2 and JRA-55 reanalyses data sets over Gadanki. Although the magnitudes are of the same order between the observations and reanalyses,significant differences are identified in the figures. It is to be noted that convective days are discarded in the radar analysis (observations) as mentioned in the previous section, and those days are also eliminated from all the reanalysis data sets. These differences may be attributed to the spatial averaging implicit in the reanalyses products, whereas the radar measurements are for a single point. Thus in the present study, we only discuss the tendency of *w*

as it is used to represent the global variation of *w,* rather than its magnitude. The IMSTR

observations show updrafts between 8 and 20 km, with the largest values in the tropical tropopause layer (TTL, 12-16 km), from December to April. These features are not reproduced by any of the reanalyses, which all show downdrafts from December to April between 1 km and the tropopause level (mean tropopause is ~ 16.5 km). Comparatively, downdrafts are observed in the IMSTR below 6 km in April, which may be attributed to pre-monsoon (March-May)

precipitation and evaporation (*Uma and Rao*, 2009a). Vertical velocity in ERAi differs in both magnitude and direction from other reanalyses, especially in the lower troposphere from March to June. Meanwhile, the magnitude of vertical velocity in ERA-5 is a little larger than that in the other reanalyses from May to June. Updrafts are observed in the TTL by the IMSTR during

June, when all reanalyses show similar features but located below the TTL. During July and

August both the radar observations and the reanalyses show updrafts in the vicinity of the TTL.

[Figure]

Updrafts are observed in the TTL from September to November but the peak in the updrafts is shifted lower than that observed by the IMSTR. Below 8 km, IMSTR shows downdrafts from

April to October. It is notable that the reanalyses only produce downdrafts below 2 km and are unable to reproduce the downdrafts above 2 km. Earlier studies using the IMSTR showed similar seasonal characteristics for $w$ (*Rao et al., 2008*).

  *Uma and Rao* (2009b) have reported the diurnal variation of $w$ in different seasons, Their observations have only 1-2 diurnal cycles per month over Gadanki. They found significant variations as large as 6 cm/s over Gadanki using IMSTR. The present observations are limited to

16:30 to 17:30 IST, with all reanalysis data over Gadanki taken at 12 GMT (17:30 IST). Thus, time-averaged climatological mean biases can be neglected. We have also analyzed $w$ from the

EAR (Kototabang) where the observations are available for the full diurnal cycle (measurements of hourly averages for 24 hrs of observations). All reanalysis data over Kototabang are averaged for the full diurnal cycle. Figure 2 shows the monthly mean climatology of daily mean of $w$

observed by the EAR and five reanalyses over Kototabang. All the reanalyses agree well with each other over Kototabang. Radar measurements of $w$ at this location consistently show updrafts in TTL region and downdrafts below 6 km (e.g. *Rao et al., 2008*). The updrafts in the TTL are well reproduced by all the reanalyses although the peak magnitude of $w$ and its vertical location remain lower than observed.. However none of the reanalyses reproduces the downdrafts. A

distinct bimodal distribution in $w$ from May to September (two peaks between 8-10 km and 14-

17 km) with a local minimum between 12 and 13 km is observed in the EAR measurements. The magnitudes of both updrafts and downdrafts are larger than those observed over Gadanki. JRA-

55 produces the largest $w$ among the reanalyses. The monthly means show significant differences in the direction of $w$ between the observations and the reanalyses below 6 km.

[Figure]

To establish the robustness of the results we have used different averaging procedures to assess the consistency of the variability in $w$ at monthly scales. Monthly mean climatological profiles of $w$ from radar observations and various reanalyses over Gadanki and Kototabang  are shown in

Figure 3. Downdrafts in the troposphere are not captured by any of the reanalyses over either location. By contrast, updrafts in the TTL are generally reproduced in the monthly mean, though they are often overestimated by the reanalyses. ERAi underestimates the magnitudes of both
       _their magnitudes_

updrafts and downdrafts over Gadanki,  while NCEP-2 underestimates the magnitude of updrafts over Kototabang.

Monthly means calculated over five-year periods from both radar and ERAi are shown in
       _the data_

Figure 4 for Gadanki and Figure 5 for Kototabang. The reanalysis shows a similar behavior to the overall climatology in each five-year average. The overall patterns of updrafts and downdrafts in the radar measurements of vertical velocity are also similar, indicating a consistent performance of the radar over the full 20 year analysis period.

To further elucidate potential biases in the results due to averaging, we have taken ERA-5 _ERA5_

at 12 GMT and compared it to the daily mean (obtained by averaging $w$ at different times of the day) to show that the sampling restrictions at Gadanki do not bias the results obtained. Figures 6

and 7 show the mean $w$ obtained at 12 GMT and also the mean obtained by averaging hourly analyses for each day for Gadanki and Kototabang, respectively. ERA5 is chosen for this evaluation as the data are available at one-hour intervals. The analysis shows _some differences_ in the magnitude of

$w$, with 12 GMT generally showing larger magnitudes compared to the daily means over

Gadanki (although no such systematic differences is observed in Kototabang). The directional

tendencies are also similar in both the profiles at both locations. This analysis shows that the results are not biased by taking data only at 12 UTC over Gadanki.

Our analysis to this point shows the level of consistency between the features observed by the radar and the reanalyses. To further understand the relative differences among the reanalyses we perform a monthly mean comparative analysis among the reanalyses, as shown in

Figure 8. In this case, we took ERAi as a reference and compare it with *w* products from other reanalyses. We chose ERAi because the zonal and meridional winds from this reanalysis have been shown to compare well with radiosonde and rocket sounding observations over the Indian equatorial region (*Das et al.*, 2015). The solid lines in Figure 8 show the differences over

Gadanki, while the dashed lines show differences over Kototabang. Over Gadanki, the difference between the ERAi and other reanalyses is less than ±0.5 cm/s during December-January-

February (DJF, winter). ERAi underestimates ERA5 compared to other reanalyses, while values based on MERRA-2 are relatively larger than those in other reanalyses. During MAM, strong downdrafts are found below 5 km with comparable magnitudes in all five reanalyses. ERAi underestimates ERA5 and NCEP-2 during March, and all other reanalyses from April to

September. Values of *w* in ERAi are larger than those in NCEP-2 above 8 km. All five reanalyses compare well at all atltiudes above 18 km. As expected, magnitudes are larger during

JJA than during other months. From October to November, the magnitude reduces to ±1 cm/s

with values from ERAi smaller than those from all other reanalyses except NCEP-2.

Over Kototabang, the magnitude of *w* is relatively larger than over Gadanki. It is interesting to note that, ERAi underestimates MERRA-2 in all months over this location also (MERRA-2 shows larger magnitudes compared to other reanalyses). Similarly values based on

EARi are larger than those based on NCEP-2. From December to February ERAi underestimates

(28)

MERRA-2 below 10 km and ERA5 between 10 and 15 km while overestimates NCEP-2 and

JRA-55. The overall bias pattern remains the same during MAM, except for differences relative to JRA-55. From June-November, ERAi underestimates NCEP-2 and overestimates  the other three reanalyses.

reanalyses with?

The direction of $w$ is an essential metric for comparing the observations .

We therefore show the directional tendencies from the IMSTR and the EAR measurements relative to those from the reanalysis data. Figure 9a shows the directional tendencies based on the

IMSTR and the reanalyses over Gadanki, while Figure 9b shows the directional tendencies based on the EAR and the reanalyses over Kototabang. The directional tendency is calculated at each height for every month when the radar or reanalysis data exceed 1 cm/s in either direction. The directional tendency for each month is estimated and then aggregated into seasons. These directional tendencies are given in terms of percentage of occurrence with respect to height. The     B

directional tendency is calculated for $w$ only if the magnitudes lie above $\pm 0.1$ cm/s for both radar retrievals and reanalyses. The tendency is calculated separately for updrafts and downdrafts.

(29)

(30)    Over Gadanki during DJF all reanalyses produce updrafts at rates of less than 10 % of around updrafts throughout the profile. During MAM these ratios increase to 15 %, with NCEP-2

producing updrafts about 25 % of the time. During JJA and SON, the percentage occurrence frequency?

increases with height from 25 % to a maximum of 50 % between 12 and 14 km. The percentage of updrafts occurrence then decreases from 14 to 20 km. This tendency trend is similar for all in reanalyses except ERA5, for which the percentage occurrence is less than 25 % during all seasons. The maximum ratio of updrafts over Gadanki is located between 12 and 15 km altitude.

The percentage occurrence of downdrafts over Gadanki is also less than 50 % at all levels. During DJF and MAM the reanalyses produce downdrafts 40 to 50 % of the time, a much

higher frequency compared to the updrafts (<10 %). However, these ratios decrease above 10

km. By contrast, the percentage of downdrafts produced during JJA and SON is less than that of the updrafts, with frequencies less than 25 % in all the levels during these seasons. The performance of ERA5 over Gadanki is very poor, as the occurrence frequencies are very small for both updrafts and downdrafts.

Over Kototabang the percentage occurrence of updrafts increases with height in all seasons, reaching a maximum of 75- 90 % between 10 and 14 km. Above 14 km the percentage decreases to a minimum of 5 % at 19 km. Updrafts are rarely produced by the reanalyses at altitudes less than 4 km. It is important to note that none of the reanalyses produce daily mean downdrafts exceeding 1 cm/s between 6 and 16 km. The percentage of downdrafts increases both below 6 km and above 17 km, where it reaches a maximum of about 25 to 50 %. MERRA-2,

NCEP-2 and JRA-55 show occurrence frequencies of downdrafts around 65 to 75 % above 18

km. The performance of ERA5 appears to be poor compared to the other reanalyses over this location as well.

**4    Summary**

The present study assesses the vertical motion ($w$) in reanalyses against direct radar observations from the convectively active regions Gadanki and Kototabang. The assessment is carried out for five different reanalyses: ERA-Interim, ERA-5, MERRA-2, NCEP-2 and JRA-55.

Measurements were collected using VHF radar at both locations. We have used 20 years of data from Gadanki and 17 years of data from Kototabang. The following points summarize the results of this unique study a.  The magnitude of $w$ obtained from reanalyses is underestimated by 10-50% relative to the radar observations.

[Figure]

b.  Observations over Gadanki showed updrafts from 8 to 20 km year around. The reanalyses only reproduced this feature during JJA and SON when magnitudes were larger than 0.5 cm/s in the reanalyses. However, the vertical location of the updrafts differs between the observations and the reanalyses. Downdrafts below 8 km are not captured well by reanalyses.

c.  Over Kototabang, the reanalyses did not consistently produce downdrafts below 8 km in all months. Updrafts in the UTLS are captured well; however, the peak in the vertical distribution of $w$ is different as over Gadanki.

d.  Inter-comparison among the reanalyses shows that ERAi overestimates NCEP-2 and underestimates the other three reanalyses with respect to the magnitude of $w$ over both

Gadanki and Kototabang.

e.  Assessment of directional tendencies shows that updrafts are reproduced reasonably well in all the five reanalyses but downdrafts are not reproduced at all.

Our analysis reveals that downdrafts are not well produced in reanalyses, and also the location of the largest updrafts is shifted lower than in the observations. Hence, the reanalyses should be used with care for representing various atmospheric motion calculations (viz. diabatic heating, convection, etc.) that mainly depend on the direction of $w$. This study provides the reanalysis community an initial basis to improve the methodology for calculating $w$ in reanalyses, as this is a much sought-parameter for atmospheric circulation calculations and analyses.

**Acknowledgements**

Authors would like to acknowledge all the technical and scientific staffs of National

Atmospheric Research Laboratory (NARL) and Research Institute of Sustainable Humanosphere (RISH), who directly or indirectly involved in the radar observations. Thanks to all the reanalysis

data centre for providing the data through the portal of Research data archival (RDA) of

NCEP/UCAR. One of the author KVS thank Indian Research Organisation for providing research associateship during this study.

**Data availability:** Analysed data (both radars and reanalyses) used in this study can be obtained on request. Raw time series data are available through open access in the following websites:

For Indian MST Radar : www.narl.gov.in

For EAR radar : www.rish-kyoto-u.ac.jp/ear/index-e.html

For ERAi, ERA-5, JRA-55 and NCEP-2.:  https://rda.ucar.edu          [E]

For MERRA-2 : https://disc.gsfc.nasa.gov.in

**Author's Contributions**

KNU conceived the idea for validation of vertical velocity among the reanalyses. SSD, MVR, and KVS collected and analysed the MST radar spectrum data. All the authors contribute for generation of figures, interpretation and manuscript preparation. The data used in the present study can be obtained on request.

**Conflict of Interest**

The authors declare that there is no conflict of interest.

[revised manuscript text omitted]

**Figure Captions**

**Figure 1.** Climatological monthly mean altitude profile of vertical velocity obtained from MST Radar and 5-reanalysis at 12 GMT over Gadanki. Horizontal lines indicate the standard error.

**Figure 2.** Same as Fig.1, but for diurnal mean over Kototabang.

**Figure 3 :** Monthly mean climatology of vertical velocity obtained from (a) radars, (b) ERAi, (c) ERA-5, (d) MERRA-2, (e) NCEP-2, and JRA-55 over Gadanki (left) and Kototabang (right). Gadanki data are at 12 GMT and Kototabang data are diurnal mean.

**Figure 4.** Monthly mean vertical velocity obtained from (a) MST Radar and (b) ERAi for 5 years interval (from top to bottom) over Gadanki (12 GMT).

**Figure 5.** Same as Fig.4 but for diurnal mean over Kototabang.

**Figure 6.** Height profile of vertical velocity at 12 GMT and diurnal mean (with 1 hour resolution) over Gadanki extracted from ERA-5 (highest available time resolution).

**Figure 7.** Same as Figure 6 but over Kototabang.

**Figure 8.** Comparison of relative differences in vertical velocity (*w*) between the reanalysis for Gadanki (solid line) and Kototabang (dash line). Individual month differences are estimated and then averaged for each month. Over Gadanki, data is taken for 12 GMT and for Kototabang it is diurnal.

**Figure 9.** Comparison of directional tendency simultaneously observed in radar and various reanalysis data sets for (a) Gadanki and (b) Kototabang. Updrafts are shown in top and third panels and downdrafts are shown in middle and bottom panels (for details see text).

[Figure]

[Figure]

**Figure 1.** Climatological monthly mean altitude profile of vertical velocity obtained from the MST Radar and five reanalyses over Gadanki at 12 UTC. Horizontal lines indicate the standard error in each data set.

[Figure]

[Figure]

[Figure]

**Figure 2.** Same as Fig.1, but for daily mean profiles over Kototabang.

[Figure]

[Figure]

**Figure 3 :** Monthly mean climatologies of vertical velocity obtained from (a) radars, (b) ERAi, (c) ERA5, (d) MERRA-2, (e) NCEP-2, and JRA-55 over Gadanki (left) and Kototabang (right). Gadanki data are at 12 GMT and Kototabang data are daily means.

[Figure]

[Figure]

velocities                                                    in

**Figure 4.** Monthly mean vertical velocity obtained from (a) MST Radar and (b) ERAi for 5-years intervals (from top to bottom) over Gadanki (12 GMT).

[Figure]

[Figure]

[Figure]

[Figure]

**Figure 5.** Same as Fig.S2 but for daily means over Kototabang.

[Figure]

[Figure]

[Figure]

**Figure 6.** Height profiles of vertical velocity for 12 GMT and from daily mean (with 1 hour resolution) over Gadanki extracted from ERA5 (highest available time resolution).

[Figure]

[Figure]

[Figure]

**Figure 7.** Same as Fig.6, but for Kototabang.

[Figure]

[Figure]

[Figure]

**Figure 8.** Comparison of relative differences in vertical velocity (*w*) between the reanalysis for Gadanki (solid line) and Kototabang (dash line). Individual month differences are estimated relative to ERAi and then averaged for each month.

[Figure]

[Figure]

**Figure 9.** Comparison of directional tendencies between the radars and various reanalysis data sets for (a) Gadanki and (b) Kototabang. Updrafts are shown in the upper panels and downdrafts are shown in the lower panels for each site (for details see text).

[Figure]

---

## Author Response (AR1)

**Response to Editor's Comments**

We would like to sincerely thank the Executive Editor, Editor (Dr.Gabriele Stiller) and all the three referees (2 anonymous and 1 interactive comments by Dr. Ulrich Schumann) for kind suggestions and comments, which helped in revising the manuscript. We have addressed all the reviewers' comments in order to make the manuscript publishable in your esteemed journal "Atmospheric Chemistry and Physics (ACP)". Point-by-point response on how we have addressed each recommendations/suggestions is given in the reply to the reviewer's comments and same is also implemented in the revised manuscript.

Please note that we have already responded to the open interactive comment by Dr. Ulrich Schumann and same is now implemented in the revised manuscript.

Now we are herewith submitting the revised manuscript and figures along with reply to the reviewer's comments for the consideration of publication.

All the authors listed on the manuscript concur with submission of the above mentioned manuscript.

We request Executive Editor and Editor to kindly process further and do the needful.

**Response to anonymous Referee #1's Comments**

**Q1:** This study compares and assesses vertical wind data from five global atmospheric reanalysis data sets. As independent measurements, it uses VHF radar measurements from two tropical stations, one at Gadanki, India, and the other at Kototabang, Indonesia. This is a very important trial and should be published in the end.

**R1**: We thank the reviewer-1 for his/her very positive comments and suggestions. We have addressed all the comments and implemented in the revised manuscript.

**Q2:** The main issue of the manuscript is, I think, in the apparent large discrepancy between the radar measurement results and the reanalysis data. On the other hand, I think we can say that the five reanalysis data sets show qualitatively similar seasonal and vertical distribution of w; there are differences, but considering that omega depends heavily on forecast model of each reanalysis system, without direct observations assimilated, these differences (among the reanalyses) may be understandable.

**R2**: Any reanalysis products assimilates as much as 107 observations per day, which is inclusive of both conventional (radiosonde, tower, aircrafts, wind profilers (wherever possible), etc.) as well as various satellite observations. It is to be noted that the vertical velocity provided by any reanalysis data centers is estimated indirectly from the horizontal wind components and temperature, which itself have mismatch among various reanalysis data (e.g., Das et al., 2016; Kawatani et al., 2016). Thus, this can possibly induce the discrepancy in the estimated vertical velocity among various reanalysis. Any wind profiler radar gives direct measurements of vertical velocity but over a single observational point. Whereas all the reanalysis data are averaged over the grid (e.g. ERA-Interim with 0.75 degree (latitude) x 0.75 degree (longitude)) which is possibly one of the main reasons for the mismatch between radar and reanalysis. We have now made a spatial sampling of w by considering different grid resolutions for a particular year and month to show the effect of spatial averaging on w, which is not included in the revised manuscript. In addition, it may also be due to the fact the different reanalysis uses different schemes and assimilation techniques.

**Q3:** I am afraid that we even need to start from suspecting any errors in the data processing and analysis for reanalysis omega. Did the data really come from the correct (intended) grid point? Is the conversion from omega to w really correct? I do believe that the authors did the correct procedure, but we need some more cross-check information to confirm that they really did, simply because the difference from the radar measurements is too large. One note is that the authors should remove data at some lowest levels where the reanalysis systems simply extrapolate data below the surface (the altitude of Gadanki and Kototabang is 360 m and 865 m, respectively).

**R3**: As mentioned in the manuscript, we have taken the data at the corresponding grid which is close to the radar location. We have used the equation (2) given in the revised manuscript to convert the omega (Pa s-1) into vertical velocity (m s-1). We have cross-checked every code and found no error. There are several literatures available, where vertical velocity is used from reanalysis (e.g. Uma et al., 2014; Das and Suneeth, 2020) as well as measured from MST/ST radar (e.g., Gage et al., 1991, 1992; Rao et al., 2003; Yamamoto et al., 2007; Rao et al., 2008; Uma and Rao 2009b, and both the magnitudes (re-analyses and radar) are comparable with the present study.

VHF radar at Gadanki provides observations from 3.6 km and Kototabang radar provides from 2 km, we have compared the re-analyses only from those altitudes.

**As suggested by the reviewer, we have removed the data from the re-analyses below 2 km.**

**Q4:** Regarding the radar measurements, it would be very useful to discuss why w shows such seasonal and vertical distributions. What processes produce upper tropospheric ascending motion and lower tropospheric descending motion, the latter for Kototabang for all seasons and for Gadanki for April to October? Are there any publications that discuss this? There are several publications on the measurements of VHF wind profiler at Christmas Island (2N, 157W) (Gage et al., 1988, 1991, 1992). Their vertical wind profiles may look rather similar to what Kototabang measurements show. Thus, their discussion may be useful. Once we (rather theoretically) understand how the actual w distributions may look like, we can get more insight on why all these reanalyses show such distributions.

**R4**: The vertical distribution of vertical velocity over Gadanki and Kototabang shows that the transport of air from the troposphere to the stratosphere is a twostep process. The downdraft below and updrafts above 10 km shows the convection transport of the air parcel up to 10 km and the slow ascending motion above 10 km slowly lifts the air parcel from 10 km to the stratosphere. This is prevalent in almost all the seasons over Gadanki and Kototabang. The detailed discussion on the seasonal and the vertical distribution of vertical velocity over Gadanki is explained by Rao et al., (2003), Rao et al., (2008) and Uma and Rao (2009). Rao et al., (2008) in addition to Gadanki data have also used Kototabang radar data to describe the seasonal and vertical characteristics of vertical velocity.

Following the reviewer-1's suggestion, we have discussed the VHF radar measured vertical velocity measurements over Christmas Island in the revised manuscript (Gage et al., 1991 and 1992). Gage et al. (1988) studied the comparison of only horizontal winds measured with VHF wind profiler with NMC and ECMF reanalysis thus, we are not including in our discussion.

**Q5:** In summary, I think we need much more information (maybe direct or maybe indirect) that may be useful to understand why the radar measurements and reanalysis data show such different distributions.

**R5**: Suggestion of the referee is well taken and we discussed these possibilities of differences between the reanalysis and radar observations in the revised manuscript. Additional analysis is also performed in one of the reanalysis data (ERAi) to evaluate the spatial averaging.

**Q6:** Page 2, lines 43-49, and page 3, line 68, and other places: The use of the terms "direct" and "indirect" may need to be reconsidered. "Direct" may be used for in situ measurements (e.g., radiosonde horizontal wind measurements), while "indirect" may be used as "indirect estimation" e.g., of w from horizontal wind measurements/data to consider their divergence/convergence. For the case of radar measurements, we may use the term "remote sensing" measurements, because these may not be "direct" measurements (they are not in situ measurements) but at the same time these may not be "indirect" which implies indirect estimation from other variables in the context of this manuscript.

**R6:** As suggested by reviewer we have modified the direct and indirect measurements.

**Q7: Pages 5-6, Section 2.1:**

The full location information on the two radar sites needs to be written in this section. The information on altitude, country/island, and the institutes that operate these radars is missing. Also, please explain the topography around each of these radar sites rather extensively. Gadanki is

located within high land of a continent (with a horizontal distance of \*\* km from the oceans), while Kototabang is located within a narrow mountain range of an island (of a scale of \*\* km in northwest-southeast and \*\* km in northeast-southwest), etc. The topographic information may be very important to judge the representativeness of reanalysis data at a particular grid point (and at the same time, the representativeness of each of these radar measurements).

The direct time information in UTC should be provided, because reanalysis data are in UTC. It would be useful to show the profiles of data number, i.e., of the original ones, of the quality controlled ones, of the finally used ones (after discarding data points >1 m/s and <-1 m/s), etc. The information on the quality control procedure is also needed. (The authors listed possible issues in the radar measurements, but they did not explain what they actually did to avoid such issues.)

**R7**: We have now given the topography information of both the sites, Gadanki and Kototabang. The figures are provided in UTC in the manuscript. The number of data points that have been discarded falls less than 1 % of the total data. As we know that the radar signal strength decreases with height following inverse square law. So we have adopted a procedure to neglect the data which does not lie within the specific threshold of SNR. This is described in the manuscript. The possible issues that can bias the vertical wind and how it is negligible in the mean *w* is already discussed by Rao et al., (2008).

**Q8:** Page 7, line 157: Which MERRA-2 data product was used, ASM or ANA?

**R8**: We have used ASM in MERRA-2. It is now mentioned in the revised manuscript.

**Q9**: Pages 9-, Section 3: Please see my comments above. Also, investigation of a case or two (e.g., for a week or for a month) might be useful to understand what is going on for both radar measurements and reanalysis data.

R9: Following the reviewer-1's suggestion, we have shown a case study for day-today comparison between the observations (Gadanki MST Radar and EAR) and reanalysis (ERAi). We thank referee for the valuable suggestion.

**Q10:** Figure 8: Results from Gadanki and Kototabang should be shown in separate panels/ figures. ERA-Interim is used as a reference. But, considering that ERA-Interim the Reanalysis Ensemble Mean (REM) may be a better reference. Finally, a reference should be "subtracted", e.g., ERA5 minus REF, not REF minus ERA5.

R10: Following the reviewer's suggestion, figures are now separated for Gadanki and Kototabang. Also, we have calculated the Reanalysis Ensemble Mean to show the inter-comparison between different re-analyses. The difference between the ensemble mean and the reanalysis is provided in the revised manuscript. We thank referee for this suggestion.

**References**

Das, S. S., Uma, K. N., Bineesha, V. N., Suneeth, K. V., and Ramkumar, G.: Four-decadal climatological intercomparison of rocketsonde and radiosonde with different reanalysis data: results from Thumba equatorial station, Q. J. Roy. Meteor. Soc., 142, 91–101, doi: 10.1002/qj.2632, 2016.

Das, S. S., & Suneeth, K. V.: Seasonal and interannual variations of water vapor in the upper troposphere and lower stratosphere over the Asian Summer Monsoon region-in perspective of the tropopause and ocean-atmosphere interactions. Journal of Atmospheric and Solar-Terrestrial Physics, 105244, https://doi.org/10.1016/j.jastp.2020.105244, 2020

Gage et al., A Comparison of Winds Observed at Christmas Island using a Wind- Profiling Doppler Radar with NMC and ECMWF Analyses, Bulletin American Meteorological Society, Vol. 69, No. 9, 1041-1046, 1988.

Gage et al., Long-Term Mean Vertical Motion over the Tropical Pacific: Wind-Profiling Doppler Radar Measurements, Science, Vol. 254, No. 5039, 1771-1773, 1991.

Gage et al., Diurnal variation in vertical motion over the central equatorial pacific from VHF wind profiling Doppler radar observations at Christmas Island (2 N , 157W ), Geophysical Research Letters, Vol. 19, No. 18, 1827-1830, 1992.

Kawatani, Y., Hamilton, K., Miyazaki, K., Fujiwara, M., and Anstey, J. A.: Representation of the tropical stratospheric zonal wind in global atmospheric reanalyses, Atmos. Chem. Phys., 16, 6681-6699, doi: 10.5194/acp-16-6681-2016, 2016.

Rao, T.N, K. N. Uma, D. Narayana Rao, and S. Fukao.: Understanding the transportation process of tropospheric air entering the stratosphere from direct vertical air motion measurements over Gadanki and Kototabang, Geophys. Res. Lett., 35, L15805, https://doi.org/10.1029/2008GL034220, 2008.

Rao, V., D. Rao, M. V. Ratnam, K. Mohan, and S. Rao.: Mean Vertical Velocities Measured by Indian MST Radar and Comparison with Indirectly Computed Values, J. App. Meteo, 42(4), 541-552, https://doi.org/10.1175/1520-0450(2003)042<0541:MVVMBI>2.0.C0;2, 2003.

Uma, K. N., and Rao, T. N.: Diurnal variation in vertical air motion over a tropical station, Gadanki (13.5oN, 79.2oE), and its effect on the estimation of mean vertical air motion, J. Geophys. Res., 114, D20106, https://doi.org/10.1029/2009JD012560, 2009b.

Uma, K. N., Das, S. K., & Das, S. S.: A climatological perspective of water vapor at the UTLS region over different global monsoon regions: observations inferred from the Aura-MLS and reanalysis data. Climate dynamics, 43(1-2), 407-420, https://doi.org/10.1007/s00382-014-2085-9, 2014

Yamamoto, M. K., N. Nishi, T Horinouchi, M Niwano, and S Fukao.: Vertical wind observation in the tropical upper troposphere by VHF wind profiler: A case study, Rad, Sci., 42, RS3005, https://doi.org/ 10.1029/2006RS003538, 2007.

**Response to anonymous Referee #2's Comments**

I like the premise of this paper very much: observational evaluation of reanalysis vertical velocities is much needed, perhaps especially within the broader Asian monsoon region. This is an important topic and I would like to see the paper published in the end. However, the presentation contains some significant gaps and unclear reasoning that should be addressed.

**We thank the reviewer for his positive comments and suggestions. We have addressed all the comments and implemented in the revised manuscript.**

**Q1:** My main reservation is that the comparison as presented is almost entirely descriptive, with little analysis of the causes of biases or how they might inform further improvement of the reanalysis products (see also comment D below). It would be very helpful – if not essential – to include more interpretation of both the differences among the reanalyses and the biases relative to observations. For example, the introduction indicates that section 4 includes both a discussion and a summary of the results, but section 4 itself includes little discussion, only summary. Some questions to consider:

**R1:** Any reanalysis products assimilate as much as 107 observations per day, which is inclusive of both conventional (radiosonde, tower, aircrafts, wind profilers (wherever possible), etc.) as well as various satellite observations. It is to be noted that the vertical velocity provided by any reanalysis data centres is estimated indirectly from the horizontal wind components and temperature, which itself have mismatch among various reanalysis data (e.g., Das et al., 2016; Kawatani et al., 2016). Thus, this can possible induce the discrepancy in the estimated vertical velocity among various reanalysis. Any wind profiler radar gives direct measurements of vertical velocity but over a single observational point. Whereas, all reanalysis data are averaged over a grid (e.g. ERA-Interim with 0.75 degree (latitude) x 0.75 degree (longitude)) which is probably one of the main reasons for the observed mismatch between radar and reanalysis.

Following the reviewer's suggestions we have revised the manuscript accordingly and estimated the following :

- (1) Difference between the individual reanalysis with ensemble averaging of all reanalysis (Reviewer-1).
- (2) We have now made a temporal and spatial sampling of w by considering different time and grid resolutions to show the effect of spatio temporal averaging on w.

Section-3 provides the results and discussion and section-4 only contains a summary. Now we have rephrased the sentence in the Introduction section.

**Q2:** Can we understand anything about what the reanalyses are doing wrong (or right) from the observational validation or reanalysis-only intercomparison results?

**R2**: We would like to point out to the reviewer that it is difficult to say whether reanalysis is doing wrong or right. In this aspect, we have contacted the reanalysis centre several times (through e-mail) but we could not find any definite answers from them. The inter-comparison of vertical velocity showed that there exist differences among the reanalysis itself (magnitude) as well with the radar observations. At this juncture, we would like to report the same which has never been reported earlier. However, we would like to point out that vertical velocity

in the reanalysis is an indirect estimation from the horizontal wind component and temperature, and earlier observations show that there is a mismatch among the horizontal wind of various reanalysis. In addition, it may also be due to the fact the different reanalysis uses different schemes and assimilation techniques, which is difficult to account for the bias in w. A table on schemes of different reanalyses data used in the present study is now included in the revised manuscript (Table 2).

**Q3:** Do the differences indicate major problems or can they be largely understood in terms of spatiotemporal sampling? For example, the narrow column observed by the radar relative to the reanalysis grid scale – do comparisons of reanalyses with grid scales spanning a factor ten offer any context here? In a related question, does resolution of the nearby topography come into play in any obvious ways?

**R3**: Following the reviewer's suggestion, we have done a small exercise with different spatial sampling with ERAi data set for a particular year and month and it is now included in the revised manuscript. The different grid spatial averaging does not significant impact on the *w*. The topography over the two locations can generate mountain waves if strong low-level winds are prevailing. Strong low-level winds are prevalent over Gadanki only from June to August and during these months, there is a critical level existing between 6 and 7 km due to the presence of strong wind shear, which will not support the propagation of mountain waves to higher altitudes. This wind shear exists throughout the year over Kototabang. Hence the effect of mountain waves will be minimal over both these locations on vertical velocity.

**Q4:** Are there any clues as to how different types of data assimilation (3D-Var vs IAU vs 4D-Var) influence biases in vertical velocities? What about details of the model physics, such as convective or boundary layer scheme?

**R4**: A table is attached in the revised manuscript about different schemes used by different reanalysis systems. It is very difficult to point out at this juncture that bias in the vertical velocity due to various assimilation schemes (Please see our R2).

**Q5:** How robust are the results between the two sites? Does this have any implications for which conclusions, if any, can be generalized?

**R5**: The results are robust from radars as it is only the technique for direct profiling the vertical velocity. There have been several global studies on vertical velocity during clear air as well as disturbed weather conditions using wind profiler radars. These results from the radar are consistent and hence the conclusions can be generalized.

**Q6:** I appreciate the authors' attention to earlier editorial comments. I have included an annotated manuscript with some additional (optional) suggestions, which also references the specific comments below.

**R6**: We sincerely thank the reviewer for his kind effort for providing an annotated manuscript. We have tried to address all the suggestions and comments raised by reviewer-2.

**Q7:** A (Sect. 2) The descriptions of the reanalyses in section 2 should include indications of how vertical velocities are computed in each reanalysis, whether these estimates are impacted by data assimilation (i.e. forecast versus analysis versus IAU in the case of MERRA-2), and whether they represent time-average or instantaneous estimates. It would also be helpful to give some basic

information about the model vertical coordinates, as I think vertical pressure velocities are usually estimated in these coordinates and then interpolated to the pressure grid. To the extent that these procedures are the same across multiple reanalyses the description could be consolidated, emphasizing differences in each subsection. Any other aspects of the model / data assimilation that might aid interpretation of the differences should also be included here.

**R7**: We thank the reviewer-2 for the suggestion. Now we include a separate tabular column describing the schemes used in various reanalysis. However, based on the schemes used in reanalysis, it is difficult for the authors to conclude the bias in omega. Reanalysis computes vertical velocity using kinematic and adiabatic methods. The omega obtained from reanalysis is time-averaged and not instantaneous. A generalized model vertical coordinates which are interpolated to the pressure grid and other details are now described in the revised manuscript.

**Q8.** (Method) Some of the specifics of the analysis are difficult to follow. For example, in 1.287 it is indicated that directional tendencies are reported for a given height for every month when either the radar or reanalysis data exceeded  $\pm 1$  cm s-1. This screening is based on the monthly mean, correct? Then, a few lines later, it is indicated that the directional tendency is calculated only for absolute magnitudes greater than 0.1cms-1. Is this now referring to the daily data? Are the results sensitive to these thresholds, especially in terms of the reanalysis evaluation?

**R8**. We only estimated the directional tendency, if the data lies above ±0.1 cms-1 in either direction in both reanalysis and radars. Reviewer has rightly pointed out that the screening is based on the monthly mean. These sentences are modified accordingly in the revised manuscript.

**Q9.** (Sect. 3, last two paragraphs) The directional tendency results for ERA5 are difficult to understand. The lack of strong updrafts or downdrafts at Gadanki seems to contradict the results shown in Fig. 3 (where ERA5 seems to have relatively strong vertical velocities and it is ERA-Interim more than ERA5 that looks like the outlier) and Fig. 6 (which shows a pretty robust seasonal cycle with many monthly averages well above the threshold). The results for Kototabang are likewise perplexing in the context of Fig. 3 and Fig. 7. How do you reconcile the directional tendency results in Fig. 9 with the profiles shown in these earlier figures?

**R9.** We extremely thank reviewer-2 for point out this issue w.r.t ERA5. This is issue has come due to the two different periods of data used in the analysis. We used 1995 to 2015 data for ERAi, MERRA2, JRA55 and NCEP2, whereas for ERA5 the data used is from 2002 to 2015 for Gadanki. Similar things happened for Kototabang. Now we have corrected in the revised figure and sincerely apologize for the mistake.

**Q9.** (Sect. 4, I.339-340) I like this thought, but more needs to be done to really provide a useful platform for improving the reanalyses. For one, it is not clear whether it is the 'methodology for calculating w' that lies behind the identified biases, as opposed to, e.g., different diurnal or day-today variations in convective occurrence (see comment 20), interactions between subgrid physical parameterizations and the large-scale flow, crude representation of local topography or land surface conditions, even just differences in spatial sampling area. This last is even suggested as the main reason for the differences in l.195-196, and it is not clear how reanalyses can address this beyond moving to finer and finer horizontal resolutions, which they are already doing. I recognize that it would be a monumental task to try to diagnose all of these and do not ask for a exhaustive investigation, but some investigation and discussion would be warranted. For example, operating on the hypothesis that the differences are mainly related to averaging, you could try imposing 'subgrid fluctuations' on the reanalysis products. What scale of fluctuation would need to be imposed to bring the reanalysis grid size? Do the results make physical sense, or do they suggest that other factors must contribute? **R9.** Suggestion of the referee is well taken and we discussed these possibilities of biases and differences between the reanalysis and radar observation in the revised manuscript. Additional analysis is also performed in one of the reanalysis data (ERAi) to evaluate the spatial and temporal averaging (please refer R3).

**Q10.** (Data availability) The data citations are incomplete. Both NASA (for MERRA-2) and the NCAR RDA (for all other reanalyses) have assigned doi numbers to the datasets used in this paper. These doi values should be used in data citations (input data doi at https://citation.crosscite.org/ for citation details) to help the data providers track the impact of their investment. Dates of access should also be included (since reanalyses occasionally undergo reprocessing to fix errors), along with the specific variables and resolutions used (to ensure reproducibility).

**R10**.The data citations are now provided along with DOI in the revised manuscript. The data resolution is now provided in section-2.**

**Q11.** (Figure 3) Is it possible that the vertical profiles for MERRA-2 at Kototabang have been inverted somehow? The differences between these and ERA5/ERA-Interim are pretty striking, perhaps especially the downward shift of the maximum during May–June in MERRA-2 relative to the upward shift of the maximum in ERA5 and ERA-Interim (not to mention the radar profiles). I know that the orientation of the vertical coordinate may differ (top-to-bottom, bottom-to-top) in data files released by these different reanalysis producers. Please double-check this for MERRA-2, and perhaps also for NCEP-2.

**R11.** We have rechecked the program/coding and found no error in the orientation of vertical coordinate (height).**

**Q12.** (Figure 8) Could Fig. 8 be made more effective building off of the presentation in Fig. 3, using difference plots relative to a particular reanalysis-based benchmark? I agree that the current presentation could help in terms of explicitly comparing quantitative biases across different reanalyses, but it is very difficult to pick out details of the individual profiles in the current figure. Another option might be to consolidate some months with similar profiles (it looks like the canonical seasons might work, but warm/cold/transition seasons could also work) and then split the Gadanki and Kototabang profiles (results for the two sites do not seem to share that much in common in the vertical distributions).

**R12.** Fig. 8 (old manuscript) is now modified by doing ensemble averaging of all the reanalysis and each reanalysis is subtracted from the ensemble averaging. We have given separate figures for Gadanki and Kototabang in the revised manuscript.

**Q13.** (Figure 9) The caption says this figure shows a comparison between the radars and various reanalysis products. Where are the directional tendencies based on the radar data? It is difficult to evaluate the reanalyses without this information. Please excuse me if I am missing something really basic about the presentation.

**R13**. The directional tendency is calculated if both radar and reanalysis observe updrafts/downdraft simultaneously at the same height considered. So there will not be any tendency for radar and reanalysis separately.

**Q14.** (1.33) Perhaps also mention the role of subsidence and adiabatic warming in the formation of stable inversion layers?

**R14**. Sentence is included in the revised manuscript.

**Q15.** (*l.36,38*) I think the use of 'control' here rather overstates the case, especially in l.36. It would be enough to delete 'controlling'; the second use in l.38 is ok on its own. **R15. The word controlling is now omitted.**

**Q16.** (1.49) Please clarify what is meant by 'global estimates' and its relationship to 'direct measurements' versus 'indirect estimates'; there is also an extra comma here.

**R16**. The word "globally" is now omitted and the sentence is revised accordingly.

**Q17.** (1.57) Maybe mention here that this source of uncertainty is particularly important for reanalyses, where assimilation increments in horizontal winds may be comparable to this. It might also be helpful to rephrase the sentence to emphasize this assimilation adjustment rather than 'error'.

**R17.** The above sentences are rephrased in the revised manuscript.

**Q18.** (1.65) The connection of the above discussion to reanalysis estimates of vertical velocity should be made more explicit (i.e., why do these concerns apply to reanalysis products specifically) **R18**. These sentences are rephrased in the revised manuscript.

**Q19.** (1.66) Suggest rewriting this sentence: 'reanalyses involve many approximations and assimilation-related adjustments, and are not error-free' **R19**. The above sentence is rephrased in the revised manuscript.

**Q20.** (1.84) Technically a reanalysis vertical profile is also a column over a single location, just one with a broader footprint. Here it would be helpful to specify how the area of the column differs between the radar, the finest-grid reanalysis (0.25°, right?) and the coarsest-grid reanalysis (2.5°).

**R20.** The area the radar looks at the height of 20 km is about 14 km (East-West) x 14 km (North-South), which is about 0.12 degree x 0.12 degree and it is much finer at a lower height (like an inverted cone). The finest grid is 0.28 (ERA5) and the coarsest grid is 2.5 (NCEP2/DOE) with respect to reanalysis and the grid resolution is uniform with height.

**Q21.** (1.86) Phrasing needs care here: a number of studies have evaluated vertical motion across reanalyses (in the context of trajectories, wave activity, large-scale motion, etc.), so the primary novelty of this work is the evaluation against radar observations.

**R21**. The above sentences are included in the revised manuscript.

**Q22.** (1.88) This point is a little repetitive.

**R22.** The point is deleted in the revised manuscript.

**Q23.** (1.119) Rephrase: 'Quality control metadata for the EAR measurements are available online' **R23**. The above sentence is rephrased in the revised manuscript.

**Q24.** (1.140) How long is 'long'?**

**Q24.** The vertical velocity is averaged over ten years. This is now included in the revised manuscript.**

**Q25.** (l.146) How is this 21 km upper limit identified in the reanalysis profiles, and approximately what pressure level does this correspond to? More generally, many of the results are presented in altitude coordinates. Are heights computed from the pressure levels assuming a constant temperature, from geopotential outputs from each reanalysis, another approach? Are data linearly interpolated to a common height grid? Could this have any influence on the comparisons in this paper (e.g., the reanalysis-derived updraft maxima being located lower than those observed by the radar)?

**R25.** As we know that the radar signal strength decreases with height following inverse square law. Thus, the maximum height coverage by Gadanki MST radar as well as Kototabang Equatorial Atmospheric Radar is up to 21 km which is about 50 hPa, beyond which the signal is noisy. Thus, we took all reanalysis maximum

up to 21 km (if available). No reanalysis data were interpolated. The pressure coordinate is directly converted into height using Hypsometric equation. For intercomparison, common height between radar and reanalysis is chosen.

**Q25.** (l.152) Citation year for Hoffmann et al. should be 2019 **Reply: Corrected.**

Q26.(l.153) The ERA5 paper is now in early online release (https://rmets.onlinelibrary.wiley.com/doi/10.1002/qj.3803) R26. Citation corrected

**Q27.** (l.165) NCEP-2 (as denoted here) was undertaken as a cooperation between NCEP and the Department of Energy (DOE); care should be taken in this text to acknowledge this and distinguish it from the NCEP-NCAR Reanalysis 1

**R27.** Corrected in the revised manuscript.**

**Q28.** (l.171) The paragraph mentions only the original model resolution, but from this description the data are taken from the 2.5° data grid

**R28.** Data is chosen from the nearest grid.**

**Q29.** (1.178) This information is not provided for the other reanalyses nor is it clear how it relates to the estimation of vertical velocity. I suggest that the authors try to provide a consistent and concise set of information for each reanalysis in this section, focusing on the points relevant to the data used in this paper.

**R28.** Now description about each reanalysis and also a table about all the reanalyses is provided in the revised manuscript.

**Q30.** (l.186) for dry air **R30.** Corrected.

**Q31.** (l.188) daily mean is evaluated for 00–24UTC or shifted to match local solar time? I guess it shouldn't matter much as long as this is consistent between the radar and reanalysis

**R31.** For Gadanki the instantaneous value at 12 GMT is taken whereas, for Kototabang, the entire 24 hrs of reanalysis data is averaged for the daily mean.

**Q32.** (1.195) It looks like convective days in the reanalysis products are defined based on the screening from the radar data. How consistently do the reanalyses identify convective versus non-convective days based on measurements at the radar site? Wouldn't this screening also be sensitive to the differences in grid size? Some sensitivity testing would be helpful here, perhaps using precipitation thresholds as well as w.

**R32**. Convective days may influence the long-term averaging of vertical velocity (for details see Uma and Rao., 2009), thus it has to be removed from the analysis. The radar vertical velocity threshold is a good proxy to identify the presence of convection (Uma and Rao., 2009) and thus the convective days are identified from the radar vertical velocity, if it is above  $\pm 1$  m s-1. These convective days are removed from both radar data as well as reanalysis data sets.

Precipitation thresholds cannot be used for removing the convective days as there may be dry convection where vertical velocity may exceed ±1 m s-1 but no precipitation is observed at surface level (e.g. Uma and Rao., 2009).

Short scale convection both in spatial as well as temporal scale is difficult to be captured by any reanalyses.

**Different grid averaging of reanalysis (ERAi) is now included and discussed in the revised manuscript.**

**Q33.** (1.198) The meaning of 'global' here is not clear — should this be 'qualitative' to set it against 'quantitative' differences that might result from differences in sampling area (but see also comment 20)?

**R33**. The word "Global" is omitted.**

**Q34.** (l.206) ERA5 should be written without the hyphen (as it is now in much of the manuscript; thank you)

**R34.** Corrected.**

**Q35.** (1.216) It is not clear from the text whether these 'significant variations' are in the seasonal cycle, in the diurnal cycle, or both (since the previous sentence discusses seasonal variations in the diurnal cycle). Some additional clarification would be helpful.

**R35.** It is seasonal variations in the diurnal cycle. The sentence is revised accordingly.

**Q36.** (1.220) The temporal treatment is another source of potential differences in vertical velocities between the observational data (time averages over at least one hour) and reanalyses (usually instantaneous outputs, I think – please check – and usually only four times per day). Naively, it seems like this might offset some of the smoothing effect of the spatial sampling difference, but it should be mentioned and discussed either way.

**R36**. The reanalysis outputs are averaged products. The issue of spatial sampling of reanalysis is addressed in figure 9 in the revised manuscript. Please refer R3.

**Q37.** (l.231) Could this comparison be sensitive to the definition of 'convective' days? This may be especially relevant for Gadanki where the reanalyses may even have different diurnal cycles of convection. For example, Bechtold et al. (2014) reported that changes to the forecast model between ERA-Interim and ERA5 resulted in substantially improved representation of the diurnal cycle of convection over land.

**R37.** It is to be noted that radar been a single point observation, even the small scale convection can be captured over both the radar location. During such event (even in a small scale convections), we observed high vertical velocity beyond  $\pm 1$  ms-1. Hence, data used in the present study after the screening is convection free which will not affect the results.

**Q38.** (1.249) Does this result generalize to the observational validation — i.e. do the EAR results for Kototabang also support this conclusion? This information could be added to Fig. 7 and related discussion.

**R38.** Yes, Gadanki results will be valid for Kototabang and it is already shown in Fig.9 (revised manuscript).

**Q39.** (1.272) Some additional care might be needed in the presentation here, to distinguish when the values being quoted are absolute magnitudes of w (as here) versus when they are biases relative to a particular benchmark (as earlier in the paragraph).

**R39.** We have now revised as ensemble averaging and discussion is modified in the revised manuscript.

**Q40.** (1.278) This presentation ('underestimates'/'overestimates') is a little strange, since it seems to imply an evaluation of ERA-Interim against multiple benchmarks as opposed to an intercomparison among (presumably) equally uncertain reanalysis-based products. A more specific and less judgmental phrasing might help, something like: 'XX shows smaller positive values

than YY and larger positive values than ZZ', or 'XX and YY both show downdrafts, but with larger amplitudes in XX', or maybe using stronger/weaker updrafts/downdrafts if you prefer. **R40. Following reviewer-2's suggestion, the phrases are revised accordingly.**

**Q41.** (1.292) Should delete the space between 10 and % (also elsewhere in this and following paragraphs).

**R41**. Deleted.

**Q42.** (l.293) Is this ratio indicating that 10% of all 12UTC values exceed 0.1 cms-1 or that 10% of all positive values have magnitudes greater than 0.1 cms-1? If the first, it is enough to remove 'of updrafts'; if the second, some additional text to clarify is necessary.

**R42**. The figure represents that 10% of the time both radar and reanalysis observed updrafts simultaneously over a particular location. Now this sentence is rephrased in the revised manuscript.

**Q43.** (1.301) Suggest to be more explicit: 'The fraction of downdrafts decreases above ...' **R43.** Corrected.

**Q44**.Is this 'reaches a maximum' specifically referring to the increase above 17 km or to both the increase below 6km and above 17 km? If the latter, 'a maximum' should be changed to 'maxima'. **R44. Corrected.**

**Q45.** (1.330) Here, do you mean 'the location of the peak w differs between radar and reanalysis data, as it also does over Gadanki' or just that 'the vertical location of the w maximum over Kototabang is different from that over Gadanki'?

**R45**. The location of the peak w differs between radar and reanalysis.

**Q46.** (1.331) Again, suggest revising this presentation style to be clearer and more objective (see also comment 28).

**R46.** Corrected.

**Q47.** (I.338) These examples are a little strange. It is true that the behavior of physical parameterizations in the reanalyses (used to generate diabatic heating and convective mass fluxes) may be impacted by large-scale convergence / divergence (and hence by the same factors used to compute w), though the feedbacks between w and model physics are two-way, complex, and pretty different from how they are implied to be here. Or perhaps the authors refer to diagnosed diabatic heating (e.g., Yanai et al., 1973) and vertical motion along Lagrangian trajectory pathways? Note that the latter should be distinguished from 'convection', which is included in some transport models but I think usually based on vertical stability rather than w.

**R47.** Following reviewer-2's suggestion, we have modified these sentences accordingly.

**Q48.** (Fig. 5 caption) Should this reference be to Fig. 4? **R48.** Corrected

**References** :**

Das, S. S., Uma, K. N., Bineesha, V. N., Suneeth, K. V., and Ramkumar, G.: Four-decadal climatological intercomparison of rocketsonde and radiosonde with different

reanalysis data: results from Thumba equatorial station, Q. J. Roy. Meteor. Soc., 142, 91–101, doi: 10.1002/qj.2632, 2016.

Kawatani, Y., Hamilton, K., Miyazaki, K., Fujiwara, M., and Anstey, J. A.: Representation of the tropical stratospheric zonal wind in global atmospheric reanalyses, Atmos. Chem. Phys., 16, 6681-6699, doi: 10.5194/acp-16-6681-2016, 2016.

**Response to Interactive comments (by Dr. Ulrich Schumann)**

**Q1.** I fully agree that vertical air motion is crucial for atmospheric dynamics and under resolved in most of present numerical weather prediction products. This is true not only for the tropics but at all latitudes. I appreciate this study as a relevant contribution using interesting radar data (with which I have only little experiences).Please let me point out that vertical wind can be measured – at least in principle – also by other instruments including research aircraft. I recently published a study (Schumann, 2019) on relationships between horizontal kinetic energy spectra of vertical wind and horizontal divergence of the divergent horizontal wind components, which can be separated from the rotational wind components by known Helmholtz decomposition methods. I compared with airborne wind measurements in the upper troposphere and lower stratosphere at mid-latitudes and compared to some available model data. In particular, I found a total of 80 % of w variance near the tropopause occurring at scales between about 0.5 and 80 km. Perhaps these findings, and some of the related literature cited in my paper are worth mentioning in your paper.

**R1.** We are very much thankful to the Dr. Schumann for going through our manuscript and providing positive suggestions. We have gone through this interesting paper and included all the necessary points in the revised manuscript.

1 2

3 4

5

**Assessment of vertical air motion among reanalyses and qualitative comparison with VHF radar measurements over the two tropical stations**

K. N. Uma1, Siddarth Shankar Das1, M. Venkat Ratnam2, and K. V. Suneeth1

1Space Physics Laboratory, Vikram Sarabhai Space Centre, ISRO, Trivandrum-695022, India 2National Atmospheric Research Laboratory, Dept. of Space, Gadanki-517112, India

6 7

**8 \*e-mail : urmi\_nmrf@yahoo.com**

**9 Abstract**

Vertical wind (w) is one of the most important meteorological parameters for 10 understanding a range of different atmospheric phenomena. Very few direct measurements of 11 w are available so that most of the time one must depend on reanalysis products. In the 12 present study, assessment of w among selected reanalyses, (ERAj, ERA5, MERRA-2, 13 NCEP/DOE-2 and JRA-55) and qualitative comparison of those datasets with VHF radar 14 measurements over the convectively active regions Gadanki (13.5°N and 79.2°E) and 15 Kototabang (0°S and 100.2°E) are presented for the first time. The magnitude of w derived 16 17 from reanalyses is 10-50% less than that from the radar observations. Radar measurements of 18 w show downdrafts below 8 to 10 km and updrafts above 8-10 km over both locations. Inter-19 comparison between the ensemble of reanalyses with respect to individual reanalysis shows 20 that ERAi, MERRA-2 and JRA-55 compares well with the ensemble compared to ERA5 and NCEP/DOE-2. There is no significant improvement in the w due to the effect of different 21 spatial sampling. Directional tendency shows that the percentage of updrafts captured is 22 reasonably good, but downdrafts are not well captured by all reanalyses. Thus, caution is 23 advised when using vertical velocities from reanalyses. 24 25

26

27

| Formatted: \ | Nidth: | 8.27", Height: | 11.69" |  |
|--------------|--------|----------------|--------|--|
|              |        |                |        |  |

| Deleted: Only v    |  |
|--------------------|--|
|                    |  |
| Deleted: and       |  |
|                    |  |
|                    |  |
| Deleted: -Interim  |  |
| Deleted: - Interim |  |

Key Words: Vertical velocity, MST Radar, Equatorial Atmosphere Radar, Reanalysis

| ĺ  |                                                                                                         | / |
|----|---------------------------------------------------------------------------------------------------------|---|
| 38 | 1 Introduction                                                                                          |   |
| 39 | Vertical air motion (w) in any region of the Earth's atmosphere reflects the structure                  |   |
| 40 | and dynamical features of that region. Importantly, in the lower part of the atmosphere,                |   |
| 41 | sudden widespread changes in the weather are usually associated with variations in vertical             |   |
| 42 | air motion. The magnitude of $w$ is a factor of ten or more smaller than the horizontal wind;           |   |
| 43 | nevertheless, it is crucial in the evolution of severe weather (Peterson and Balsley, 1979).            | < |
| 44 | Adiabatic cooling associated with upward motion leads to the formation of clouds and                    |   |
| 45 | precipitation and adiabatic warming associated with downward motion leads to the                        |   |
| 46 | dissipation of clouds. In addition, subsidence leads to adiabatic warming, which results in the         |   |
| 47 | formation of stable inversion layers. Extensive studies have been done on the relationships             |   |
| 48 | between w and precipitation/convection over the tropics (Back and Bretherton, 2009; Uma                 |   |
| 49 | and Rao, 2009a; Rao et al., 2009; Uma et al., 2011 and references therein). Thus, w plays a             |   |
| 50 | vital role in day-to-day changes in the weather. Different scales of variability exist in w             |   |
| 51 | ranging from microscale to meso synoptic, and planetaryscales (Uma and Rao, 2009b). It                  |   |
| 52 | also controls energy and mass transport between the upper troposphere and lower                         | < |
| 53 | stratosphere (Yamamoto et al., 2007, Rao et al., 2008). In a nutshell, knowledge of w is                |   |
| 54 | helpful for evaluating virtually all physical processes in the atmosphere. Hence precise                |   |
| 55 | measurements of w could serve a guiding factor for studying many processes in the                       |   |
| 56 | atmosphere.                                                                                             |   |
| 57 | The small magnitudes of $w$ make it very difficult to measure, as the errors involved in                |   |
| 58 | measurements often exceed the actual values. Direct and indirect methods exist to measure w      |   |
| 59 | (e.g. Doppler measurements using radars for profiling, sonic anemometers in the boundary                |   |
| 60 | layer and also aircrafts ) as well as indirect computational methods (e.g., adiabatic, kinematic |   |
| 61 | and quasi-geostrophic vorticity/omega methods). Remote sensing measurements of w are thus        |   |
| 62 | restricted to locations where radars are situated. Using aircrafts Schumann, (2019) studied the         |   |

| Deleted: co  | ntrolling |      |  |
|--------------|-----------|------|--|
| Deleted: lik | e         |      |  |
|              |           |      |  |
| Deleted: the | e         |
 |  |
| Deleted: the | e         |      |  |

| Deleted: are often larger than                                                                                                      |
|-------------------------------------------------------------------------------------------------------------------------------------|
| Deleted: and                                                                                                                        |
| Deleted: Direct                                                                                                                     |
| Formatted: Font: Italic, Complex Script Font: Italic                                                                                |
| Deleted: using aircrafts                                                                                                            |
| Formatted: Font: (Default) Times New
Roman, 12 pt, Not Italic, Complex Script Font:
Times New Roman, 12 pt, Not Italic |
| Deleted: on                                                                                                                         |
| Formatted: Font: (Default) Times New
Roman, 12 pt, Not Italic, Complex Script Font:
Times New Roman, 12 pt, Not Italic |

| 78 | relationships between horizontal kinetic energy spectra of vertical wind and horizontal      |
|----|----------------------------------------------------------------------------------------------|
| 79 | divergence of the divergent horizontal wind components, by separating it from the rotational |
| 80 | wind components by known Helmholtz decomposition methods. In general, w is derived           |
| 81 | diagnostically from horizontal winds and temperatures, which is an indirect estimation, This |
| 82 | estimation gives a general view on the distribution of ascending and descending motion on    |
| 83 | the synoptic-scale within the quasi-geostrophic framework (Tanaka and Yatagai, 2000; Rao     |
| 84 | et al. , 2003).                                                                       |

| 85  | Reanalyses evaluate the vertical pressure velocity (omega) using indirect estimation                     |
|-----|----------------------------------------------------------------------------------------------------------|
| 86  | (e.g., Dee et al., 2011). Any reanalyses products assimilate as much as 10 7 observations per |
| 87  | day, which is inclusive of both conventional (radiosonde, tower, aircrafts, wind profilers               |
| 88  | (wherever possible), etc.) as well as various satellite observations. However, reanalyses                |
| 89  | combine both observations and model outputs to produce systematic variation in the                       |
| 90  | atmospheric state (e.g., Fujiwara et al., 2017). It is to be noted that the vertical velocity            |
| 91  | provided by any reanalysis data center is estimated indirectly from the horizontal wind                  |
| 92  | components and temperature, which itself has mismatch among various reanalyses data (e.g.,               |
| 93  | Das et al., 2016; Kawatani et al., 2016). Thus, this can possibly induce the discrepancy in the          |
| 94  | estimated vertical velocity among various reanalyses. For example, in the kinematic method,              |
| 95  | omega is estimated by integrating the mass continuity equation assuming inviscid adiabatic               |
| 96  | flow. However, this kinematic estimate suffers from uncertainties in the observations as          |
| 97  | omega is estimated from horizontal divergence (Tanaka and Yatagai, 2000). This source of                 |
| 98  | uncertainty is particularly important for reanalyses, where assimilation increments in                   |
| 99  | horizontal winds may be comparable to the uncertainty. A 10% error in the wind may lead to               |
| 100 | a 100% error in the estimated divergence (Holton, 2004). Omega from the thermodynamic                    |
| 101 | energy equation is less sensitive to horizontal winds as it mainly depends on the temperature            |
| 102 | gradient. However, in this method the local rate of change in temperature must be measured               |

| Formatted: Font: (Default) Times New
Roman, 12 pt, Not Italic, Complex Script Font:
Times New Roman, 12 pt, Not Italic |
|------------------------------------------------------------------------------------------------------------------------------|
| Deleted: Globally                                                                                                            |
|                                                                                                                              |
| Deleted: Global estimates are derived                                                                                        |
| Deleted:                                                                                                                     |
| Deleted: Indirect estimation,                                                                                                |
|                                                                                                                              |
| Deleted:                                                                                                                     |

| -{ | Deleted: i |
|----|------------|
| -{ | Deleted: s |
| Y  | Deleted: s |

| Deleted:                                                    |
|-------------------------------------------------------------|
|                                                             |
| Formatted: Font: Italic, Complex Script Font: Italic |
| Deleted: i                                                  |
| Deleted: e                                                  |
| Deleted: s                                                  |
| Deleted: ve                                                 |
| Formatted: Font: Italic, Complex Script Font: Italic        |
| Formatted: Font: Italic, Complex Script Font: Italic |
|                                                             |
|                                                             |

| -{ | Deleted: errors         |
|----|-------------------------|
| -{ | Deleted: uncertainities |

| - | Deleted: uncertainity                                |
|---|------------------------------------------------------|
|   |                                                      |
| - | Formatted: Font: Italic, Complex Script Font: Italic |
| 4 | Deleted:                                             |

accurately, meaning that observations must be taken at frequent intervals in time to estimate

∂T /∂t accurately (*Holton*, 2004). This methodology fails in areas of strong diabatic heating,
especially where condensation and evaporation are involved. The quasi-geostrophic method
for estimating omega neglects ageostrophic effects, friction and diabatic heating (*Stepanyuk et al.*, 2017). It is to be noted from the above discussions that calculating *w* from indirect
estimation has more uncertainties. Hence reanalyses that use indirect estimation, involve
underlying approximations and assimilations and are not error-free (*Kennedy et al.*, 2012).

Other indirect methods can be used to derive w from radar measurements in the 127 middle and upper atmosphere, where direct measurements of vertical wind are not possible 128 129 due to technical constraints. These methods include Doppler weather radar, Medium 130 Frequency (MF) radar and meteor radar. Doppler weather radar uses an indirect method to calculate vertical winds (Liou and Chang, 2009; Matejka, 2002). Meteor radar also cannot 131 determine vertical velocity directly as the winds are determined from meteor showers using a 132 wide beamwidth. As a consequence, Laskar et al. (2017) calculated vertical wind from 133 134 meteor wind radar data based on a "Kinematic" method using the continuity equation and 135 hydrostatic balance. Dowdy et al. (2001) have calculated vertical wind using the horizontal 136 momentum and mass continuity equations from the MF radar data. However, indirect 137 methods are only adopted when direct methods cannot be used.

Very-high frequency (VHF) and ultra-high frequency (UHF) vertical pointing radars arethe most powerful tools for determining vertical air motion (velocity) with high temporal and vertical resolution. However, the magnitude may still not be directly comparable between reanalysis products and observations as the reanalyses provide the intensity of vertical air motion over wide areas (> 25 km2), whereas the radar measurements provide information for a narrower column over a single location. Thus, the best way to assess reanalysis estimates of w against radar measurements is to compare its directional tendencies, A number of studies Deleted: .
Italic

| Formatted: Font: Italic, Complex Script Font
Italic                          | : |
|----------------------------------------------------------------------------------------|---|
| Deleted: that                                                                          |   |
| Deleted:                                                                               |   |
| Deleted: s                                                                             |   |
| Deleted: are not                                                                       |   |
| Deleted: owing to the many underlying approximations and assimilations involved |   |
| Deleted: There are few                                                                 |   |
| Deleted: by which we can                                                               |   |

|               | Formatted: No Spacing, Indent: First line: 0.31", Adjust space between Latin and Asian text, Adjust space between Asian text and numbers |
|---------------|-------------------------------------------------------------------------------------------------------------------------------------------------|
| $\overline{}$ | Deleted: the                                                                                                                                    |
|               | Deleted: directly                                                                                                                               |
|               |                                                                                                                                                 |
|               | Deleted: reanalyses                                                                                                                             |
|               |                                                                                                                                                 |
|               | Deleted: direct                                                                                                                                 |
|               |                                                                                                                                                 |
|               | Deleted: the                                                                                                                                    |
|               |                                                                                                                                                 |
| _             | Deleted: with those of radar                                                                                                                    |

| 164 | have evaluated vertical motion across reanalyses (in the context of trajectories, wave activity,            |                                   |
|-----|-------------------------------------------------------------------------------------------------------------|-----------------------------------|
| 165 | large-scale motion, etc.), so the primary novelty of this work is the evaluation against radar              |                                   |
| 166 | observations. The present study focuses on the assessment of $w$ among various reanalyses                   | Deleted: To                       |
| 167 | using VHF radar measurements from two tropical stations where the convective activity is                    | derived from
these product     |
| 168 | frequent: Gadanki and Kototabang, Evaluations of this type are critically important as                      | Deleted: , v
| 169 | reanalyses estimates of $w$ are widely used by the scientific community to understand and                   | Deleted:                          |
| 170 | simulate a variety of atmospheric processes. In section 2, the data and methodology are                     |                                   |
| 171 | described. Section 3 provides results and discussion followed by summary and concluding                     |                                   |
| 172 | remarks in section 4.                                                                                       | Deleted: . S followed by a |
| 470 |                                                                                                             | in section 4.
Formatted        |
| 173 | 2 Data and Methodology
2 1 Radar measurements                                                            |                                   |
| 175 | Remote sensing measurements of w are obtained from the Indian Mesosphere-                                   | Deleted: D                        |
| 176 | Stratosphere Troposphere Pader (IMSTP) located at Gadanki (13.5°N and 79.2°E) and the                       |                                   |
| 170 | Stratosphere Troposphere Radai (INISTR) focated at Gadanki ( $(15.5)$ N and $(75.2)$ and the                |                                   |
| 177 | Equatorial Atmosphere Radar (EAR) located at Kototabang ( $0.2^{\circ}$ S and $100.2^{\circ}$ E). Figure 1a |                                   |
| 178 | and 1b show, the topography map of the location of both the radars, i.e. Gadanki and                        | Deleted: s                        |
| 179 | Kototabang, respectively, generated by using the Shuttle Radar Topography Mission (SRTM)                    | Deleted: ,                        |
| 180 | data (Farr et al., 2007). Gadanki is located in the southern peninsula of tropical India, about             | Formatted
Italic               |
| 181 | 90 km off the east coast and it is surrounded by hills. Kototabang is located in the western                | Deleted: r                        |
| 182 | part of Sumatra Island and EAR is situated in the mountainous region with the highest peak                  | Deleted: S                        |
| 183 | of about 2 km. Both the IMSTR and EAR are pulsed coherent radars operating at 53 MHz                        |                                   |
| 184 | and 47 MHz, respectively. These instruments are used to estimate w by measuring the                         | Deleted: (II                      |
| 185 | Doppler shift in the vertical beam. The technical details and operational parameters of the                 | Deleted: (B                       |
| 186 | IMSTR have been given by Rao et al. (1995) while those for the EAR have been given by                       | Deleted: ,                        |
| 187 | Fukao et al, (2003). Both the radars specifications and parameters used for the present                     | Deleted: ,                        |
| 188 | measurements are listed in Table 1.                                                                         |                                   |
|     |                                                                                                             |                                   |

To the author's knowledge, no studies yet ning with the assessment of *w* products n different reanalyses and evaluation of cts against radar measurements. which is therefore first of its kind, 3.5°N and 79.2°E) (0.2°S and 100.2°E)

Section 3 contains the main results a discussion and summary of the results 1: Font color: Text 1

[revised manuscript text omitted]

| Formatted: Font: Italic, Complex Script Font:
Italic     |
|-------------------------------------------------------------|
| Deleted: ,                                                  |
| Formatted: Font: Italic, Complex Script Font:
Italic     |
| Formatted: Font: Italic, Complex Script Font:
Italic     |
| Deleted: &                                                  |
| Formatted: Font: Italic, Complex Script Font:
Italic     |
| Formatted: Font: Italic, Complex Script Font: Italic |

| -[ | Formatted: Font: Italic, Complex Script Font: Italic    |
|----|---------------------------------------------------------|
| 1  | Deleted: &                                              |
|    | Formatted: Font: Italic, Complex Script Font:
Italic |
| Y  | Deleted: ,                                              |
| _  | Deleted: not affected                                   |
| _  | Deleted: not affected                                   |

|   | Deleted: The ERA-Interim (          |
|---|-------------------------------------|
| 1 | Formatted: Indent: First line: 0.4" |
| 1 | Deleted: )                          |

| 276 | satellite observations. The model has improved in the representation of moist physical                              |
|-----|---------------------------------------------------------------------------------------------------------------------|
| 277 | processes. Advances have also been made with respect to soil hydrology and snow in land                             |
| 278 | surface models. The detail of the model is given in (Dee et al., 2011). We use 6-hourly                             |
| 279 | vertical velocities from the ECMWF Interim reanalysis (ERAi) from 1995 to 2015. The grid                            |
| 280 | resolution of ERAi is $0.75^{\circ}$ (latitude) x $0.75^{\circ}$ (longitude). The nearest grid points are taken for |
| 281 | Gadanki (13.68°N, 79.45°E) and Kototabang (0.35°S, 100.54°E). Although 37 pressure levels                           |
| 282 | up to 1 hPa resolution are available, we have restricted the dataset to 21 km, which is about                       |
| 283 | 50 hPa, as that is the maximum radar range.                                                                         |
| 284 | 2.3 ERA5                                                                                                            |
| 285 | ERA fifth-generation (ERA5) is the atmospheric reanalysis produced by ECMWF. It is                                  |
| 286 | an improved version of ERAi. The data assimilation scheme used is 4D-Var and it assimilates                         |
| 287 | the NCEP stage IV quantitative precipitation estimates produced over the USA by combining                           |
| 288 | precipitation estimates from the Next-Generation Radar (NEXRAD) network with gauge                                  |
| 289 | measurements. The moist physics scheme is improved by including freezing rain. The long                             |
| 290 | wave radiation scheme is modified in ERA5. The evolution of the top soil layer, snow and                            |
| 291 | sea ice temperatures are included. It uses observations from various satellites which include                       |
| 292 | upper air temperature, humidity and ozone. It also used bending angles from GNSS. It                                |
| 293 | provides much higher spatial (30 km) and temporal resolution (hourly) from the surface up to                        |
| 294 | 80 km (137 levels). ERA5 also features much improved representation especially over the                             |
| 295 | tropical regions of the troposphere and better global balance of precipitation and evaporation.                     |
| 296 | Many new data types not assimilated in ERAi are ingested in ERA5 ( Hoffmann et al. , 2019 ).          |
| 297 | The grid resolution of ERA5 is 0.28° (latitude) x 0.28° (longitude). The details are available                      |
| 298 | in (Hersbach et al., 2020). We have taken hourly data from ERA5. The nearest grid points are                        |
| 299 | again taken for Gadanki (13.63°N, 79.31°E) and Kototabang (0.14°S, 100.40°E), and the data                          |
| 300 | period is 2002-2015.                                                                                                |
|     |                                                                                                                     |

eleted:

| Moved (insertion) [1]                                                                               |
|-----------------------------------------------------------------------------------------------------|
| Deleted: are                                                                                        |
| Formatted: Font: Italic, Complex Script Font:
Italic                                             |
| Deleted: European Centre for Medium-Range
Weather Forecasts (                             |
| Deleted: )                                                                                          |
| Moved (insertion) [2]                                                                               |
| Deleted: degree                                                                                     |
| Deleted: degree                                                                                     |
| Moved up [1]: (Dee et al. , 2011).                                                           |
| Moved up [2]: The grid resolution of ERAi is 0.7 degree (latitude) x 0.7 degree (longitude). |
| Formatted: Font color: Custom
Color(RGB(35,31,32))                                               |
| Deleted:                                                                                            |
| Formatted: Indent: First line: 0.4"                                                                 |
| Deleted:                                                                                            |
| Deleted: fifth generation of                                                                 |

| Formatted: Font: Italic, Complex Script Font:
Italic |
|---------------------------------------------------------|
| Deleted: 2018                                           |
|                                                         |

| Deleted: The details are available in Copernicus climate change service report |  |
|---------------------------------------------------------------------------------------|--|
| Deleted: and Dee 2016 and
https://cds.climate.copernicus.eu/cdsapp#!/home). |  |
| Deleted: °°                                                                           |  |

| 322 | 2.4 MERRA-2                                                                                                            |   |
|-----|-------------------------------------------------------------------------------------------------------------------------------|---|
| 323 | The Modern-Era Retrospective analysis for Research and Applications, version 24                                               |   |
| 324 | (MERRA-2) is the latest reanalysis of the modern satellite era produced by the National                                       |   |
| 325 | Aeronautics and Space Administration's (NASA) Global Modelling and Assimilation Office                                        |   |
| 326 | (GMAO). The scheme used in MERRA-2 is an improved version of MERRA. It uses a three-                                          |   |
| 327 | dimensional variational (3D-Var) algorithm based on the grid point statistical interpolation                                  |   |
| 328 | and also uses an incremental analysis update. It assimilates bending angle observations,                                      |   |
| 329 | satellite radiances from both polar as well as geostationary infra-red and microwave                                          |   |
| 330 | sounders. In addition it also assimilates water vapor and ozone. MERRA-2 includes aerosol                                     |   |
| 331 | analysis and provide data for 42 pressure levels from the surface to 0.01 hPa with a temporal                                 |   |
| 332 | resolution of 3 h and horizontal resolution of 0.5° (latitude)x_0.625° (longitude), We used                                   |   |
| 333 | MERRA-2 Assimilation (ASM) data. Details have been provided by Gelaro et al. (2017).                                          |   |
| 334 | The nearest grid points are used for Gadanki (13.5°N, 79.37°E) and Kototabang (0.14°S,                                        |   |
| 335 | 100.00°E), with data spanning from 1995 to 2015.                                                                       |   |
| 336 | 2.5 NCE P/DOE-2                                                                                                        |   |
| 337 | The National Center for Atmospheric Research and Department of Energy                                                         |   |
| 338 | (NCEP/DOE-2) reanalysis is an updated version of NCEP-1 by fixing the known processing                                        |   |
| 339 | errors in NCEP-1. The variational scheme used is 3D-Var and it provides more accurate                                         |   |
| 340 | pictures of soil wetness and near-surface temperature over land, the land surface hydrology                                   |   |
| 341 | budget, snow cover, and radiation fluxes over the ocean. It is based on the NCEP operational                                  |   |
| 342 | model with a horizontal resolution of 209 km and 28 vertical levels. The temporal coverage is                          |   |
| 343 | four times per day. NCEP/DOE-2 products are improved relative to NCEP-1, having fixed                                         |   |
| 344 | errors and updated parameterizations of physical processes, as evaluated by Kanamitsu et al.                                  |   |
| 345 | (2002). The grid resolution of NCEP/DOE-2 is $2.5^{\circ}_{\tau}$ (latitude) x $2.5^{\circ}_{\tau}$ (longitude). The data for | < |
|     | 1                                                                                                                             |   |

**Deleted: ¶**

| Deleted: MERRA-2 data                                                                                |
|------------------------------------------------------------------------------------------------------|
| Deleted: are                                                                                         |
| Deleted: d                                                                                           |
| Deleted: on                                                                                          |
| Deleted: in                                                                                          |
| Deleted: by                                                                                          |
| Deleted: in                                                                                          |
| Deleted: •                                                                                           |
| Deleted: coverage                                                                                    |
| Deleted: The grid resolution of MERRA2 is 0.5 degree (latitude) x 0.625 degree (longitude). ¶ |
| Deleted: P-2                                                                                         |
|                                                                                                      |

the present study covers from 1995 to 2015 and is extracted at the nearest grid points to
Gadanki (12.5°N, 77.5°E) and Kototabang (0, 100.00°E).

369 2.6 JRA-55

The Japanese 55-year reanalysis (JRA-55) is an updated version of the earlier JRA-370 25 with new data assimilation and prediction systems (Kobayashi et al., 2015). New radiation 371 schemes, higher spatial resolution and 4D-var data assimilation with variational bias 372 373 correction for satellite radiances have been used to generate the JRA-55 products. This 374 reanalysis includes variation in greenhouse gas concentrations with time, as well as the new 375 representations of land surface parameters, aerosols, ozone and sea surface temperature, The grid resolution of JRA-55 is 1.25° (latitude) x 1.25° (longitude). The nearest grid points are 376 taken for Gadanki (13.75°N, 78.75°E) and Kototabang (0, 100°E) and the data period is 1995-377 2015. 378

- 379 For all the reanalyses data, w (in cm s-1) is estimated using the formula\_:
- 380

$$w = -\frac{1}{g}\omega \frac{RT}{p}$$

where  $\omega$  is the vertical velocity in pressure coordinates (in Pa s-1), T is the absolute temperature (K), p is the atmospheric pressure (hPa) and R (=287 J kg-1 K-1) is the gas constant for dry air. To compare measured vertical wind with the reanalysis products, we take the reanalysis data corresponding to 12 GMT for Gadanki and the daily mean for Kototabang. The details of the schemes used in reanalysis are provided in Table 2.

386 3 Results and Discussion

```
Figure 2 shows the inter-comparision of layer averaged daily w measured from
IMSTR with different reanalyses (ERAi, ERA5, MERRA-2, NCEP/DOE-2, and JRA-55)
over Gadanki for (a) January 2007, and (b) August 2007. Both radar and all the reanalyses
data sets are taken at 12 UTC, and the month and year are chosen in such a way to have
maximum days of radar observations in two different seasons (winter and summer).
```

| Deleted: S                                                                            |
|---------------------------------------------------------------------------------------|
| Deleted: S                                                                            |
| Deleted: Temperatue                                                                   |
| Deleted: T                                                                            |
| Deleted: (SST)                                                                        |
| Deleted: s                                                                            |
| Deleted: degree                                                                       |
| Deleted: degree                                                                       |
| Deleted: The horizontal resolution of the forecast model is ~60 km for JRA-55. |
| Deleted: ¶                                                                            |
| Formatted: Indent: First line: 0"                                                     |

(2)

| -(     | Deleted: t |  |
|--------|------------|--|
| $\neg$ | Deleted: 1 |  |
|        |            |  |

| ĺ | Formatted: Font: Italic, Complex Script Font:
Italic |
|---|---------------------------------------------------------|
| ( | Deleted: Radar                                          |
| ~ |                                                         |
| ł | Deleted: i                                              |

| 408                             | Similarly, EAR observation is also compared with different reanalysis data but for January                                                                                                                                                                                                                                                                                                                                                                                                                                                              |
|---------------------------------|---------------------------------------------------------------------------------------------------------------------------------------------------------------------------------------------------------------------------------------------------------------------------------------------------------------------------------------------------------------------------------------------------------------------------------------------------------------------------------------------------------------------------------------------------------|
| 409                             | 2008 and August 2008 as shown in Fig.3. However, both EAR and reanalysis data are diurnal                                                                                                                                                                                                                                                                                                                                                                                                                                                               |
| 410                             | averaged (24 hrs). It is observed that the magnitude of w measured from radar observations is                                                                                                                                                                                                                                                                                                                                                                                                                                                    |
| 411                             | an order higher than the reanalysis data over both the locations (Gadanki and Kototabang).                                                                                                                                                                                                                                                                                                                                                                                                                                                              |
| 412                             | Most of the time, reanalysis data are comparable in direction with radar observations,                                                                                                                                                                                                                                                                                                                                                                                                                                                                  |
| 413                             | whenever updrafts are observed. It is also observed that there is mismatch between the $w_{-}$                                                                                                                                                                                                                                                                                                                                                                                                                                                          |
| 414                             | estimated in the different reanalyses. Gage et al. (1992) described that by averaging radar                                                                                                                                                                                                                                                                                                                                                                                                                                                             |
| 415                             | data for a long-period of time can give a better measurement of w in clear-air condition and                                                                                                                                                                                                                                                                                                                                                                                                                                                            |
| 416                             | thus in this context, we have taken long-term averaging.                                                                                                                                                                                                                                                                                                                                                                                                                                                                                                |
| 417                             | Figure $4$ shows the climatological monthly mean altitude profile of w obtained from                                                                                                                                                                                                                                                                                                                                                                                                                                                                    |
| 418                             | the IMSTR (observations) and the ERAi, ERA5, MERRA-2, NCEP/DOE-2 and JRA-55                                                                                                                                                                                                                                                                                                                                                                                                                                                                             |
| 419                             | reanalysis data over Gadanki. Although the magnitudes are of the same order between the                                                                                                                                                                                                                                                                                                                                                                                                                                                          |
| 420                             | observations and reanalyses, significant differences are identified in the figures. Convective                                                                                                                                                                                                                                                                                                                                                                                                                                                          |
| 421                             | days are discarded from the radar data (observations) as mentioned in the previous section,                                                                                                                                                                                                                                                                                                                                                                                                                                                             |
| 422                             | and those days are also eliminated from all reanalysis data sets. The quantitative differences                                                                                                                                                                                                                                                                                                                                                                                                                                                          |
| 423                             | may be attributed to the spatial averaging implicit in the reanalyses products, whereas the                                                                                                                                                                                                                                                                                                                                                                                                                                                             |
| 424                             | radar measurements are for a single point. Thus we only discuss the tendency of $w$ as it is                                                                                                                                                                                                                                                                                                                                                                                                                                                            |
| 425                             | used to represent the variation of w, rather than its magnitude. The IMSTR observations show                                                                                                                                                                                                                                                                                                                                                                                                                                                            |
| 426                             | updrafts between 8 and 20 km from December to April, with the largest values in the tropical                                                                                                                                                                                                                                                                                                                                                                                                                                                            |
| 427                             |                                                                                                                                                                                                                                                                                                                                                                                                                                                                                                                                                         |
|                                 | tropopause layer (TTL, 12-16 km), These features are not reproduced by any of the                                                                                                                                                                                                                                                                                                                                                                                                                                                                       |
| 428                             | tropopause layer (TTL, 12-16 km), These features are not reproduced by any of the reanalyses, which all show downdrafts from December to April between 1 km and the                                                                                                                                                                                                                                                                                                                                                                                     |
| 428
429                      | tropopause layer (TTL, 12-16 km), These features are not reproduced by any of the reanalyses, which all show downdrafts from December to April between 1 km and the tropopause level (mean tropopause is ~ 16.5 km). By comparison, downdrafts are observed in                                                                                                                                                                                                                                                                                          |
| 428
429
430               | tropopause layer (TTL, 12-16 km), These features are not reproduced by any of the reanalyses, which all show downdrafts from December to April between 1 km and the tropopause level (mean tropopause is ~ 16.5 km). By comparison, downdrafts are observed in the IMSTR below 6 km in April, which may be attributed to pre-monsoon (March-May)                                                                                                                                                                                                        |
| 428
429
430
431        | tropopause layer (TTL, 12-16 km), These features are not reproduced by any of the reanalyses, which all show downdrafts from December to April between 1 km and the tropopause level (mean tropopause is ~ 16.5 km). By comparison, downdrafts are observed in the IMSTR below 6 km in April, which may be attributed to pre-monsoon (March-May) precipitation and evaporation ( Uma and Rao , 2009a). Vertical velocity in ERAi differs in                                                                                                      |
| 428
429
430
431
432 | tropopause layer (TTL, 12-16 km), These features are not reproduced by any of the reanalyses, which all show downdrafts from December to April between 1 km and the tropopause level (mean tropopause is ~ 16.5 km). By comparison , downdrafts are observed in the IMSTR below 6 km in April, which may be attributed to pre-monsoon (March-May) precipitation and evaporation ( Uma and Rao , 2009a). Vertical velocity in ERAi differs in both magnitude and direction from other reanalyses, especially in the lower troposphere from |

| Deleted: radar                                       |
|------------------------------------------------------|
| Deleted: sets                                        |
| Deleted: radar                                       |
|                                                      |
| Formatted: Font: Italic, Complex Script Font: Italic |
| Deleted: sets in                                     |
|                                                      |
| Deleted: sets                                        |
|                                                      |
| Deleted: are                                         |
| Formatted: Font: Italic, Complex Script Font: Italic |
| Deleted: have                                        |

| Formatted: Indent: First line: 0.5"      |   |
|------------------------------------------|---|
| Deleted: 1                               |   |
| Deleted: 2                               |   |
|                                          |   |
| Deleted: reanlalyses                     |   |
| Deleted: reanlalysis                     |   |
| Deleted: sets                            |   |
| Deleted: It is to be noted that c |   |
| Deleted: in                              |   |
| Deleted: analysis                        | _ |
| Deleted: s                               |   |
| Deleted: the                             |   |
| Deleted: These                           |   |
| Deleted: the                             |   |
| Deleted: in the present study,    |   |
| Deleted: global                          |   |
|                                          |   |
| Deleted: ,                               |   |
|                                          |   |
| Deleted: from December to April.         |   |
|                                          |   |

| Deleted: Comparatively |  |
|------------------------|--|
| Deleted: comparisson   |  |

| 458 | March to June. Meanwhile, the magnitude of vertical velocity in ERA5 is a little larger than               |           |
|-----|------------------------------------------------------------------------------------------------------------|-----------|
| 459 | that in the other reanalyses_from May to June. Updrafts are observed in the TTL by the                     |           |
| 460 | IMSTR during June, when all reanalyses show similar features but only located below the                    |           |
| 461 | TTL. During July and August both the radar observations and the reanalyses show updrafts in                |           |
| 462 | the vicinity of the TTL. Updrafts are observed in the TTL from September to November but                   |           |
| 463 | the peak in the updrafts is shifted lower than that observed by the IMSTR. Below 8 km, the                 |           |
| 464 | IMSTR shows downdrafts from April to October. The reanalyses data are unable to                            |           |
| 465 | reproduce downdrafts above 2 km.                                                                           | $\langle$ |
| 466 | We have also analyzed w from the EAR (Kototabang) where the observations are                               |           |
| 467 | available for the full diurnal cycle (measurements of hourly averages for 24 hrs of                        |           |
| 468 | observations). All reanalyses data over Kototabang are averaged for the full diurnal cycle.                |           |
| 469 | Figure 5 shows the monthly mean climatology of daily mean w from the EAR observations                      |           |
| 470 | and the five reanalyses over Kototabang. All the reanalyses agree well with each other over                |           |
| 471 | Kototabang. The updrafts in the TTL are well reproduced by all five reanalyses although the                |           |
| 472 | magnitude and vertical location of the maximum in w remain lower than observed. However                    |           |
| 473 | none of the reanalyses reproduces the downdrafts. A distinct bimodal distribution in $w$ from              |           |
| 474 | May to September (two peaks between 8-10 km and 14-17 km) with a local minimum                             |           |
| 475 | between 12 and 13 km is observed in the EAR measurements which is not observed in the                      |           |
| 476 | reanalysis. The magnitudes of both updrafts and downdrafts are larger than those observed                  |           |
| 477 | over Gadanki. JRA-55 produces the largest w among the reanalyses. The monthly means                        |           |
| 478 | show significant differences in the direction of w between the observations and the reanalyses      |           |
| 479 | below 6 km.                                                                                                |           |
| 480 | Gage et al. (1992) studied the long-term diurnal variability of w at Christmas Island•×             | $\langle$ |
| 481 | (2°N) and found the w varies between $\pm 4$ cm s -1 . The observations showed updrafts below 4 |           |

| Deleted: 2 |  |
|------------|--|
| Deleted: 3 |  |

**Formatted: Indent: First line: 0.5"**

km, downdrafts between 4-14 km and updrafts above 12 km. Gage et al. (1991) have

482

| 501 | explained that the downward motion in the troposphere is consistent with a heat balance in                |
|-----|-----------------------------------------------------------------------------------------------------------|
| 502 | the clear-air between adiabatic warming of descending air and radiative cooling to space. The             |
| 503 | ascending motion in the upper troposphere and lower stratosphere is due to large diabatic                 |
| 504 | heating caused by ice particle in the cirrus. Rao et al. (2008) have shown the long-term mean      |
| 505 | of w over Gadanki and Kototabang and found w varies between -0.3 to +0.6 cm s -1 . The         |
| 506 | authors observed downdrafts below 6 km and updrafts above it in all the seasons. The mean                 |
| 507 | pattern of w profile observed by radars over all the tropical sites (i.e. Christmas Island,               |
| 508 | Gadanki and Kototabang ) show similar characteristics and explain that the vertical transport             |
| 509 | of air from the troposphere to the lower stratosphere is a two-step process as discussed by               |
| 510 | Rao et al. (2008). Uma and Rao (2009b) have reported the diurnal variation of w in different       |
| 511 | seasons, although their observations had only 1-2 diurnal cycles per month over Gadanki.                  |
| 512 | They found significant variations in the seasonal variability of diurnal cycle as large as $\pm 6$ cm.    |
| 513 | $s_{\star}^{-1}$ over Gadanki using IMSTR. The present observations are limited to 16:30 to 17:30 IST,    |
| 514 | with all reanalyses data over Gadanki taken at 12 UTC (17:30 IST) . Thus, time-averaged     |
| 515 | climatological mean biases can be neglected.                                                              |
| 516 | To establish the robustness of the results we have used different averaging procedures                    |
| 517 | to assess the consistency of the variability in $w$ at monthly scales. Monthly mean                       |
| 518 | climatological profiles of w from radar observations and various reanalyses over Gadanki and              |
| 519 | Kototabang are shown in Figure \$1 (supplementary) . Downdrafts in the troposphere are not         |
| 520 | captured by any of the reanalyses over either location. By contrast, updrafts in the TTL are              |
| 521 | generally reproduced in the monthly mean, though their magnitudes are often underestimated                |
| 522 | by the reanalyses. ERAi underestimates the magnitude of both updrafts and downdrafts over                 |
| 523 | Gadanki, while NCEP/DOE-2 underestimates the magnitude of updrafts over Kototabang.                       |
| 524 | Monthly means calculated over five-year periods from both the radar data and ERAi                         |
| 525 | are shown in Figure 6 for Gadanki and Figure 7 for Kototabang. The reanalysis shows similar |
|     |                                                                                                           |

| Deleted: balnce                                                                                                                                                                                                                                                                                                                                                                                                                                                                                                                    |
|------------------------------------------------------------------------------------------------------------------------------------------------------------------------------------------------------------------------------------------------------------------------------------------------------------------------------------------------------------------------------------------------------------------------------------------------------------------------------------------------------------------------------------|
| (                                                                                                                                                                                                                                                                                                                                                                                                                                                                                                                                  |
| Moved (insertion) [10]                                                                                                                                                                                                                                                                                                                                                                                                                                                                                                             |
| Deleted: 6 km                                                                                                                                                                                                                                                                                                                                                                                                                                                                                                                      |
| Deleted:                                                                                                                                                                                                                                                                                                                                                                                                                                                                                                                           |
| Deleted: above                                                                                                                                                                                                                                                                                                                                                                                                                                                                                                                     |
| Formatted: Font: Italic, Complex Script Font:
Italic                                                                                                                                                                                                                                                                                                                                                                                                                                                                            |
| Deleted: s                                                                                                                                                                                                                                                                                                                                                                                                                                                                                                                         |
| Deleted: a                                                                                                                                                                                                                                                                                                                                                                                                                                                                                                                         |
| Deleted: explained                                                                                                                                                                                                                                                                                                                                                                                                                                                                                                                 |
| Formatted: Font: Italic, Complex Script Font: Italic                                                                                                                                                                                                                                                                                                                                                                                                                                                                        |
| Deleted: ,                                                                                                                                                                                                                                                                                                                                                                                                                                                                                                                         |
| Deleted: ¶                                                                                                                                                                                                                                                                                                                                                                                                                                                                                                                         |
| Deleted: . T                                                                                                                                                                                                                                                                                                                                                                                                                                                                                                                       |
| Deleted: have                                                                                                                                                                                                                                                                                                                                                                                                                                                                                                                      |
| Deleted: /                                                                                                                                                                                                                                                                                                                                                                                                                                                                                                                         |
| Formatted: Superscript                                                                                                                                                                                                                                                                                                                                                                                                                                                                                                             |
| Deleted: reanalysis                                                                                                                                                                                                                                                                                                                                                                                                                                                                                                                |
| Deleted: GMT                                                                                                                                                                                                                                                                                                                                                                                                                                                                                                                       |
| Deleted: Further                                                                                                                                                                                                                                                                                                                                                                                                                                                                                                                   |
| Moved up [10]: Rao et al. (2008) have shown the long-term mean of $w$ over Gadanki and Kototabang and found $w$ varies between -0.3 to +0.6 cm s -1 . The authors observed downdrafts below 6 km and updrafts above 6 km in all the seasons. The above mean pattern explains that the vertical transport of air from the troposphere to the stratosphere is a two-step process as explained by Rao et al. (2008).                                                                                         |
|                                                                                                                                                                                                                                                                                                                                                                                                                                                                                                                                    |
| Formatted: Font: Italic, Complex Script Font: Italic                                                                                                                                                                                                                                                                                                                                                                                                                                                                               |
| Formatted: Font: Italic, Complex Script Font:
Italic Deleted: ¶ ¶ Radar measurements of w at this location consistently show updrafts in the TTL region and downdrafts below 6 km (e.g. Rao et al. , 2008, Gage et al., 1991; 1992). ¶                                                                                                                                                                                                                                                                            |
| Formatted: Font: Italic, Complex Script Font:
Italic Deleted: ¶ ¶ Radar measurements of w at this location
consistently show updrafts in the TTL region and
downdrafts below 6 km (e.g. Rao et al. , 2008, Gage
et al., 1991; 1992). ¶ Deleted: ¶                                                                                                                                                                                                                                                        |
| Formatted: Font: Italic, Complex Script Font:
Italic Deleted: ¶ ¶ Radar measurements of w at this location consistently show updrafts in the TTL region and downdrafts below 6 km (e.g. Rao et al. , 2008, Gage et al., 1991; 1992). ¶ Deleted: ¶                                                                                                                                                                                                                                                                 |
| Formatted: Font: Italic, Complex Script Font:
Italic Deleted: ¶ ¶ Radar measurements of w at this location consistently show updrafts in the TTL region and downdrafts below 6 km (e.g. Rao et al. , 2008, Gage et al., 1991; 1992). ¶ Deleted: ¶ Formatted Deleted: chrained from both the chearantings of                                                                                                                                                                                                       |
| Formatted: Font: Italic, Complex Script Font:
Italic Deleted: ¶ ¶ Radar measurements of w at this location consistently show updrafts in the TTL region and downdrafts below 6 km (e.g. Rao et al. , 2008, Gage et al., 1991; 1992). ¶ Deleted: ¶ Formatted Deleted: obtained from both the observations af Deleted: respectively                                                                                                                                                                                        |
| Formatted: Font: Italic, Complex Script Font:
Italic Deleted: ¶ ¶ Radar measurements of w at this location consistently show updrafts in the TTL region and downdrafts below 6 km (e.g. Rao et al. , 2008, Gage et al., 1991; 1992). ¶ Deleted: ¶ Formatted Deleted: obtained from both the observations al Deleted: respectively Deleted: 3                                                                                                                                                                      |
| Formatted: Font: Italic, Complex Script Font:
Italic Deleted: ¶ ¶ Radar measurements of w at this location consistently show updrafts in the TTL region and downdrafts below 6 km (e.g. Rao et al. , 2008, Gage et al., 1991; 1992). ¶ Deleted: ¶ Formatted Deleted: obtained from both the observations a Deleted: respectively Deleted: 3 Deleted: 4                                                                                                                                                                   |
| Formatted: Font: Italic, Complex Script Font:
Italic Deleted: ¶ ¶ Radar measurements of w at this location consistently show updrafts in the TTL region and downdrafts below 6 km (e.g. Rao et al., 2008, Gage et al., 1991; 1992). ¶ Deleted: ¶ Deleted: 1 Deleted: contained from both the observations al Deleted: respectively Deleted: 3 Deleted: 4 Deleted: they                                                                                                                                                          |
| Formatted: Font: Italic, Complex Script Font: Italic         Deleted: ¶         ¶         Radar measurements of w at this location consistently show updrafts in the TTL region and downdrafts below 6 km (e.g. Rao et al., 2008, Gage et al., 1991; 1992). ¶         Deleted: ¶          Formatted          Deleted: ¶          Deleted: ¶          Deleted: 1          Deleted: obtained from both the observations af       Deleted: respectively         Deleted: 3       Deleted: 4         Deleted: they       Deleted: they |
| Formatted: Font: Italic, Complex Script Font:
Italic Deleted: ¶ ¶ Radar measurements of w at this location consistently show updrafts in the TTL region and downdrafts below 6 km (e.g. Rao et al. , 2008, Gage et al., 1991; 1992). ¶ Deleted: ¶ Deleted: ¶ Deleted: obtained from both the observations at Deleted: respectively Deleted: 3 Deleted: 4 Deleted: they Deleted: overestimated Deleted: and                                                                                                               |
| Formatted: Font: Italic, Complex Script Font:
Italic Deleted: ¶ ¶ Radar measurements of w at this location consistently show updrafts in the TTL region and downdrafts below 6 km (e.g. Rao et al. , 2008, Gage et al., 1991; 1992). ¶ Deleted: ¶ Formatted Deleted: obtained from both the observations a Deleted: respectively Deleted: 3 Deleted: 4 Deleted: they Deleted: and Deleted: and Deleted: 4                                                                                                                |

behavior to the overall climatology in each five-year average. The overall patterns of updrafts
and downdrafts in the radar measurements of vertical velocity are also similar, indicating a
consistent performance of the radar over the full 20 year analysis period.

602 To further elucidate potential biases in the results due to averaging, we have taken ERA5 at 12 UTC and compared it to the daily mean (obtained by averaging w at different 603 604 times of the day) to show that the sampling restrictions at Gadanki do not bias the results 605 obtained. Figures  $\underline{8}$  and  $\underline{9}$  show the mean w obtained at 12  $\underline{\text{UTC}}$  and also the mean obtained 606 by averaging hourly analyses for each day for Gadanki and Kototabang, respectively. ERA5 607 is chosen for this evaluation as the data are available at one-hour intervals. The analysis 608 shows some differences in the magnitude of w, with 12 UTC generally showing larger magnitudes compared to the daily means over Gadanki (although no such systematic 609 differences are observed in Kototabang). The directional tendencies are also similar in both 610 the profiles at both locations. This analysis shows that the results are not biased by taking 611 612 data only at 12 UTC over Gadanki.

613 Our analysis to this point shows the level of consistency between the features 614 observed by the radar and those in the reanalysis. To further understand the relative 615 differences among the reanalyses we perform a monthly mean comparative analysis among 616 the reanalyses, as shown in Figures 10 and 11 for Gadanki and Kototabang, respectively. We 617 take an ensemble mean of all the five reanalyses and then subtracted the ensemble mean from each reanalysis. The differences are less than  $\pm 0.5$  cm  $s^{-1}$  during December-January-February 618 (DJF, winter), During MAM, the difference between the ensemble and reanalysis show  $\pm 2$ 619 cm s-1 below 5 km. Below 5 km NCEP/DOE-2 and ERAi is less, whereas ERA5, Merra-2 620 and JRA-55 are more than the ensemble. The difference above 6 km is less than  $\pm 0.5$  cm s-1 621 above 6 km. JRA-55 shows a good comparison with the ensemble and above 10 km all the 622 reanalyses the differences are minimal with the ensemble. During the monsoon (JJA), the 623

| Deleted: -   |  |
|--------------|--|
| Deleted: GMT |  |

| Deleted: 6        |
|-------------------|
| Deleted: 7        |
| Deleted: 7        |
| Deleted: 8        |
| Deleted: GMT      |
| Deleted: analysis |
| Deleted: GMT      |

| Deleted: is                                                                                                                                                                                                                                                                                                                                                                                                                                                                                                                                                                                                                                                                                                                                                                                                                                                                 |
|-----------------------------------------------------------------------------------------------------------------------------------------------------------------------------------------------------------------------------------------------------------------------------------------------------------------------------------------------------------------------------------------------------------------------------------------------------------------------------------------------------------------------------------------------------------------------------------------------------------------------------------------------------------------------------------------------------------------------------------------------------------------------------------------------------------------------------------------------------------------------------|
| Deleted:                                                                                                                                                                                                                                                                                                                                                                                                                                                                                                                                                                                                                                                                                                                                                                                                                                                                    |
| Deleted: the                                                                                                                                                                                                                                                                                                                                                                                                                                                                                                                                                                                                                                                                                                                                                                                                                                                                |
| Deleted: reanalyses                                                                                                                                                                                                                                                                                                                                                                                                                                                                                                                                                                                                                                                                                                                                                                                                                                                         |
| Deleted: 8                                                                                                                                                                                                                                                                                                                                                                                                                                                                                                                                                                                                                                                                                                                                                                                                                                                                  |
| Deleted: 9                                                                                                                                                                                                                                                                                                                                                                                                                                                                                                                                                                                                                                                                                                                                                                                                                                                                  |
| Deleted: 9                                                                                                                                                                                                                                                                                                                                                                                                                                                                                                                                                                                                                                                                                                                                                                                                                                                                  |
| Deleted: 0                                                                                                                                                                                                                                                                                                                                                                                                                                                                                                                                                                                                                                                                                                                                                                                                                                                                  |
| Deleted: i                                                                                                                                                                                                                                                                                                                                                                                                                                                                                                                                                                                                                                                                                                                                                                                                                                                                  |
| Deleted: s                                                                                                                                                                                                                                                                                                                                                                                                                                                                                                                                                                                                                                                                                                                                                                                                                                                                  |
| Deleted: /                                                                                                                                                                                                                                                                                                                                                                                                                                                                                                                                                                                                                                                                                                                                                                                                                                                                  |
| Formatted: Superscript                                                                                                                                                                                                                                                                                                                                                                                                                                                                                                                                                                                                                                                                                                                                                                                                                                                      |
| Deleted: except below 2 km where the difference is greater than $\pm 0.5$ cm s -1 cm/s during January and February                                                                                                                                                                                                                                                                                                                                                                                                                                                                                                                                                                                                                                                                                                                                        |
| Deleted: In this case, we took ERAi as a reference
and compare it with w products from other
reanalyses. We chose ERAi, because the zonal and
meridional winds from this reanalysis have been
shown to compare well with radiosonde and rocket
sounding observations over the Indian equatorial
region ( Das et al. , 2015). The solid lines in Figure 8
show the differences over Gadanki, while the dashed
lines show differences over Gadanki, while the dashed
lines show differences over Kototabang. Over
Gadanki, the difference between the ERAi and other
reanalyses is less than $\pm 0.5$ cm/s during December-
January-February (DJF, winter). ERAi
underestimates ERA5 compared to other reanalyses,
while values based on MERRA-2 are relatively
larger than those in other reanalyses. During MAM |
| Deleted: cm/s                                                                                                                                                                                                                                                                                                                                                                                                                                                                                                                                                                                                                                                                                                                                                                                                                                                               |
| Deleted: -                                                                                                                                                                                                                                                                                                                                                                                                                                                                                                                                                                                                                                                                                                                                                                                                                                                                  |
| Deleted: cm/s                                                                                                                                                                                                                                                                                                                                                                                                                                                                                                                                                                                                                                                                                                                                                                                                                                                               |

| 665  | difference is comparatively high in June compared to July and August. NCEP/DOE-2 and                    |   |                                                             |
|------|---------------------------------------------------------------------------------------------------------|---|-------------------------------------------------------------|
| 666  | ERA5 are more and other reanalyses are less than the ensemble, however during July and                  |   | Deleted: -                                                  |
| 667  | August NCEP/DOE-2 it is less in the upper troposphere (10-18 km). Merra-2 and ERAi                      |   |                                                             |
| 668  | shows a good comparison with respect to the ensemble during July and August, JRA-55 also                | _ | Deleted: in                                                 |
| 669  | shows a good comparison in addition to Merra-2 and ERAi. During SON, the differences are                |   |                                                             |
| 670  | comparatively less than MAM and JJA. The difference is less than $\pm 0.5$ cm s -1 during    |   | Deleted: cm/s                                               |
| 671  | October and November except in September between 10 and 15 km where ERA5 and Merra-                     | _ | Deleted: -                                                  |
| 672  | 2 are more and ERAi and NCEP/DOE-2 are less than the ensemble. In general, ERA5 and                     | _ | Deleted: -                                                  |
| 673  | NCEP/DOE-2 shows considerably more difference with the ensemble and other reanalyses,                   | < | Deleted: i                                                  |
| 674  | (ERAi, Merra-2 and JRA-55) compare well with the ensemble.                                              |   | Deleted: s                                                  |
| 675  | Over Kototabang (Figure 11), it is interesting to note the difference between the                       |   | Deleted: 9                                                  |
| 676  | ensemble and different reanalyses show a consistent pattern during all the months. JRA-55               |   | Deleted: 0                                           |
| 677  | and ERAi show good comparison with the ensemble, as the differences are less than $\pm 0.2$ cm          |   | Deleted: a                                                  |
| 678  | $s^{-1}$ jn all the seasons, except in November where it exceeds $\pm 0.5$ cm $s^{-1}$ jn the lower and |   | Deleted: cm/s                                               |
| 679  | middle troposphere. Merra-2 is more and NCEP/DOE-2 is less than the ensemble at all the                 |   | Deleted: cm/s                                               |
| 67.5 | height regions. EDA5 is less helew 10 km and more shows with respect to the grouphle                    |   | Deleted                                                     |
| 680  | neight regions. ERAD is less below 10 km and more above with respe

---

## Author Response (AR2)

**Response to Editor's Comments**

We would like to sincerely thank the Executive Editor, Editor (Dr.Gabriele Stiller) and all the referees for kind suggestions and comments, which helped in revising the manuscript. We have addressed all the reviewers' comments in order to make the manuscript publishable in your esteemed journal "Atmospheric Chemistry and Physics (ACP)".

Point-by-point response on how we have addressed each recommendations/suggestions is given in the reply to the reviewer's comments and same is also implemented in the revised manuscript.

Now we are herewith submitting the following for the consideration of publication:

- (1) Replies to the reviewer's comments (in .pdf)
- (2) Track change manuscript along with figures and tables (in .pdf)
- (3) Revised manuscript with figures and tables (in LaTex)

All the authors listed on the manuscript concur with submission of the above mentioned manuscript.

We request Executive Editor and Editor to kindly process further and do the needful.

--END---

**Response to Reviewer-1's comments**

The authors have revised the manuscript reasonably in response to my first review. Below, I list several minor suggestions, which the authors may consider when they prepare the final manuscript. (The following line numbers refer to those in acp-2020-18-author\_response-version2.pdf, i.e., the changes-tracked version.)

We would like to sincerely thank the referee-1 for the second evaluation and very positive and constructive suggestions and recommendation for publication. We have implemented the suggestions raised by the referee.

Point-by-point responses on how we have addressed each recommendation/ suggestions are given below.

**Q1:** *Lines 14: add ", India"* **R1: Corrected in the revised manuscript.**

**Q2:** *Lines 15: add ", Indonesia"* **R2: Corrected in the revised manuscript.**

Q3: Lines 21-22: I think this sentence needs clarification. One possibility is ". . . in the w when testing different spatial sampling for reanalysis data around the Gadanki station." R3: Corrected in the revised manuscript.

**Q4:** *Line 168: add ", India" and ", Indonesia"* **R4: Added in the revised manuscript.**

**Q5:** Line 188, Table 1: add the information on the horizontal extent (in km or  $km^2$ ) of the radar sampling volume at e.g., 10 km and 20 km altitudes.

**R5:** The sampling volume is 0.85 km2 at 10 km and 3.4 km2 at 20 km.

**Q6:** *Line 214: Change "daily mean profiles" to "daily 16:30-17:30 IST (11:00-12:00 UTC) averaged profiles"* **R6: Changed in the revised manuscript.**

**Q7:** *Line 254: "This wind shear" – add the information on the altitude* **R7: Added in the revised manuscript.**

**Q8:** *Lines* 255-256: *Do you mean "above 6-7 km altitudes"*? *If so, please explicitly write so.* **R8:** Added in the revised manuscript.

**Q9:** *Line 415: "long-period" – please explicitly write the temporal scale.* **R9: It is from 1986 to 1989. Now added in the revised manuscript.**

**Q10:** Line 416: "long-term" – again, please explicitly write the temporal scale. **R10:** Gage et al. (1991) has taken three years of data and mentioned it as long-term mean. Corrected in the revised manuscript. **Q11:** Line 480: "long-term" – same as above. "diurnal" – is this "day-to-day"? ("diurnal" may mean 1-day (and 0.5-day) periodicity)

**R11:** Corrected in the revised manuscript.

**Q12:** *Line 715: "any" – did you do the same analysis for other reanalyses? Or, only ERA-Interim?*

**R12:** Yes, it is true for all the re-analysis data. We have done the same analysis for all the re-analyses data and the results are consistence.**

**Q13:** Lines 717-718: Change this sentence to: "We therefore show the directional tendencies of reanalysis data relative to the radar measurements." Here, I assume that the authors use the term "tendencies" as the ratio that reanalysis reproduces (i.e., agree with) radar measurements in terms of vertical wind direction. Also, I think we need one more sentence, right after this, explaining/defining what is "directional tendencies" more clearly, or please add something like: "The directional tendencies would be 100% when all radar measurements at certain height range are reproduced by a reanalysis in terms of vertical wind direction."

**R13:** Revised in the manuscript.**

**Q14:** *Lines* 726-951: *All* "produce/producing/produced" should be changed to "reproduce/reproducing/ reproduced".

**R14:** Corrected in the revised manuscript.

Q15: *Line 1004: "any": again, did you do the same analysis for other reanalyses?* R15: It will be true for all reanalyses. We have changed the sentences accordingly.

**Q16:** *Line 1010: change "care" to "caution"* **R16: Corrected in the revised manuscript.**

**Q17:** *Figure 2, caption: change "MST Radar" to "IMSTR"* **R17: Corrected in the revised manuscript.**

**Q18:** Figure 4, caption: change "MST Radar" to "IMSTR"

**R18:** Corrected in the revised manuscript.

**Q19:** Figure 10, caption: add the following sentence (between the two sentences), "The reference is the reanalysis ensemble mean."

**R19:** The figure caption is revised by following the reviewer's suggestion.

**Q20:** *Figure 12, caption: add the information that these are the ERAi cases.* **R20: The figure caption is revised.**

**Q21:** Table 2: Change "Kobayachi" to "Kobayashi". Also, commas after the author names are unnecessary if "(year)" is used.

**R21:** Corrected in the revised manuscript.

--END---

**Response to Reviewer-2's comments**

Overall the manuscript provides a novel intercomparison of reanalysis vertical velocities with ground-based measurements from 2 subtropical VHF radar locations. The study illustrates how widely vertical velocities can vary among reanalyses and observations, meaning that caution should be used when interpreting results from studies using reanalysis-based vertical velocities in the troposphere and lower stratosphere.

We would like to sincerely thank anonymous referee-2 for the evaluation and providing constructive comments/suggestions, which helped to improve the manuscript considerably.

Point-by-point responses to the reviewer's comments are given below. Please note that changes are also made in the revised manuscript by taking consideration of referee #1s' comments.

**Q1:** First, throughout the manuscript the authors should emphasize that this study focuses on a very limited geographical area, so the results do not necessarily apply for reanalyses in general. I think this is important, especially given the conclusion (lines 440-442) that the results somehow provide an initial basis to improve calculation of w in reanalyses. Instead, I would say that the results demonstrate that how approaches to generating global reanalysis products (encompassing different models, assimilation methods, spatial resolution, etc) can impact estimates of w. I think this is important since providing uncertainty estimates for derived meteorological products like w is currently needed by the SPARC community.

**R1:** We do agree with the referee's assertion. Now we have re-written the concluding remarks in the revised manuscript.**

**Q2:** In that spirit, I think one thing that is currently missing from this paper, which is needed before I can recommend publication, is a quantitative discussion of the uncertainties in the retrieved vertical velocities from the radar. As other referees have commented, the authors do mention sources of uncertainty but I don't have an idea for what a typical error bar would be. For example, can the vertical velocities from an individual profile be determined with an accuracy of a cm/s or less? Perhaps this is described in other papers, but it needs to be discussed here so we can make sense of the comparisons with w from reanalyses, and perhaps also included in Table 1?

**R2:** Following the reviewer's suggestion, we have provided a separate sub-section in the revised manuscript for accuracy and uncertainty in the measurements of w from. We request reviewer to kindly follow the revised manuscript for details. We thank referee for the suggestion.

**We have also included the velocity resolution by EAR and IMSTR in Table 1.**

**Q3:** Another item that must be addressed is the process for computing vertical velocities from reanalyses on an altitude grid. Again, earlier referee comments requested clarification, particularly because vertical velocity is essentially a model-produced variable and so is subject to the details of each system (model vertical coordinate, vertical resolution especially in the TTL region, in addition to model physical parameterizations, etc). I only saw a brief mention in one of the author responses that the conversion to altitude is done using the hypsometric equation. Details are needed, and I'm not sure that is the best way to do things, if the authors are saying the performed some kind of integration themselves using temperature profiles to determine geometrical altitude (z). The most straightforward method would seem to be using the reanalysis Geopotential height fields to specify the altitude of each pressure level at each grid point where pressure velocity is evaluated. Can the authors please provide specific details about how geometric altitude conversion was performed. This could potentially clear up any underlying biases or disagreements among the observations and reanalyses w profiles.

**R3:** Reanalysis gives the omega at different pressure levels. The pressure is converted to geometric altitude using the hypsometric equation which is (1)

 $P=P_0exp^{(-Z/H)}$

where, P is the pressure at a particular altitude and  $P_0$  is the surface pressure. Z is the height and H is the scale height.

After the pressure conversion to height, omega is converted into vertical velocity using the equation

$$\mathbf{W} = -\frac{1}{g}\omega \frac{RT}{p} \tag{2}$$

**Hence no integration or interpolation of omega is done.**

**Q4:** Finally, the manuscript does not mention that one very large source of observations in the troposphere and lower stratosphere come from radiosondes (in fact, a search of the manuscript finds no mention of radiosondes at all, which I find quite surprising). First, I believe radiosondes can provide vertical wind information that is directly assimilated into these analysis systems. I would assume there have been comparisons between the VHR radar vertical velocities and nearby radiosonde observations (a very brief search provides many results, including some early studies by some of the coauthors of the present manuscript). If so, these should be mentioned, and how do they compare? This would further support the validity of the radar observations and also further highlight possible issues with reanalyses (i.e, if they assimilate radiosonde vertical velocities and still giving different results, that's an issue).

**R4:** There are few studies that have calculated vertical velocity from dropsondes and radiosondes. Wang et al. (2009) derived the vertical velocity from radiosonde and dropsondes, however the authors themselves have pointed out the several uncertainties like requirement of high resolution radiosonde data, amount of helium gas associated with such retrievals and accuracy of the estimated vertical velocity was not quantified. Zhang et al. (2019) estimated vertical velocity using a descending radiosonde system. Here also the authors have pointed out the uncertainties involved especially with the radiosonde descent speed, calculation of drag coefficient and also on the validation of the retrieval's on vertical velocity obtained. However, reanalysis does not assimilate the vertical velocity obtained from the radiosonde, it only derives the vertical wind using the horizontal wind divergence. The horizontal winds from the radiosonde are well assimilated in all the reanalysis system. Now we have briefly discussed the above in the revised manuscript.

Q5: Throughout the manuscript please use UTC consistently instead of UTS and GMT. R5: Corrected in the revised manuscript.

Q6: Make sure all figure captions clearly indicate the time period of the results. This is missing from, e.g., caption of Figs. 4, 5, 8, 9, 10, 11, 12, 13.

**R6:** Implemented in the revised manuscript.

Q7: Lines 51-52: This statement clearly is not true for aircraft measurements. Please revise to more accurately capture what the relevant point is (e.g., that independent ground based observations are limited in their geographic distribution).

**R7:** Modified in the revised manuscript.

Q8: lines 89-91: Meteor radar is a technique for the mesosphere, and is not applicable in the present study, this should be removed.

**R8:** Removed in the revised manuscript.

Q9: The abstract and introduction should clearly state that this study is focusing on the troposphere and lower stratosphere. I had to read quite far into the manuscript to determine the scope of the study. Since this is submitted to the SRIP special issue, revisions that specifically address the relevance of this study to SRIP would be greatly beneficial.

**R9:**Following the reviewer's suggestion we have modified the abstract and introduction in the revised manuscript. A brief description of SPARC/s-rip is provided also described.

Q10: Lines 101-102 and elsewhere: Throughout the manuscript the authors use the term tendency to describe the sign of the vertical velocity (positive or negative). This is in opposition to the usage I am most familiar with, i.e., a time tendency, or specifically, a time derivative. I think it is better to say here that the best way (in the authors' opinion) is to evaluate time-mean profiles, which is what they are doing in figures 4, 5, 6, etc.

**R10:** Following the reviewer-1's comment (Q13), we have includes a brief note on directional tendency in the revised manuscript.

**References :**

Wang, J., J. Bian, W. O. Brown, H. Cole, V. Grubisic and K. Young.: Vertical air motion from T-REX radiosonde and dropsonde data, J. atmos. Oce. Tech., 26, 928-942, https://doi:10.1175/2008JTECHA1240.1, 2009

Zhang, J., H. Chen, Y. Zhu, H. Shi, Y. Zheng, X. Xia, Y. Teng, F. Wang, X. Han, J. Li and Y. Xuan.: 2019, 
[revised manuscript text omitted]
.                                                                                                                                                                                                                                                                                                                                                                                                               | ſ       |  |  |  |
| Formatted: Font: Not Italic, Complex Script
Font: Not Italic                                                                                                                                                                                                                                                                                                                                                                                                                                                                              |         |  |  |  |
| Formatted: Font: Not Italic, Complex Script
Font: Not Italic                                                                                                                                                                                                                                                                                                                                                                                                                                                                              |         |  |  |  |
| Deleted: ¶                                                                                                                                                                                                                                                                                                                                                                                                                                                                                                                                   |         |  |  |  |
| Formatted: Font: Not Italic, Complex Script
Font: Not Italic                                                                                                                                                                                                                                                                                                                                                                                                                                                                              |         |  |  |  |
| Formatted: Font: Not Italic, Complex Script
Font: Not Italic                                                                                                                                                                                                                                                                                                                                                                                                                                                                       |         |  |  |  |
|                                                                                                                                                                                                                                                                                                                                                                                                                                                                                                                                              |         |  |  |  |

[revised manuscript text omitted]

Roman, Bold, Italic, Font color: Text 1, Complex Script Font: Bold

| Moved (insertion) [3]                                                                                                                                                                                                                                                                                                                      |
|--------------------------------------------------------------------------------------------------------------------------------------------------------------------------------------------------------------------------------------------------------------------------------------------------------------------------------------------|
| Deleted: The resolution of velocity using IMSTR is about 0.03 m s -1 if FFT points are 256, Coherent integration is 64 and inter-pulse period is 1000 $\mu$ s -1 and about 0.002 m s -1 if FFT points are 256, Coherent integration is 512 and inter-pulse period is 1000 $\mu$ s -1 .¶ |
| Formatted: Font: Italic, Complex Script Font: Italic                                                                                                                                                                                                                                                                                |
| Formatted: Font: Not Italic, Complex Script
Font: Not Italic                                                                                                                                                                                                                                                                            |
| Moved (insertion) [2]                                                                                                                                                                                                                                                                                                                      |
| Deleted: the                                                                                                                                                                                                                                                                                                                               |

234 (Fukao et al., 2003). The uncertainty in the w due to beam pointing error by an angle ( $\theta$ ) with a horizontal wind ( $\mu$ ) is given by ( $\mu$ .sin  $\theta$ ). Thus, with a horizontal wind of 10 m s-1 and 235 beam pointing error of 0.2 degree turns out to be 0.03 m s-1 uncertainty in the w measured 236 237 from VHF radar. The beam pointing accuracy can further be determined by comparing the vertical wind obtained using two orthogonal polarizations, i.e., east-west and north-south 238 239 polarizations, which are phased independently, Significant correlation was observed between both the polarizations, suggesting that the radar measures the true vertical velocity 240 241 (Viswanathan et al., 1993, In addition, Rao et al. (2008) also estimated the vertical velocities 242 from zenith beam and compared it with those estimated from 10-degree off-zenith beams using IMSTR. The differences were observed to be meager, which shows that the error due to 243 244 beam pointing is negligible.

245 Tilting of reflecting layers contributing to the diffuse reflection can also adversely bias 246 in the mean w (Röttger, 1980). These tilting layers can be due to the presence of Kelvin-247 Helmholtz instabilities(Muschinski, 1996), gravity waves, which includes inertia-gravity 248 waves and mountain waves and causes imbalance in the echo power between the two 249 polarizations in the same plane (Yamamoto et al. 2003). Rao et al. (2008) estimated the echo power imbalance in the east-west and north-south polarizations for both EAR and IMSTR 250 251 and found the difference to be within  $\pm 1$  dB, statistically indicating the bias due to the tilting layers is negligible over both the locations. 252

253 Nastrom and VanZandt (1994) proposed that *w* can be biased by gravity waves. Thus,
254 Rao et al. (2008) have investigated the biases caused by gravity waves by calculating the
255 variances and found that downward wind measurements below 10 km are essentially
256 unaffected by gravity waves. It is also to be noted that the topography over the two locations
257 can generate mountain waves, if strong low-level winds are prevailing. Strong low-level
258 winds are prevalent over Gadanki only from June to August and during these months, there is

| Formatted: Font: Italic, Complex Script Font: Italic                                                                                                                                                                                                                                                                                                                                                                                                                                                                                                                                                                                                                                                                                                                                                                                                                                                                                                                                                                                                                                                                                                                                                                                                                                                                                                                                                                                                                                                                                                                                                                                                                                                                                                                                                                                                                              |                                       |
|------------------------------------------------------------------------------------------------------------------------------------------------------------------------------------------------------------------------------------------------------------------------------------------------------------------------------------------------------------------------------------------------------------------------------------------------------------------------------------------------------------------------------------------------------------------------------------------------------------------------------------------------------------------------------------------------------------------------------------------------------------------------------------------------------------------------------------------------------------------------------------------------------------------------------------------------------------------------------------------------------------------------------------------------------------------------------------------------------------------------------------------------------------------------------------------------------------------------------------------------------------------------------------------------------------------------------------------------------------------------------------------------------------------------------------------------------------------------------------------------------------------------------------------------------------------------------------------------------------------------------------------------------------------------------------------------------------------------------------------------------------------------------------------------------------------------------------------------------------------------------------------|---------------------------------------|
| Deleted: The error caused                                                                                                                                                                                                                                                                                                                                                                                                                                                                                                                                                                                                                                                                                                                                                                                                                                                                                                                                                                                                                                                                                                                                                                                                                                                                                                                                                                                                                                                                                                                                                                                                                                                                                                                                                                                                                                                                | -                                     |
| Formatted: Font: Italic, Complex Script Font:
Italic                                                                                                                                                                                                                                                                                                                                                                                                                                                                                                                                                                                                                                                                                                                                                                                                                                                                                                                                                                                                                                                                                                                                                                                                                                                                                                                                                                                                                                                                                                                                                                                                                                                                                                                                                                                                                                  |                                       |
| Deleted: by the beam pointing, due to the uncertainty in the                                                                                                                                                                                                                                                                                                                                                                                                                                                                                                                                                                                                                                                                                                                                                                                                                                                                                                                                                                                                                                                                                                                                                                                                                                                                                                                                                                                                                                                                                                                                                                                                                                                                                                                                                                                                                      |                                       |
| Formatted                                                                                                                                                                                                                                                                                                                                                                                                                                                                                                                                                                                                                                                                                                                                                                                                                                                                                                                                                                                                                                                                                                                                                                                                                                                                                                                                                                                                                                                                                                                                                                                                                                                                                                                                                                                                                                                                                |                                       |
| Deleted: of                                                                                                                                                                                                                                                                                                                                                                                                                                                                                                                                                                                                                                                                                                                                                                                                                                                                                                                                                                                                                                                                                                                                                                                                                                                                                                                                                                                                                                                                                                                                                                                                                                                                                                                                                                                                                                                                              |                                       |
| Formatted                                                                                                                                                                                                                                                                                                                                                                                                                                                                                                                                                                                                                                                                                                                                                                                                                                                                                                                                                                                                                                                                                                                                                                                                                                                                                                                                                                                                                                                                                                                                                                                                                                                                                                                                                                                                                                                                                | -                                     |
| Deleted: using the relation¶                                                                                                                                                                                                                                                                                                                                                                                                                                                                                                                                                                                                                                                                                                                                                                                                                                                                                                                                                                                                                                                                                                                                                                                                                                                                                                                                                                                                                                                                                                                                                                                                                                                                                                                                                                                                                                                      |                                       |
| Formatted                                                                                                                                                                                                                                                                                                                                                                                                                                                                                                                                                                                                                                                                                                                                                                                                                                                                                                                                                                                                                                                                                                                                                                                                                                                                                                                                                                                                                                                                                                                                                                                                                                                                                                                                                                                                                                                                                |                                       |
| Deleted: in                                                                                                                                                                                                                                                                                                                                                                                                                                                                                                                                                                                                                                                                                                                                                                                                                                                                                                                                                                                                                                                                                                                                                                                                                                                                                                                                                                                                                                                                                                                                                                                                                                                                                                                                                                                                                                                                              |                                       |
| Formatted                                                                                                                                                                                                                                                                                                                                                                                                                                                                                                                                                                                                                                                                                                                                                                                                                                                                                                                                                                                                                                                                                                                                                                                                                                                                                                                                                                                                                                                                                                                                                                                                                                                                                                                                                                                                                                                                                |                                       |
| Deleted: polarization with that of vertical wind                                                                                                                                                                                                                                                                                                                                                                                                                                                                                                                                                                                                                                                                                                                                                                                                                                                                                                                                                                                                                                                                                                                                                                                                                                                                                                                                                                                                                                                                                                                                                                                                                                                                                                                                                                                                                                  |                                       |
| Formatted                                                                                                                                                                                                                                                                                                                                                                                                                                                                                                                                                                                                                                                                                                                                                                                                                                                                                                                                                                                                                                                                                                                                                                                                                                                                                                                                                                                                                                                                                                                                                                                                                                                                                                                                                                                                                                                                                |                                       |
| Deleted: The correlation was observed to be hig                                                                                                                                                                                                                                                                                                                                                                                                                                                                                                                                                                                                                                                                                                                                                                                                                                                                                                                                                                                                                                                                                                                                                                                                                                                                                                                                                                                                                                                                                                                                                                                                                                                                                                                                                                                                                                   |                                       |
| Deleted: ed that                                                                                                                                                                                                                                                                                                                                                                                                                                                                                                                                                                                                                                                                                                                                                                                                                                                                                                                                                                                                                                                                                                                                                                                                                                                                                                                                                                                                                                                                                                                                                                                                                                                                                                                                                                                                                                                                         |                                       |
| Formatted                                                                                                                                                                                                                                                                                                                                                                                                                                                                                                                                                                                                                                                                                                                                                                                                                                                                                                                                                                                                                                                                                                                                                                                                                                                                                                                                                                                                                                                                                                                                                                                                                                                                                                                                                                                                                                                                                |                                       |
| Formatted                                                                                                                                                                                                                                                                                                                                                                                                                                                                                                                                                                                                                                                                                                                                                                                                                                                                                                                                                                                                                                                                                                                                                                                                                                                                                                                                                                                                                                                                                                                                                                                                                                                                                                                                                                                                                                                                                |                                       |
| Formatted                                                                                                                                                                                                                                                                                                                                                                                                                                                                                                                                                                                                                                                                                                                                                                                                                                                                                                                                                                                                                                                                                                                                                                                                                                                                                                                                                                                                                                                                                                                                                                                                                                                                                                                                                                                                                                                                                |                                       |
| Formatted                                                                                                                                                                                                                                                                                                                                                                                                                                                                                                                                                                                                                                                                                                                                                                                                                                                                                                                                                                                                                                                                                                                                                                                                                                                                                                                                                                                                                                                                                                                                                                                                                                                                                                                                                                                                                                                                                |                                       |
| Deleted: ). The authors found a difference $< 0.1$                                                                                                                                                                                                                                                                                                                                                                                                                                                                                                                                                                                                                                                                                                                                                                                                                                                                                                                                                                                                                                                                                                                                                                                                                                                                                                                                                                                                                                                                                                                                                                                                                                                                                                                                                                                                                                |                                       |
| Moved (insertion) [4]                                                                                                                                                                                                                                                                                                                                                                                                                                                                                                                                                                                                                                                                                                                                                                                                                                                                                                                                                                                                                                                                                                                                                                                                                                                                                                                                                                                                                                                                                                                                                                                                                                                                                                                                                                                                                                                                    |                                       |
| Deleted: vertical enith beams using IMSTR. T                                                                                                                                                                                                                                                                                                                                                                                                                                                                                                                                                                                                                                                                                                                                                                                                                                                                                                                                                                                                                                                                                                                                                                                                                                                                                                                                                                                                                                                                                                                                                                                                                                                                                                                                                                                                                                      | -                                     |
|                                                                                                                                                                                                                                                                                                                                                                                                                                                                                                                                                                                                                                                                                                                                                                                                                                                                                                                                                                                                                                                                                                                                                                                                                                                                                                                                                                                                                                                                                                                                                                                                                                                                                                                                                                                                                                                                                          | ÷                                     |
| Moved up [3]: The resolution of velocity usin                                                                                                                                                                                                                                                                                                                                                                                                                                                                                                                                                                                                                                                                                                                                                                                                                                                                                                                                                                                                                                                                                                                                                                                                                                                                                                                                                                                                                                                                                                                                                                                                                                                                                                                                                                                                                                            |                                       |
| Moved up [3]: The resolution of velocity using                                                                                                                                                                                                                                                                                                                                                                                                                                                                                                                                                                                                                                                                                                                                                                                                                                                                                                                                                                                                                                                                                                                                                                                                                                                                                                                                                                                                                                                                                                                                                                                                                                                                                                                                                                                                                                           |                                       |
| Moved up [3]: The resolution of velocity using Formatted Formatted                                                                                                                                                                                                                                                                                                                                                                                                                                                                                                                                                                                                                                                                                                                                                                                                                                                                                                                                                                                                                                                                                                                                                                                                                                                                                                                                                                                                                                                                                                                                                                                                                                                                                                                                                                                                                       |                                       |
| Moved up [3]: The resolution of velocity using
Formatted
Formatted
Formatted                                                                                                                                                                                                                                                                                                                                                                                                                                                                                                                                                                                                                                                                                                                                                                                                                                                                                                                                                                                                                                                                                                                                                                                                                                                                                                                                                                                                                                                                                                                                                                                                                                                                                                                                                                                                    | · · · · · ·                           |
| Moved up [3]: The resolution of velocity using
Formatted
Formatted
Formatted
Moved (insertion) [5]                                                                                                                                                                                                                                                                                                                                                                                                                                                                                                                                                                                                                                                                                                                                                                                                                                                                                                                                                                                                                                                                                                                                                                                                                                                                                                                                                                                                                                                                                                                                                                                                                                                                                                                                                                           | ·                                     |
| Moved up [3]: The resolution of velocity using
Formatted
Formatted
Formatted
Moved (insertion) [5]
Deleted: As proposed by                                                                                                                                                                                                                                                                                                                                                                                                                                                                                                                                                                                                                                                                                                                                                                                                                                                                                                                                                                                                                                                                                                                                                                                                                                                                                                                                                                                                                                                                                                                                                                                                                                                                                                                                                | ·                                     |
| Moved up [3]: The resolution of velocity using                                                                                                                                                                                                                                                                                                                                                                                                                                                                                                                                                                                                                                                                                                                                                                                                                                                                                                                                                                                                                                                                                                                                                                                                                                                                                                                                                                                                                                                                                                                                                                                                                                                                                                                                                                                                                                           | ·                                     |
| Moved up [3]: The resolution of velocity using
Formatted
Formatted
Formatted
Moved (insertion) [5]
Formatted                                                                                                                                                                                                                                                                                                                                                                                                                                                                                                                                                                                                                                                                                                                                                                                                                                                                                                                                                                                                                                                                                                                                                                                                                                                                                                                                                                                                                                                                                                                                                                                                                                                                                            | · · · · · · · · · · · · · · · · · · · |
| Moved up [3]: The resolution of velocity using
Formatted
Formatted
Formatted
Moved (insertion) [5]
Formatted
Deleted: on the                                                                                                                                                                                                                                                                                                                                                                                                                                                                                                                                                                                                                                                                                                                                                                                                                                                                                                                                                                                                                                                                                                                                                                                                                                                                                                                                                                                                                                                                                                                                                                                                                                                                         |                                       |
| Moved up [3]: The resolution of velocity using
Formatted
Formatted
Formatted
Moved (insertion) [5]
Formatted
Formatted
Formatted                                                                                                                                                                                                                                                                                                                                                                                                                                                                                                                                                                                                                                                                                                                                                                                                                                                                                                                                                                                                                                                                                                                                                                                                                                                                                                                                                                                                                                                                                                                                                                                                                                                                                      |                                       |
| Moved up [3]: The resolution of velocity using         Formatted         Formatted         Formatted         Moved (insertion) [5]         Deleted: As proposed by         Formatted         Formatted         Indent: First line: 0.4"         Formatted         Deleted: on the         Formatted         Image: Second Sec                                                                                                                  |                                       |
| Moved up [3]: The resolution of velocity using
Formatted
Formatted
Formatted
Moved (insertion) [5]
Formatted
Poleted: Indent: First line: 0.4"
Formatted
Formatted
Formatted
Formatted
Formatted
Formatted
Formatted
Formatted
Formatted
Formatted
Formatted
Formatted
Formatted
Formatted                                                                                                                                                                                                                                                                                                                                                                                                                                                                                                                                                                                                                                                                                                                                                                                                                                                                                                                                                                                                                                                                                                                                                                                                                                                                                                                                                                                                                                                                                     |                                       |
| Moved up [3]: The resolution of velocity using
Formatted
Formatted
Formatted
Moved (insertion) [5]
Formatted
Formatted
Formatted
Deleted: It is known that estimates of w derived                                                                                                                                                                                                                                                                                                                                                                                                                                                                                                                                                                                                                                                                                                                                                                                                                                                                                                                                                                                                                                                                                                                                                                                                                                                                                                                                                                                                                                                                                                                                                        |                                       |
| Moved up [3]: The resolution of velocity using
Formatted
Formatted
Formatted
Moved (insertion) [5]
Formatted
Pormatted
Formatted
Formatted
Formatted: Font color: Text 1                                                                                                                                                                                                                                                                                                                                                                                                                                                                                                                                                                                                                                                                                                                                                                                                                                                                                                                                                                                                                                                                                                                                                                                                                                                                                                                                                                                                                                                                                                                                                 |                                       |
| Moved up [3]: The resolution of velocity using
Formatted
Formatted
Formatted
Formatted
Moved (insertion) [5]
Formatted
Formatted
Formatted
Formatted Fort color: Text 1
Moved up [2]: The beam pointing error is four                                                                                                                                                                                                                                                                                                                                                                                                                                                                                                                                                                                                                                                                                                                                                                                                                                                                                                                                                                                                                                                                                                                                                                                                                                                                                                                                                                                                                                                          |                                       |
| Moved up [3]: The resolution of velocity using
Formatted
Formatted
Formatted
Moved (insertion) [5]
Formatted
Formatted
Formatted
Moved up [2]: The beam pointing error is four
Formatted                                                                                                                                                                                                                                                                                                                                                                                                                                                                                                                                                                                                                                                                                                                                                                                                                                                                                                                                                                                                                                                                                                                                                                                                                                                                                                                                                                                                                                                        |                                       |
| Moved up [3]: The resolution of velocity using
Formatted
Formatted
Formatted
Formatted
Moved (insertion) [5]
Formatted
Formatted
Formatted
Formatted
Formatted
Poleted: I                                                                                                                                                                                                                                                                                                                                                                                                                                                                                                                                                                                                                                                                                                                                                                                                                                                                                                                                                                        |                                       |
| Moved up [3]: The resolution of velocity using
Formatted
Formatted
Formatted
Moved (insertion) [5]
Formatted
Formatted
Formatted
Formatted
Formatted
Formatted
Formatted
Formatted
Formatted
Formatted
Formatted
Formatted
Formatted
Formatted
Formatted
Formatted
Formatted
Formatted
Formatted
Formatted
Formatted
Formatted
Formatted
Formatted
Formatted
Formatted
Formatted
Formatted
Formatted
Formatted
Formatted
Formatted
Formatted
Formatted
Formatted
Formatted
Formatted
Formatted
Formatted
Formatted
Formatted
Formatted
Formatted
Formatted
Formatted
Formatted
Formatted
Formatted
Formatted
Formatted
Formatted
Formatted
Formatted
Formatted
Formatted
Formatted
Formatted
Formatted
Formatted
Formatted
Formatted
Formatted
Formatted
Formatted
Formatted
Formatted
Formatted
Formatted
Formatted
Formatted
Formatted
Formatted
Formatted
Formatted
Formatted
Formatted
Formatted
Formatted
Formatted
Formatted
Formatted
Formatted
Formatted
Formatted
Formatted
Formatted
Formatted
Formatted
Formatted
Formatted
Formatted
Formatted
Formatted
Formatted
Formatted
Formatted
Formatted
Formatted
Formatted
Formatted
Formatted
Formatted
Formatted
Formatted
Formatted
Formatted
Formatted
Formatted
Formatted
Formatte |                                       |
| Moved up [3]: The resolution of velocity using
Formatted
Formatted
Formatted
Moved (insertion) [5]
Formatted
Formatted
Formatted
Formatted
Formatted
Formatted
Formatted
Formatted
Formatted
Deleted: I formatted                                                                                                                                                                                                                                                                                                                                                                                                                                                                                                                                                                                                                                                                                                                                                                                                                                                                                                                                                                                                                                                                                                                                                              |                                       |
| Moved up [3]: The resolution of velocity using
Formatted
Formatted
Formatted
Formatted
Moved (insertion) [5]
Formatted
Formatted
Formatted
Formatted
Formatted
Formatted
Formatted
Formatted
Formatted
Formatted
Formatted
Formatted
Formatted
Formatted
Formatted
Formatted
Formatted
Formatted
Formatted
Formatted
Formatted
Formatted
Formatted
Formatted                                                                                                                                                                                                                                                                                                                                                                                                                                                                                                                                                                                                                                                                                                                                                                                                                                                                                                      |                                       |
| Moved up [3]: The resolution of velocity using
Formatted
Formatted
Formatted
Moved (insertion) [5]
Formatted
Formatted
Formatted
Formatted
Formatted
Formatted
Formatted
Moved up [4]: Rao et al. (2008) estimated the                                                                                                                                                                                                                                                                                                                                                                                                                                                                                                                                                                                                                                                                                                                                                                                                                                                                                                                                                                                                                                                                                                                                                                                                      |                                       |

[revised manuscript text omitted]

---

## Author Response (AR4)

**Response to Editor's Comments**

We would like to sincerely thank the Editor for sending the manuscript for third review.  Now we have addressed the comment raised in the last review by Reviewer-3. For this, we have carried out an analysis to show that our analysis is not biased, if we used Pressure height instead of Geometric height in the reanalysis data. More details are provided in the reply to the Reviewer-3's comment. However, following the reviewer's suggestion, we have mentioned in the revised manuscript that we are using Pressure height for all the reanalyses, which do not biases our results. More details are provided in the reply to the Reviewer-3's comment.

Now we are herewith submitting the following for the consideration of publication:
- Replies to the reviewer's comments
- Track change manuscript along with figures and tables (in .pdf)
- Revised manuscript with figures and tables (in LaTex)

We request Editor to kindly process further in order to make the manuscript publishable in your esteemed journal "Atmospheric Chemistry and Physics (ACP)".

All the authors listed on the manuscript concur with submission of the above mentioned manuscript.

**--END--**

**Q.** *I have reviewed the revised manuscript and author responses. My question about altitude registration has not been satisfactorily addressed. The radar retrieves vertical velocity as a function of geometric altitude. The formula provided in the response, and in the revised manuscript, for calculating altitude is "pressure altitude" that assumes a constant scale height value H (it is not stated anywhere what value of H is used). The "pressure height" is not the same as geometric height. The differences may be small, but I think it matters in something like Fig 4 and 5 where they are plotting altitude profiles. Since reanalyses routinely output geopotential height (GH), it should be trivial to use GH to put the w profiles from reanalyses on the same geometric altitude grid as the radar w profiles. Again this may not change the results dramatically, but also speaks to maintaining a certain level of scientific rigor in the process. So I think the authors should either compute geometric height for the reanalysis results, or at the very least describe that they are using "pressure height", provide the value of H they use, and demonstrate somehow that this does not introduce biases or errors in their comparisons.*

**R. We would like to sincerely thank the referee-3 for the third evaluation and for his suggestions to check the biases (if any), if we used Pressure height (used for reanalyses) instead of Geometric height (used for Radar). For this, we have done the following exercise:**

**We have taken a typical Radiosonde ascent (i-met radiosonde which is GPS based with pressure sensor) over Gadanki. GPS based radiosonde gives geometric height ($h_g$) from which we estimated geopotential height ($h$) using the following equation(1) :**

$$h = \left(\frac{r}{r+h_g}\right)h_g \qquad \qquad (1)$$

**where, $r$ is the radius of the earth.**

**Further, we estimate pressure height from radiosonde measured pressure using the Hypsometric equation (2):**

$$(h_{P2} - h_{P1}) = \frac{R_d T}{g_0} ln\left(\frac{P_1}{P_2}\right) \qquad \qquad (2)$$

**Where, $h_{P1}$ (base) and $h_{P2}$(top) (i.e., Pressure height) are the height at pressure $P_1$ and $P_2$ respectively, $R_d$ is gas constant, $g_0$ is acceleration due to gravity and $T$ is the absolute temperature. The scale height is varying with altitude in the equation (2).**

**Pressure versus geometric, geopotential and pressure heights do not so any significant differences (Fig1 a).**

**The difference between geometric and geopotential height is found to be <50 m below tropopause (Fig.1 c) , and the difference between geometric and pressure height is found to be <50 m below 10 km and above it is < 100 m (Fig.1 d), which is insignificant for the present analysis and do not bias the result as the radar**

resolution itself is 150 m.  Thus, we conclude that there will be no significant difference if we used Pressure height instead of Geometric height and do not bias the results.

[Figure]

**Figure 1.** *Typical height profile of (a) temperature, (b) variation of pressure w.r.t. geometric, geopotential and pressure height, (c) difference between geometric and geopotential height, and (d) difference between geometric and pressure height over Gadanki.*